# Sea salt emission, transport and influence on size-segregated nitrate simulation: a case study in Northwestern Europe by WRF-Chem

Ying Chen[1,2], Yafang Cheng[2], Nan Ma[1,2], Ralf Wolke[1], Stephan Nordmann[3], Stephanie Schüttauf[1], Liang Ran[4], Birgit Wehner[1], Wolfram Birmili[1,3], Hugo A. C. Denier van der Gon[5], Qing Mu[2], Stefan Barthel[1], Gerald Spindler[1], Bastian Stieger[1], Konrad Müller[1], Guang-Jie Zheng[6], Ulrich Pöschl[2], Hang Su[2] and Alfred Wiedensohler[1]

[1]Leibniz-Institute for Tropospheric Research, Leipzig, 04318, Germany
[2]Multiphase Chemistry Department, Max Planck Institute for Chemistry, Mainz, 55128, Germany
[3]German Environment Agency, Dessau-Roßlau, 06844, Germany
[4]Key Laboratory of Middle Atmosphere and Global Environment Observation, Institute of Atmospheric Physics, Chinese Academy of Sciences, Beijing, 100029, China
[5]TNO, Dept. of Climate, Air and Sustainability, Princetonlaan 6, 3584 CB Utrecht, the Netherlands
[6]State Key Joint Laboratory of Environment Simulation and Pollution Control, School of Environment, Tsinghua University, Beijing 100084, China

*Correspondence to*: Y. Cheng (yafang.cheng@mpic.de) and Y. Chen (chen@tropos.de)

**Abstract.** Sea salt aerosol (SSA) is one of the major components of primary aerosols and has significant impact on the formation of secondary inorganic particles mass on a global scale. In this study, the fully online coupled WRF-Chem model was utilized to evaluate the SSA emission scheme and its influence on the nitrate simulation in a case study in Europe during September 10-20, 2013. Meteorological conditions near the surface, wind pattern, and thermal stratification structure were well reproduced by the model. Nonetheless, the coarse mode ($PM_{1-10}$) particle mass concentration was substantially overestimated due to the overestimation of SSA and nitrate. Compared to filter measurements at 4 EMEP stations (coastal stations: Bilthoven, Kollumerwaard and Vredepeel; inland station: Melpitz), the model overestimated SSA concentrations by a factor of 8-20. We found that this overestimation was mainly caused by overestimated SSA emissions over the North Sea during September 16-20. Over the coastal regions, SSA was injected into the continental free troposphere through an "aloft bridge" (about 500 to 1000 meters above the ground), a result of the different thermodynamic properties and planetary boundary layer (PBL) structure between continental and marine regions. The injected SSA was further transported inland and mixed downward to the surface through downdraft and PBL turbulence. This process extended the influence of SSA to a larger downwind region, for example, leading to an overestimation of SSA at Melpitz, Germany by a factor of ~20. As a result, the nitrate partitioning fraction (ratio between particulate nitrate and the summation of particulate nitrate and gas-phase nitric acid) increased by about 20% for the coarse mode nitrate due to the overestimation of SSA at Melpitz. However, no significant difference in the partitioning fraction for the fine mode nitrate was found. About 140% overestimation of the coarse mode nitrate resulted from the influence of SSA at Melpitz. On the other hand, the overestimation of SSA inhibited the nitrate particle formation in the fine mode by about 20%, because of the increased consumption of precursor by coarse mode nitrate formation.

## 1 Introduction

Atmospheric aerosols play an important role in climate change (IPCC, 2013). Further they have an adverse effect on human health (Pope et al., 2009). Aerosol particles are either emitted directly as a so-called "primary aerosol", or generated by atmospheric secondary processes ("secondary particles"). Sea-salt aerosol (SSA) is one major constituent of primary natural

aerosol particles on a global scale (Lewis, 2004), comparable with mineral dust particles in the Northern Hemisphere (IPCC, 2013; Mårtensson et al., 2010). SSA belongs to the naturally produced aerosol, and is generated mainly by bursting bubbles during whitecap formation in the open ocean (Monahan et al., 1986). Waves breaking in the surf zone, where there are more whitecaps and stronger SSA emission due to increased ocean bottom and higher intensity of wave breaking, may affect SSA concentrations at areas within 25 km distance from the coastline and can dominate the SSA concentration at the coastal region (de Leeuw et al., 2000; Monahan, 1995; Woodcock et al., 1963).

SSA exerts an influence on the aerosol burden of other aerosols, which makes the intensity of SSA emission even more important. SSA can participate in heterogeneous reactions by interacting with trace gases, leading to the formation of particulate nitrate on SSA surface and increase nitrate particle mass concentration (Seinfeld, 2006). Nitrate is one of the most important secondary inorganic aerosol and is the dominant aerosol component in western and central Europe (Schaap et al., 2011). SSA can also facilitate the formation of nitrate aerosol (Neumann et al., 2016a; Liu et al., 2015; Im, 2013; Athanasopoulou et al., 2008). However, these previous studies mainly focused on the influence of SSA on bulk nitrate mass concentration, and did not address its influence on size-segregated nitrate particles. In this study, we quantified the SSA influence on both fine mode and coarse mode nitrate particles formation respectively. and the effect could be different for the different size mode, resulting from the heterogeneous reaction on SSA surface with the formation of sodium nitrate. The timescale of this reaction is considered to be several hours (Meng and Seinfeld, 1996). Sodium nitrate is produced with a chloride displacement in the SSA (Schaap et al., 2011; Seinfeld, 2006). Importantly, thermodynamically stable sodium nitrate will not return to the gas phase as the semi-volatile ammonium nitrate does (Schaap et al., 2011). According to previous studies, sodium nitrate largely contributes to nitrates in northern and southern Europe (Pakkanen et al., 1999), whereas in western and central Europe ammonium nitrate dominates (Schaap et al., 2002; ten Brink et al., 1997). The reason is enhanced ammonia emission from husbandry and agricultural sources in central and western Europe (Backes et al., 2016b;Backes et al., 2016a).

Coarse sea salt particles have a short life-time (Grythe et al., 2014), usually depositing close to their source. SSA emitted near the shore will therefore deposit mainly in coastal regions. Its influence on nitrate particle formation is thus expected to be of less importance over Central Europe, where nitrate concentrations are high due to land-based sources (Xu et al., 2012). However, local circulations can change the vertical distributions of aerosol particles and make the long-range transport of SSA possible by lifting up aerosol from the planetary boundary layer (PBL) into the free troposphere (Chen et al., 2009; Dacre et al., 2007; Lu and Turco, 1995, 1994). The development of PBL or downdraft over the continent could drag the lifted particles downward back to the surface later on (Chen et al., 2009). These mechanisms facilitate the long-range transportation of SSA, and thereby expand their influences from coast to a broader region.

SSA contributes to the global aerosol burden multiple times more than the anthropogenic aerosol (Grythe et al., 2014). Meanwhile, in terms of global mass concentration, SSA has the largest uncertainty among all aerosols (Grythe et al., 2014). There is still high uncertainty (Grythe et al., 2014; Neumann et al., 2016a; Neumann et al., 2016b) in the parameterization scheme (Gong, 2003; Monahan et al., 1986, i.e.: GO03) of SSA emissions in WRF-Chem. Previous studies (Neumann et al., 2016a and 2016b; Nordmann et al., 2014; Archer-Nicholls et al., 2014; Zhang et al., 2013; Saide et al., 2012; Saide et al., 2013; de Leeuw et al., 2011) showed that the parameterization of Gong (2003) may overestimate the emission of SSA. Saide et al. (2012) demonstrated that GO03 overestimated SSA by a factor of 10 for sub-micron particles and a factor of 2 for super-microns in the southeast Pacific Ocean. Jaeglé et al. (2011) found that GO03 overestimated the coarse mode SSA mass concentrations by factors of 2–3 at high wind speeds over the cold waters of the South Pacific, North Pacific and North Atlantic Oceans. Other studies also indicated an overestimation of SSA emissions in varying degrees (Zhang et al., 2013; de Leeuw et al., 2011; Yang et al., 2011; Neumann et al., 2016b).

The accuracy of an SSA emission scheme is critical for the evaluation of its climate effect (Soares et al., 2016) and its influence on nitrate particle formation. The heterogeneous reaction could amplify the uncertainty of total aerosol burden due to the influence of SSA on secondary aerosol formation (e.g., nitrate, Seinfeld, 2006), and therefore, SSA has an indirect effect on the total aerosol burden. The participation of gaseous pollutants (e.g.: NOx) is needed for this indirect effect. However, NOx is not only abundant along the coast area, but also some inland regions (Fig. S1). Therefore, the long-range transport mechanism, as mentioned above, extends the impact of SSA indirect effect on nitrate particle formation to a broader region.

In this study, the long-range transport mechanism and the influence of SSA on the size resolved nitrate particle formation over the inland region were analyzed in detail in Europe. The model parameterization schemes and the observations are introduced in section 2. The background meteorological conditions are described in section 3.1. Basic physical and chemical properties obtained from model simulations are evaluated in section 3.2. The long-range transport mechanism of SSA and evaluation of SSA emission are shown in section 3.3. And In section 3.4, the influence of SSA on the size-segregated nitrate particle simulation is quantitatively analyzed.

## 2 Data & Methods

### 2.1 WRF-Chem model

The Weather Research and Forecasting/Chemistry model (WRF-Chem V3.5.1) is a fully "online" coupled regional meteorology and air quality model system. It is designed for a broad spectrum of atmospheric research, ranging from several hundred meters to thousands of kilometers in horizontal extent. In addition to the meteorology, aerosols, trace gases and interactive processes are simulated in the model (Grell et al., 2005). The gas-phase atmospheric chemistry was presented by the Carbon-Bond Mechanism version Z (CBMZ), which is coupled with the MOdel for Simulating Aerosol Interactions and Chemistry (MOSAIC, Zaveri et al., 2008).

In order to represent the properties of size-resolved aerosol particles, the fully coupled sectional aerosol module MOSAIC was chosen in this study. In MOSAIC, there are eight discrete size bins (from about 39 nm to 10 μm, see Fig. 1) of dry particles. Particles are assumed to be internally-mixed in each bin (Zaveri et al., 2008). The fine mode ($PM_1$) and coarse mode ($PM_{1-10}$) particle mass concentration can be derived from this eight size bins. The size range of the fifth bin is from 625 nm to 1250 nm. Assuming that particle mass size distribution was constant in the fifth bin, we divided the particle mass in this size bin into 60% to the fine mode ($PM_1$, diameter smaller than 1 μm) and the rest 40% to the coarse mode ($PM_{1-10}$, diameter ranging from 1 to 10 μm). In MOSAIC, sulfate, methane sulfonate, nitrate, chloride, carbonate, ammonium, sodium, calcium, elemental carbon (EC), other inorganic material and organic carbon (OC) are all treated in each bin. Both particle mass concentrations and particle number concentrations are simulated. Since the segregation of particles in the size bin is based on the dry diameter, there will be no transfer of particles between the bins due to the uptake or loss of water (Zaveri et al., 2008). However, particle growth or shrink due to chemical processes (e.g., chemical reaction, uptake/release of trace gases, etc.) and/or physical processes (e.g., coagulation, etc.) will result in the transfer of particles between the bins (Chapman et al., 2009). The formation of secondary organic aerosols is not included in the chosen MOSAIC version, but the nucleation of sulfuric acid and water vapor is considered (Zaveri et al., 2008; Fast et al., 2006). The heterogeneous reaction of nitric acid on SSA surface with the production of sodium nitrate is considered in MOSAIC. For the deliquescent aerosol particles at high RH, the ionization equilibrium and the Kelvin Effect are also taken into consideration in MOSAIC. More detailed descriptions are given in Zaveri et al. (2008).

In CBMZ, 67 prognostic species and 164 reactions are included with a lumped structure approach (Fast et al., 2006; Zaveri and Peters, 1999). Organic compounds are categorized according to their internal bond types. The rates for photolytic reactions are calculated with the Fast-J scheme (Barnard et al., 2004; Wild et al., 2000).

Dry and wet deposition of particles is treated in the WRF-Chem model (Binkowski and Shankar, 1995). The dry deposition of particles is calculated by a resistance approach, including sublayer resistance, aerodynamic resistance and surface resistance (Grell et al., 2005). The scavenging process of particles was calculated using look-up tables (Nordmann et al., 2014). It is worth mentioning that Saide et al. (2012) found the WRF-Chem model might overestimate wet deposition of particles, in the regions where drizzles re-evaporates and release the particles back into the atmosphere.

The aerosol optical depth (AOD) is online calculated in WRF-Chem model by integrating extinction coefficients over all vertical layers, and details are given in Barnard et al. (2010). In general, an internal mixture of all chemical constituents is assumed. The bulk refractive indices for each particle size bin are obtained by a mixing rule based on volume-weighted averaging. The aerosol particle optical properties, such as particle extinction and scattering cross-sections and asymmetry factor, are calculated online by a Mie code described in Ghan et al. (2001).

The simulations were performed for the HOPE-Campaign (HD(CP)² Observational Prototype Experiment, https://icdc.zmaw.de/hopm.html) held at the observatory Melpitz (12.93°E, 51.53°N, 86 m a.s.l.) from September 10 to 20, 2013. Three nested domains with 39 vertical layers were set up for the simulated case. The outer domain covers whole Europe, a part of the North Sea and North Africa with a spatial resolution of 54 km, providing the boundary conditions for the inner domains. The intermediate domain (D02, Fig. 2) was centered at Melpitz, and covers part of the North Sea, the central and southern Europe with a spatial resolution of 18 km. The innermost domain was also centered at Melpitz, and had a spatial resolution of 6 km. The spin-up time of the model run was 2 days.

Final Analysis (FNL) Operational Global Analysis (http://rda.ucar.edu/datasets/ds083.2/) and NCEP sea surface temperature (SST) datasets (http://polar.ncep.noaa.gov/sst/oper/Welcome.html), with a spatial resolution of 1 degree and a temporal resolution of 6 hours, were utilized to drive and force the model meteorological field. The chemical initial and boundary conditions were provided by the MOZART-4 global model (http://www.acom.ucar.edu/wrf-chem/ mozart.shtml) with a spatial resolution of $1.9° \times 2.5°$ and a temporal resolution of 6 hours. More details on simulation about setups and parameterizations are given in Table 1 and Chen et al. (2016).

### 2.2 Emissions

SSA is produced through the evaporation of sea sprays, which were ejected into the atmosphere from the sea surface. (Lewis, 2004). Sea spray is emitted by bubble bursting or breaking waves or torn of wave crests by winds. Strong winds exceeding 10 ms$^{-1}$ are needed for the second process (Monahan et al., 1986). The parameterization scheme (GO03) for SSA emission coupled in the WRF-Chem model follows the Gong (2003) scheme. GO03 was developed based on the semi-empirical formulation (Monahan et al., 1986) and field measurements (O'Dowd et al., 1997), including two drop-types produced by bursting bubbles (jet-drop and film-drop). The SSA flux from the ocean to the atmosphere is described as a function of 10-m wind speed and particle radius. Because the Monahan et al. (1986) scheme strongly overestimated the measurements of O'Dowd et al. (1997), Gong (2003) introduced an adjustable parameter to improve the results. He found the value 30 produced the best results. Therefore, this value was also used in WRF-Chem simulation; although Gantt et al. (2015) suggested that value 8 maybe better for the adjustable parameter in some conditions. In order to quantify the influence of SSA on the nitrate particles formation in this study, a sensitivity study was implemented with only 5% of SSA emission (R-CASE) and compared with the full (100%) SSA emission case (F-CASE).

The inventory of anthropogenic emissions ($PM_{2.5}$, $PM_{2.5-10}$, CO, NOx, $SO_2$, $NH_3$ and Non-methane volatile organic compounds), as well as temporally resolved emission factors were provided by TNO for the AQMEII project (Pouliot et al., 2012). The dataset consists of European anthropogenic emissions on $1/8^o \times 1/16^o$ lon-lat grid for the whole year 2006. The Selected Nomenclature of Air Pollution (SNAP) code was used to categorize different source types (e.g., energy transformation, industrial combustion, road transport, agriculture), with area and point emissions distinguished. More details about the anthropogenic emission inventory are given in related literatures (Pouliot et al., 2012; Wolke et al., 2012).

The anthropogenic emission of EC and OC were taken from the Pan-European Carbonaceous aerosol inventory (Visschedijk and Denier van der Gon, 2008), which was developed in the framework of the European Integrated project on Aerosol Cloud Climate and Air Quality interactions (EUCAARI, Kulmala et al., 2011). This inventory of EC/OC is also provided by TNO and has the same spatial resolution and (SNAP) code categorization as the AQMEII one. However, the point sources of EC in Germany were excluded due to their large uncertainties (Chen et al., 2016).

The Fire INventory from NCAR (FINN, Wiedinmyer et al., 2011), with the spatial resolution of 1 km and the temporal resolution of 1 hour, was also included. Biogenic emissions were presented by the Model of Emissions of Gases and Aerosols from Nature (MEGAN, Guenther et al., 2006). Dust emissions were not considered, due to the large uncertainty of the dust emission scheme in WRF-Chem (Saide et al., 2012). According to quartz-filter-based measurements (quartz-filter Type MK360, Munktell/Ahlstorn, Schweden) with high volume sampler DIGITEL DHA-80 (Walter RiemerMesstechnik, Germany), dust contributed less than 3% to the total particle mass concentration in Melpitz during the simulated period.

## 2.3 Observations

Measurements of the HOPE-Campaign and the European Monitoring and Evaluation Programme (EMEP, http://www.emep.int) were adopted to validate the model results. In addition, the modelled atmospheric vertical thermodynamic structures were validated by the radiosonde measurements all over Europe (http://www.weather.uwyo.edu/upperair/sounding.html). The Melpitz Obervatory is representative of the regional background of Central Europe (Spindler et al., 2012; Spindler et al., 2010; Poulain et al., 2011; Brüggemann and Spindler, 1999; Birmili et al., 2001). The instruments that measure aerosol physical properties were operated under dry condition, as recommended by WMO-GAW (World Meteorological Organization – Global Atmospheric Watch) and ACTRIS (Aerosols, Clouds, and Trace gases Research InfrastraStructure Network; Wiedensohler et al., 2012).. A Dual Mobility Particle Size Spectrometer (TROPOS-type dual-SMPS, Birmili et al., 1999) combined with an Aerodynamic Particle Size Spectrometer (TSI APS Model 3321) were employed to measure the particle number size distribution (PNSD) ranging from 5 nm to 10 μm in diameter. Particle number size distribution are made publicly available within the framework of the German Ultrafine Aerosol Network (GUAN; Birmili et al., 2016). Detailed information is given by Chen et al. (2016) and Heintzenberg et al. (1998). A Monitor for AeRosols and Gases in ambient Air (MARGA) system (Schaap et al., 2011; Thomas et al., 2009; ten Brink et al., 2007) continuously monitoring aerosol and gases in ambient air (Metrohm Applikon, Schiedam, The Netherlands), was operated downstream of a $PM_{10}$ inlet. This instrument provided 1-hour data of secondary inorganic aerosols ($NH_4^+$, $NO_3^-$, $SO_4^{2-}$, $Cl^-$, $Ca^{2+}$, $Mg^{2+}$ and $K^+$) and gaseous counterparts ($NH_3$, $HNO_3$, $HNO_2$, $SO_2$, HCl). The high volume samplers DIGITEL DHA-80 (Walter RiemerMesstechnik, Germany), with sampling flow of about 30 $m^3h^{-1}$, were used to collect 24-hour $PM_{10}$ and $PM_1$ filter samples simultaneously (Spindler et al., 2013). Information on the coarse mode ($PM_{1-10}$) aerosol chemical compositions, such as nitrate and sodium etc., were obtained from the difference between the results of $PM_{10}$ and $PM_1$. Additionally, 24-hour filter sampler measurements with $PM_{10}$ inlets (EMEP, 2014) at 3 coastal EMEP station near the SSA transportation pathway (Bilthoven, Vredepeel, and Kollumerwaad, see Fig. 2), which were collected every second day, were obtained from EBAS (http://ebas.nilu.no/).

The AERONET (AErosol RObotic NETwork, http://aeronet.gsfc.nasa.gov/) dataset over Europe was utilized to validate the AOD simulation. The AERONET AOD was derived from Sun photometer measurements of the direct (collimated) solar radiation. The level 2.0 AOD data, with pre and post field calibrated, automatically cloud cleared and manually inspected, were used in this study. The AOD at 500 nm wave-length and the Angstrom index are directly available in AERONET dataset, and AOD at 550 nm wave-length was derived. More detailed information is given in http://aeronet.gsfc.nasa.gov/.

## 2.4 Nitrate partitioning fraction

The participation of SSA changes the partitioning processes of nitrate. In order to quantify this effect, the nitrate partitioning fraction (PF_nitrate) was analyzed for coarse mode ($PM_{1-10}$) and fine mode ($PM_1$) particles. The definition of PF_nitrate for coarse/fine mode is shown in Eq.1.

$$PF\_nitrate_{coarse/fine} = \frac{[NO_3^-]_{coarse/fine}}{[NO_3^-]_{coarse/fine} + [HNO_3]},$$ (1)

where $[NO_3^-]_{coarse/fine}$ is the coarse/fine mode particulate nitrate mass concentration, $[HNO_3]$ is the nitric acid mass concentration.

## 3. Results & Discussion

### 3.1 Meteorology

During the HOPE-Campaign, continental air masses prevailed over Northern Germany before September 15 (Fig. S2). On September 16, a low pressure trough began to dominate the North Sea, while a high pressure ridge dominated over the continent. A frontal system was clearly formed along the coast of the North Sea as could be seen in Fig. S3, accompanied by sharp wind shear and also high convective available potential energy (CAPE). Evidently, strong vertical mixing occurred in the coastal region, which lifted SSA upward.

Meteorology simulated by WRF with hourly output frequency was evaluated with the near-ground measurements at Melpitz and radio-sounding measurements all over Europe. Both surface meteorology and the vertical structure of meteorological parameters were well captured by the model. Simulated surface temperature, relative humidity, wind speed and wind direction were in good agreement with Melpitz near-ground hourly measurements (Fig. S4), with correlation coefficients (R) of 0.94, 0.85, 0.86 and 0.86 respectively, and with mean bias (MB) 0.38 $^o$C, 9.1%, -0.18 m s$^{-1}$ and 10.62$^o$ respectively. The vertical gradient of potential temperature is an important indicator to measure the stability of the atmosphere. Fig. 3 shows the maps of R values and normalized mean bias (NMB, Balzarini et al. 2015) for potential temperature and wind speed between the simulated and measured vertical profile in planetary boundary layer (PBL, 0-3 km). High values of R and low absolute values of NMB for potential temperature vertical profile were found at all stations, especially near Melpitz, Germany (R > 0.85 and the absolute NMB < 2% ), and the coastal region of North Sea (R > 0.95 and the absolute NMB < 2%). The vertical pattern of wind speed was also captured by the model, especially over the North Sea and coastal regions (see Fig. 3b and Fig. 3d). Generally, the R values were higher than 0.6. Especially, the R values were higher than 0.9 over the SSA emission source area (the North Sea) and coastal regions; and the absolute NMB values were lower than 5% over Melpitz region and the coastal region of North Sea. Statistical results of comparisons between the simulation and Melpitz radio-sounding measurements are shown in Table S1. Several examples of vertical profiles are given in Chen et al. (2016). Simulated meteorological vertical structures were in good agreement with measurements in Melpitz. Corresponding, the averaged R values of vertical profiles were 0.99, 0.96, 0.84 and 0.92 for potential temperature, wind speed, wind direction

and water vapor mixing ratio, respectively. Therefore, the WRF simulations well reproduced the meteorological conditions for both ground level and vertical structure.

## 3.2 Aerosol physical and chemical properties

The modelled particle number size distribution (PNSD) and particle mass size distribution (PMSD) at Melpitz were compared with measurements (Fig. 4). Although the simulation of PNSD/PMSD for size bins 01-04 not exactly matched with the measurements, the agreement is in the reasonable range with a factor of ~2 (Fig. 4). But the model significantly overestimated the concentration for the size bins 05-08 (625-10,000 nm) in the F-CASE.. Since the meteorology was well reproduced by the model, it can be assumed that the air movement was also reasonably simulated. Therefore, unrealistic high sources of coarse particles might be the cause for the overestimation, which would be discussed in following.

The chemical compositions in $PM_{1-10}$ of the DIGITEL measurements and simulation results at Melpitz are displayed in Fig. 5. The simulated particles composition for each size bin is shown in Fig. 1. Sea salt ($Na^+$) and nitrate mass concentrations in $PM_{1-10}$ were overestimated by factors of ~18 and ~2, respectively (see also Table 2 and 3). Particularly, the overestimation factors could reach up to ~20 and 3 respectively for $Na^+$ and nitrate mass concentration in the period influenced by marine air masses (starting from September 16, see Fig. 5b). These results indicated that the overestimations in bins 05-08 were mainly caused by SSA and nitrate.

The column accumulated aerosol property AOD was evaluated for the R-CASE (Fig. 6a) and the F-CASE (Fig. 6b) respectively. Since AERONET AOD data only can be measured during daytime under clear-sky condition, the corresponding simulation results were analyzed.

As shown in Fig. 6a, except the Modena station in Italy, in the R-CASE the spatial distribution of AOD over Europe can be captured by the model in general with correlation coefficient (R) value 0.64: the highest AOD value (about 0.15-0.3) over inland region, relatively high value (about 0.07-0.15) over coastal region of North Sea and Baltic Sea, relatively low value (about 0.03-0.1) over the Southern Europe, and extremely low value (about 0.01-0.03) over Alpine Mountain region. And the R-CASE result showed a moderate AOD range (about 0.05-0.12) over the North Sea, which was comparable with the Southern European region (e.g.: Italy and Greece). However, R-CASE result overestimated the AOD in general with a geometric mean ratio (GMR) value 1.8, which could be resulting from the overestimation of nitrate particles. The nitrate particle mass concentrations in $PM_{10}$ were overestimated by a factor of ~5 in the R-CASE at Melpitz (see Table 2). Although some shortcomings can be identified, the overall performance of AOD simulation is satisfactory and in line with previous studies (e.g. Banzhaf et al., 2013; Li et al., 2013).

The spatial distribution of AOD was less matched between modelled result and AERONET AOD measurements in the F-CASE (Fig. 6b). The R value reduced to 0.56, with much higher overestimation of AOD and GMR increased to 2.3. The modelled AOD over the North Sea is significantly increased to an unreasonable value, which was comparable with the central Europe. This is because GO03 overestimated SSA emission over the North Sea. The detailed evaluation of SSA mass concentration is given in the next section 3.3.

## 3.3 Sea salt emission and the transport

For the sea salt event studied here, the abrupt SSA emission event happened over the North Sea. And SSA mass concentration was overall overestimated in the coastal and continental regions during September 16-20, as shown in Fig. 7. Here we used sodium ($Na^+$) as an indicator of SSA (Neumann et al., 2016a; Gustafsson and Franzén, 2000), since chloride ($Cl^-$) could be depleted due to nitrate partitioning (Schaap et al., 2011; Seinfeld, 2006). Marine air masses first arrived at the three coastal stations (Bilthoven, Vredepeel, and Kollumerwaad, see Fig. 2), and then went further inland along with the

low-pressure trough system. The Na$^+$ concentration peaked on September 16 for the coastal stations, and about 1 day later for Melpitz (Fig. 7).

As shown in Fig. 7 the day-to-day temporal pattern of Na$^+$ concentrations can be captured by the model, with the correlation coefficient (R) of 0.95, 0.81, 0.92 and 0.89 for Melpitz, Bilthoven, Kollumerwaard and Vredepeel respectively. The transport mechanism was well captured by the model in general. However, SSA mass concentrations were overestimated by about 20 times in Melpitz and around 9, 13 and 8 times respectively for the coastal stations of Bilthoven, Kollumerwaard and Vredepeel. This implies that GO03 might overestimate SSA emission by a factor of 8-20. The overestimation is consistent with previous modeling studies using WRF-Chem: in the Southeast Pacific ocean (Saide et al., 2012), in the coast region of California USA (Saide et al., 2013), over Europe (Nordmann et al., 2014;Zhang et al., 2013;Tsyro et al., 2011;Manders et al., 2010) and over the cold waters of the South Pacific, North Pacific and North Atlantic Oceans (Jaeglé et al., 2011). Similarly, (Neumann et al., 2016b) found overestimations during winter and attributed the reason to the missing of SST influence in GO03. GO03 describes SSA emission as a function of wind speed at 10 meter above the ground. The uncertainties of this scheme may be attributabled to the lack of parameters, such as sea surface temperature (Neumann el al., 2016b; Soares et al., 2016; Grythe et al., 2014; Jaeglé et al., 2011; Sofiev et al. 2011), salinity (Soares et al., 2016; Neumann et al., 2016a; Sofiev et al. 2011) and wave data (Ovadnevaite et al., 2014; Jaeglé et al., 2011). The missing of proper droplet generation processes from GO03 might be another source of the uncertainties (Neumann et al., 2016a).

After been emitted over the North Sea, the SSA might experience chemical degradation (such as Cl$^-$ depletion) and dry/wet deposition when transported over continental area. Generally, SSA is mostly in coarse mode with lifetime shorter than 2 days in the continental boundary layer, and reaching about 1 week in free troposphere (Croft et al., 2014; Petzold and Kärcher, 2012; Jaenicke, 1980). According to the simulation results, the component of the 10-m wind vector that is directed from the coast to Melpitz shows a wind speed in the range of 2-3 m s$^{-1}$(Fig. 2). It would therefore take about 1.5-2 days for SSA to be transported to Melpitz (~400 km distant to the coast). The result (Fig. S5) from the Deposition-Lifetime Concept Model (Chen et al., 2016; Croft et al., 2014) indicates that on average only 10-35% of the emitted SSA could be transported to Melpitz through the surface pathway. Whereas, according to the observed SSA peaks (Fig. 7), about 30-40% of the initial SSA mass at coastal stations was actually transported to the inland station of Melpitz. The observed transport efficiency was about 1.6 times of the expected value. So, the transport mechanism of the SSA to the inland (Melpitz) will be discussed in following with the aid of a model simulation, despite the overestimation in the WRF-Chem SSA emission scheme. As demonstrated in Fig. 8, during nighttime, the warmer sea surface resulted in a higher planetary boundary layer (PBL, black dash lines in Fig. 8) above the sea than over the continent (Dacre et al., 2007; Lu and Turco, 1995, 1994). Due to the difference of thermodynamic structure between continental and marine area, there is often a sharp gradient of PBL height over the coastal region (Fig. 8), which could serve as an "aloft bridge" connecting the marine PBL and continental free troposphere (Dacre et al., 2007; Lu and Turco, 1995, 1994). In the early morning of September 16, SSA was emitted into the surface layer of the North Sea and lifted upward by convective mixing and turbulence (Fig. 8a). According to the simulation result (Fig. 8), about 70% of SSA penetrated the marine PBL and was injected into continent free troposphere through the "aloft bridge". In the free troposphere, SSA has a much longer life time and faster transportation than in the PBL. Therefore, about 70-85% of SSA (Fig. S5) could be carried further towards the inland in free troposphere, and arrived at the Melpitz region in the early morning of September 17 (Fig. 8b). Then the downward draft resulted from high-pressure ridge and the turbulent mixing after sunrise (Fig. 8b and 8c), brought the lofted SSA back into the surface layer. The Na$^+$ mass concentration at Melpitz surface increased from ~7 μg/m$^3$ (Fig. 8b) to ~15 μg/m$^3$ (Fig. 8c). About 35% of the lofted SSA contributed to the increase of the Na$^+$ surface concentration. This result is in agreement with the previous study (Chen et al., 2009), which reported ~30% of elevated pollutants contributed to the increase of surface pollutants concentration in Beijing, due to the turbulent mixing after sunrise.

### 3.4 Influence of sea salt on nitrate simulation

As discussed above, the over-production of SSA by GO03 will lead to an 8-20 times overestimation of the primary sea salt mass concentration. However, its influence is not only on the aerosol burden of SSA itself, but also on promoting the formation of secondary inorganic particle mass, such as nitrate (Neumann et al., 2016a; Seinfeld, 2006). The gas phase nitric acid (HNO3) can be produced with the oxidization of NOx. HNO3 undergoes a partitioning process between gas phase and liquid particle phase via condensation. The condensed HNO3 deprotonates to $NO_3^-$. In MOSAIC aerosol scheme (Zaveri et al., 2008), one partitioning process between gas phase and solid phase is the equilibrium reaction with ammonia (NH3) and the formation of ammonium nitrate. The other one is the irreversible process with SSA (NaCl) and the formation of sodium nitrate with depletion of chloride. Ammonium nitrate is semi-volatile and can turn back to the gas phase precursors, while sodium nitrate is thermodynamically stable (Schaap et al., 2011). The presences of SSA might facilitate the condensation process of nitrate.

Overall, nitrate in the size range of $PM_{10}$ was overestimated by the model, with an average factor of ~5 (Table 2). Comparisons of nitrate and its precursors between the simulation results and the MARGA measurements at Melpitz are shown in Fig. 9. Assuming that the chemical mechanism is correctly described in the model, the same concentration of gaseous precursors (NOx and ammonia) should produce the same concentration of nitrate in the model and observation. The location of the data dots (Fig. 9a) may be shifted due to the uncertainty of precursors emissions, but the nitrate mass concentration is always expected to be consistent with the observed concentration in Fig. 9b when they have the same mass concentration of precursors. However, even with the same mass concentrations of precursors, the simulated nitrate mass concentrations (Fig. 9a) were still significantly higher than the observed ones (Fig. 9b). This indicated that in addition to an overestimation caused by overestimated $NH_3$ (see also Table 2), improper chemical pathway in the model also contributed to the nitrate overestimation.

In order to quantify the influence of NaCl on the nitrate particles formation, the comparisons between results of the F-CASE and the R-CASE are shown in Table 2 and Table 3. The simulated NOx concentrations were in good agreements with measurements, but total ammonia was overestimated by a factor of ~2 in both cases (Table 2), which may stem from the uncertainty of ammonia emission inventory. The prediction of SSA ($Na^+$) was significantly improved after reducing SSA emission, with a factor of 1.09 and R value of 0.94 for size range of 1-10 μm at Melpitz (Table 3). Also, results at coastal stations were not overestimated in the R-CASE. However, NOx and total ammonia concentration results of the R-CASE did not show significant changes (Table 2). The factor for nitrate between model and measurement at Melpitz dramatically decreased from 2.1 to 0.73 in coarse mode ($PM_{1-10}$, see Table 2), thus changing from overestimation to underestimation. Therefore, the difference of nitrate between the F-CASE and the R-CASE should mainly arise from the influence of the SSA concentrations.

The comparisons of the frequency distribution of PF_nitrate between the F-CASE and the R-CASE are shown in Fig. 10, respectively for fine mode ($PM_1$) and coarse mode ($PM_{1-10}$) particles during the marine period ($[Na^+]>1.8$ μg/m$^3$ in the F-CASE, 105 time steps in total). Since the MARGA measurements were only available for the size range of $PM_{10}$, PF_nitrate derived from MARGA observations should not be directly compared with the simulated one. Additionally, we need to keep in mind that high uncertainties exist in the $HNO_3$ measurements due to its sticky property (Rumsey et al., 2014; Neuman et al., 1999), which brings further difficulty into the comparison between measurements and simulation. In this study, the R-CASE had a much more reasonable SSA prediction (within a factor of ~1 at Melpitz) than the F-CASE; therefore, the simulated values of PF_nitrate from the R-CASE were used as the basic simulation. In September 17 at Melpitz, about 88% SSA mass was concentrated in the coarse mode particles in both simulation and filter measurement results. Since coarse particles have a higher dry deposition velocity than fine particle one, we can expect that the SSA emissions consisted by

more than 88% of coarse particles. The PF_nitrate results for coarse mode particles should be more representative for the influence of SSA on particulate nitrate formation, also more sensitive to the change of the SSA emission. As shown in Fig. 10a and Fig. 10b, the median value of coarse mode PF_nitrate in the R-CASE was about 0.75, with the distribution broadly spread in the range of ~0.2 to 1; whereas in the F-CASE the median value increased to 0.96 with a much narrowed distribution. In this study, the ammonium mass concentration was quite similar in both R-CASE and F-CASE; SSA was highly overestimated by the model in the F-CASE and the overestimated amount was transported to the surface layer at Melpitz; and the coarse mode nitrate partitioning fraction increased from 0.75 (R-CASE) to 0.96 (F-CASE). These indicated that the participation of SSA in the nitrate partitioning process facilitated the coarse mode nitrate particle formation, which accumulated as the thermodynamic stable sodium nitrate. About 140% overestimation of the coarse mode nitrate was resulted from this reason (Table 2).

While for the fine mode nitrate, the PF_nitrate of fine mode was insensitive to SSA emission (Fig. 10). Although the fine mode PF_nitrate revealed no significant difference between the R-CASE and the F-CASE simulations (Fig. 10c and Fig. 10d), the fine mode nitrate mass concentration was reduced by ~20% in the F-CASE due to the consumption of precursor by the coarse mode nitrate formation. Therefore nitrate particle mass moved from fine mode to coarse mode, the total nitrate mass concentrations in size range of $PM_{10}$ were similar between the R-CASE and the F-CASE (Table 2).

In order to see the influence of SSA on nitrate PMSD in a clearer way, the simulated PMSD during marine period at Melpitz was shown in Fig. 1. It clearly shows that the nitrate PMSD decreased in the smaller size bins (bins 01-04) but increased in the larger size bins (bins 05-08). In the F-CASE (Fig. 1b) when the overestimated SSA participated in nitrate particle formation, nitrate particle moved from fine mode to coarse mode compared with the R-CASE (see also Fig. 4).

In general and as illustrated in Fig. 11, the overestimation of SSA emission scheme has a significant influence on the particulate nitrate simulation in both the coarse mode (directly) and the fine mode (indirectly). In this case study, the overestimation of SSA in the F-CASE made the coarse mode nitrate partitioning fraction increased from 0.75 to 0.96. The increase of consumption of precursor in the coarse mode nitrate particle formation might slow down the formation of nitrate particle in the fine mode. The particle mass size distribution (PMSD) was thus altered due to these influences.

**4 Conclusions**

In order to investigate atmospheric sea salt aerosol (SSA) emission and transport over Central Europe and its influence on particulate nitrate prediction, the WRF-Chem model was used to simulate the aerosol physical and chemical aerosol properties during the HOPE-Campaign, September 10-20, 2013, at Melpitz. The simulated meteorological variables, vertical thermal stratification, and near-ground level particle number size distribution were validated by observations. The ground meteorology and vertical thermal stratification were well captured by the model. Coarse mode particle were, however, significantly overestimated both in number and mass, due to an overestimate in SSA emissions caused by the current SSA emission scheme.

SSA mass concentrations were evaluated at 4 ground-based EMEP stations, including 1 continental inland station (Melpitz) and 3 coastal stations (Bilthoven, Kollumerwaard and Vredepeel). The day-to-day variations of SSA mass concentrations were well captured by the model, despite an overestimation of SSA concentrations by a factor of 8-20 due to the shortcoming of the WRF-Chem SSA emission scheme. In addition to the wind speed at 10 meter above the ground, more parameters, such as sea surface temperature, salinity and wave data, might be needed for consideration in the SSA emission scheme to reduce its overestimation. With reduction of SSA emission to 10%, the simulated AOD results showed a better

agreement with the AERONET AOD measurement over Europe. The correlation coefficient (R) between AOD simulation and measurement increased from 0.56 to 0.64, and the geometric mean ratio (GMR) decreased from 2.3 to 1.8.

Transport of SSA from the North Sea into Central Europe was analyzed in detail. Due to different nighttime PBL structure over the continental and marine region, an "aloft bridge" between the marine PBL and continental free troposphere was formed over the coastal area. The overestimated SSA released from the North Sea was mixed up and injected into the continental free troposphere, and participated in the long-range transport, because the lifetime of aerosols can be about 5 times longer in free troposphere than in the PBL. This injected SSA was transported over Melpitz and later on mixed downward to the surface by the downdraft and the turbulence of fully developed PBL on September 17, 2013. The overestimation of SSA emission combined with the "aloft bridge" transport process together led the SSA overestimated by a factor of 20 at Melpitz.

The overestimation in SSA emissions did not only influence the release of primary SSA itself, but also led to significant uncertainties in the particulate nitrate prediction. As described in Fig. 11, nitrate and precursors can be locked in the particulate phase as thermodynamically stable sodium nitrate, which is produced from the heterogeneous reaction on SSA surface. Since most of the SSA was emitted as coarse mode particles, nitrate partitioning fraction of coarse mode was overestimated by ~20% when the SSA mass concentration was highly overestimated. This contributed to an overestimation of the coarse mode particulate nitrate by around 140%. Meanwhile, the nitrate partitioning fraction of fine mode was insensitive to the SSA emission. However, the increased consumption of the gas-phase precursor, caused by the coarse mode nitrate formation with the participation of SSA, may reduce the formation of fine mode nitrate.

In this work, the atmospheric transport of SSA from the North Sea to Central Europe was demonstrated in detail. The emission of SSA was evaluated and the influence of SSA on particulate nitrate prediction was quantitatively analyzed. The overestimation of SSA emission will affect not only primary aerosol concentrations, but also the formation of secondary inorganic particle mass like nitrates. It is anticipated to change heterogeneous reactions and the conversion pathways from gas phase precursors to particulate phase. Such changes will also alter the physical and chemical aerosol properties, e.g. particle mass size distribution and hygroscopicity. A nitrate coating on a SSA surface may reduce the hygroscopicity of coarse mode particles, and the re-distribution of nitrate from fine mode to coarse mode may increase its deposition rate. Furthermore, the direct and indirect radiative forcing evaluation will also be influenced, since the optical properties (e.g.: single scattering albedo) are strongly related to the size of particles. All these influences are crucial for climate change evaluation. Due to the "aloft bridge" transport mechanism, as described in this paper, the influences of SSA are not only confined to the coastal region, but are extended to a broader region reaching as far as 400 km from coast.

**Acknowledgements:** Continuous aerosol measurements at Melpitz were supported by the German Federal Environment Ministry (BMU) grant F&E 371143232 (German title: "Trendanalysen gesundheitsgefährdender Fein- und Ultrafeinstaubfraktionen unter Nutzung der im German Ultrafine Aerosol Network (GUAN) ermittelten Immissionsdaten durch Fortführung und Interpretation der Messreihen"), 2012-2014. Particle number size distributions at Melpitz can be obtained online through the German Ultrafine Aerosol Network (https://doi.org/10.5072/guan). The HOPE campaign was funded by the German Research Ministry under the project number 01LK1212 C. We would also like to thank Albert Ansmann (TROPOS, Germany) for useful suggestions on AERONET AOD data.

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

**Table 1**. Configurations of WRF-Chem

| Physics | WRF options |
|---|---|
| Micro physics | (Lin, 1983) scheme |
| Boundary layer | YSU (Hong, 2006) |
| Surface | Rapid Update Cycle (RUC) land surface model |
| Shortwave radiation | Goddard shortwave (Chou, 1998) |
| Longwave radiation | New Goddard scheme |
| Cumulus | Grell 3D |
| Urban | 3-category UCM |
| **Chemistry and Aerosol** | **Chem options** |
| Aerosol module | MOSAIC with 8 bins |
| Gas-phase mechanism | CBMZ |
| Photolytic rate | Fast-J photolysis scheme |
| Sea salt emission | Gong (2003) scheme |

**Table 2.** Comparison of WRF-Chem results with Melpitz near-ground filter measurements. The factors (simulation/measurement) and correlation coefficient (R) are shown for the F-CASE and the R-CASE respectively.

|  | F-CASE | R-CASE | F-CASE | R-CASE |
|---|---|---|---|---|
|  | Factor | Factor | R | R |
| NOx | 0.98 | 0.98 | 0.73 | 0.73 |
| Total Ammonia | 2.12 | 2.21 | 0.59 | 0.60 |
| Nitrate in $PM_{10}$ | 5.09 | 4.97 | 0.76 | 0.72 |
| Nitrate in $PM_{1-10}$ | 2.10 | 0.73 | 0.31 | 0.33 |

**Table 3.** Comparison of WRF-Chem modeled $Na^+$ mass concentration results with 4 EMEP stations measurements. The factors (simulation/measurement) and correlation coefficient (R) are shown for the F-CASE and the R-CASE respectively.

| | F-CASE | R-CASE | F-CASE | R-CASE |
| --- | --- | --- | --- | --- |
| | Factor | Factor | R | R |
| Melpitz ($PM_{10}$) | 20.10 | 1.25 | 0.95 | 0.94 |
| Melpitz ($PM_{1-10}$) | 18.10 | 1.09 | 0.94 | 0.94 |
| Bilthoven ($PM_{10}$) | 8.77 | 0.54 | 0.81 | 0.81 |
| Kollumerwaard ($PM_{10}$) | 12.85 | 0.74 | 0.92 | 0.92 |
| Vredepeel ($PM_{10}$) | 8.36 | 0.53 | 0.89 | 0.89 |

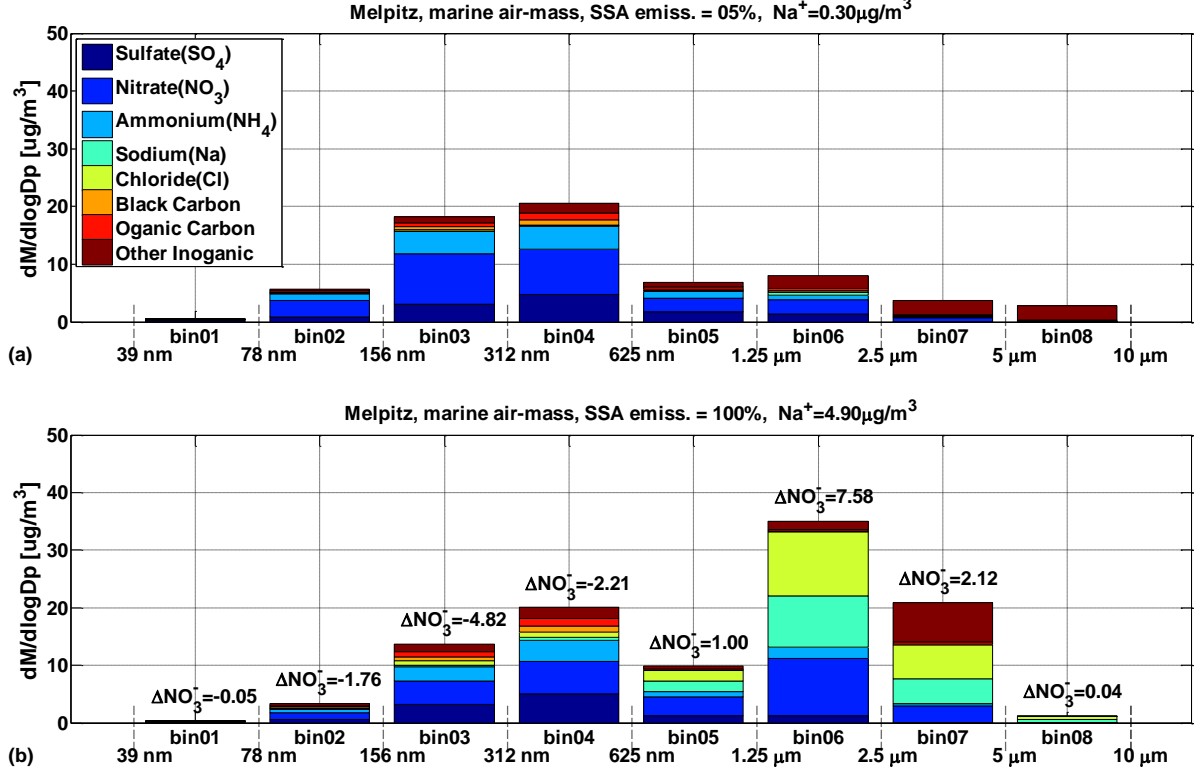

**Figure 1.** WRF-Chem simulation results of particle mass size distribution (PMSD) for each chemical compounds, averaged during marine period at Melpitz. (a) result of the R-CASE; (b) result of the F-CASE. The difference of nitrate PMSD between the R-CASE and the F-CASE for each bin is marked.

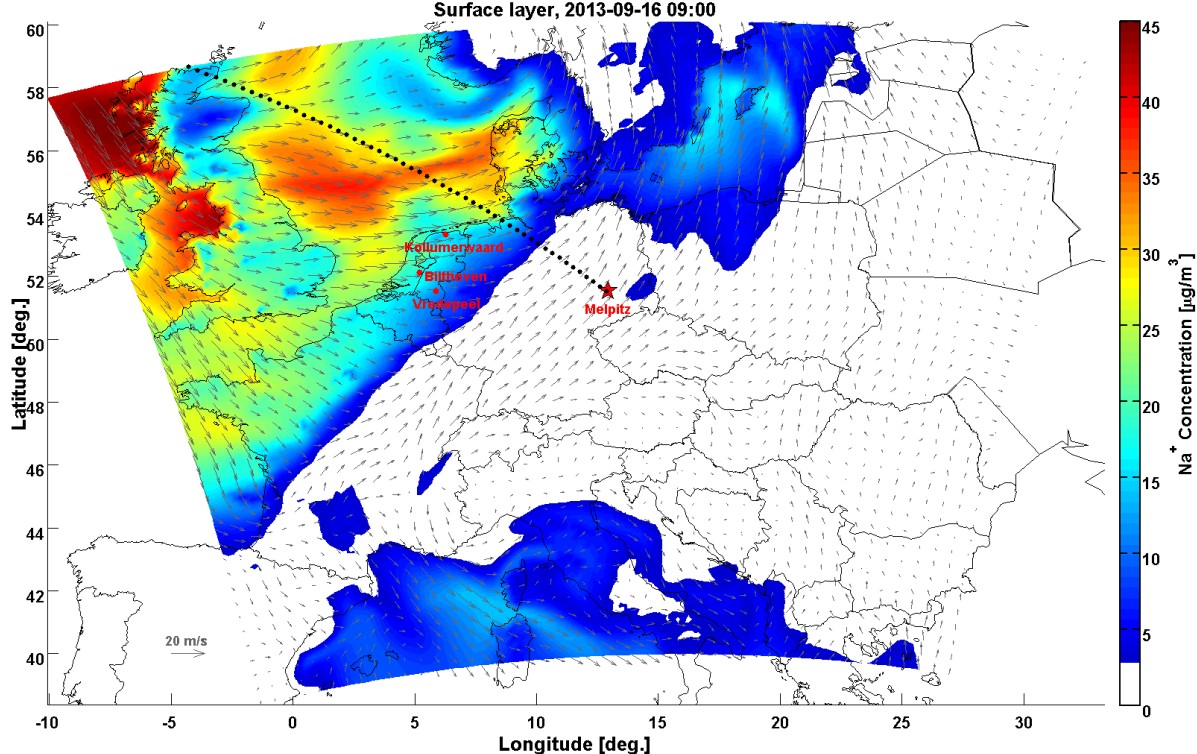

**Figure 2.** The horizontal distribution of surface Na$^+$ mass concentration in domain 02 (intermediate domain) at 2013-09-16, 09:00 LT. The grey arrows indicate the wind field. The locations of 4 EMEP stations (Melpitz, Bilthoven, Kollumerwaard and Vredepeel) are marked. The vertical cross section of dash black line is shown in Figure 8.

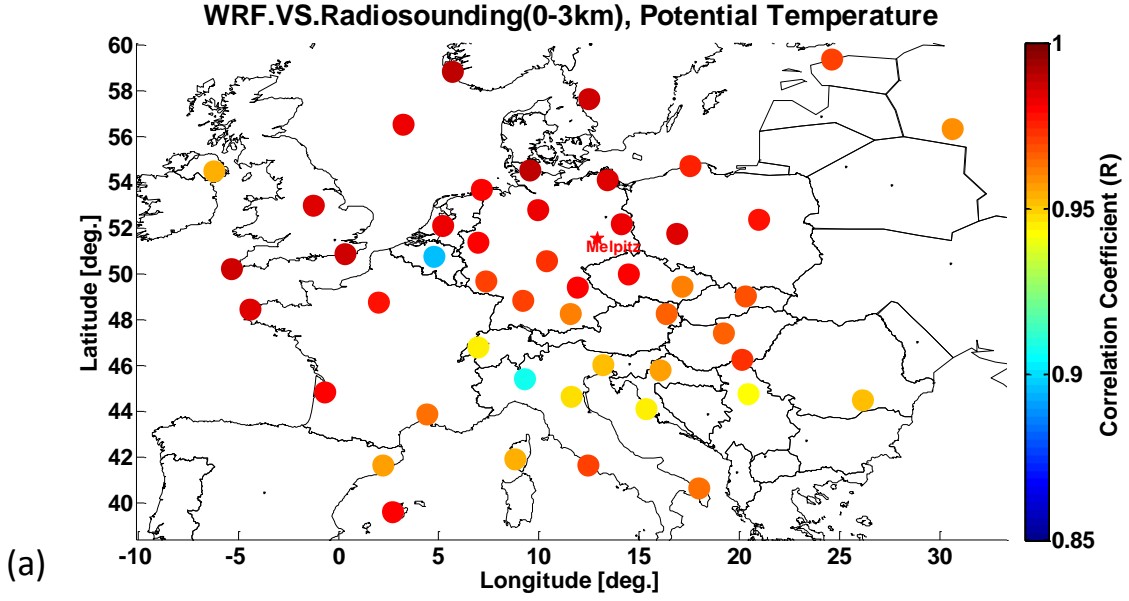

(a)

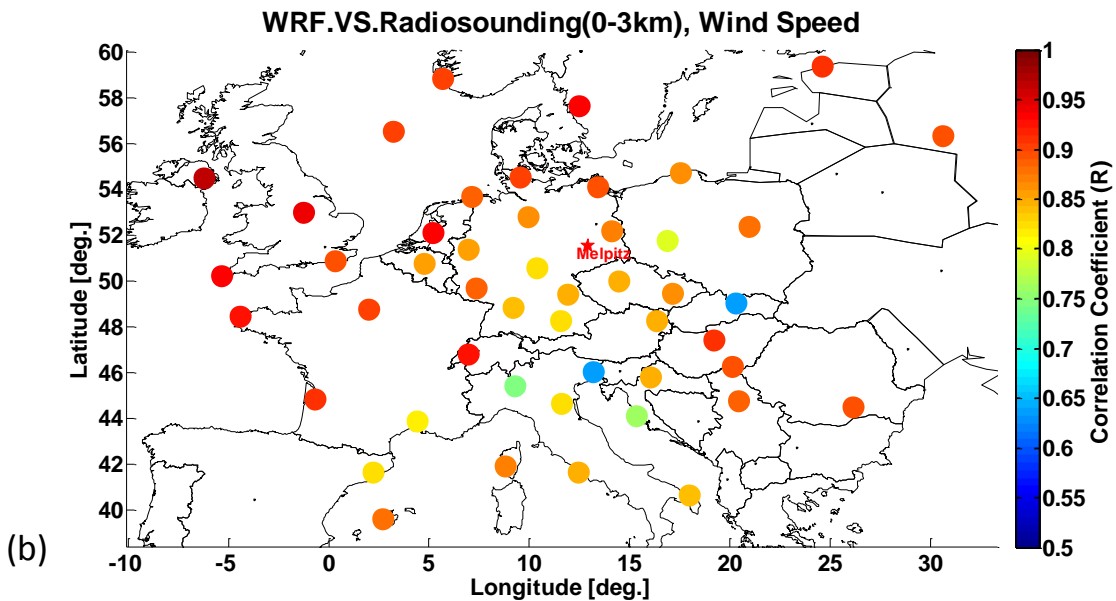

(b)

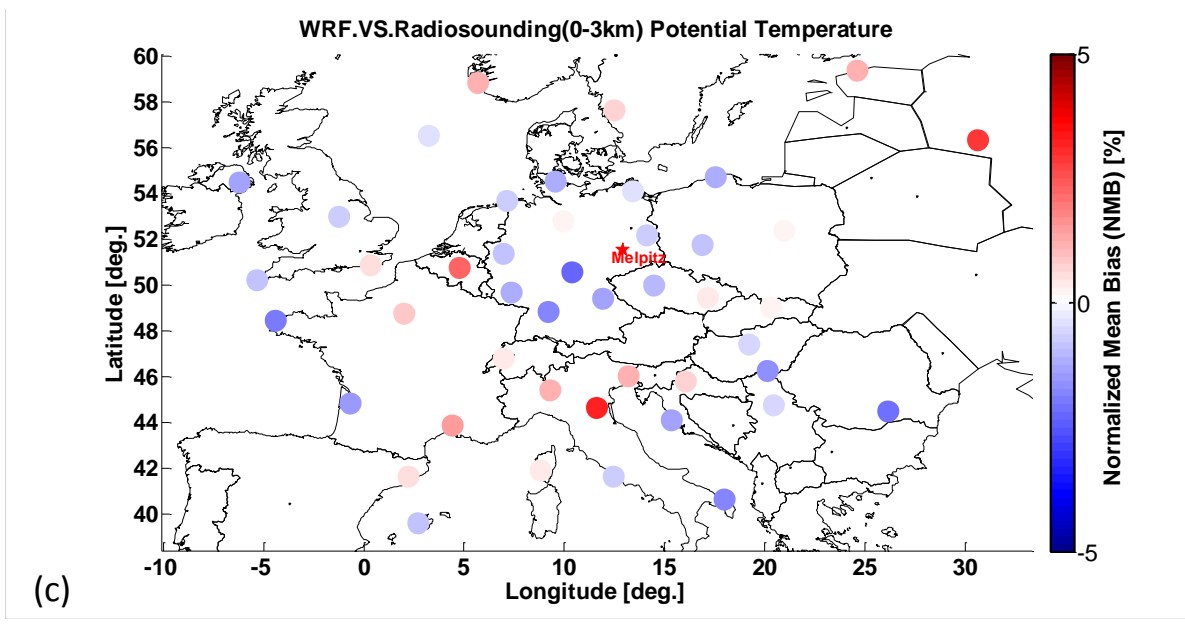

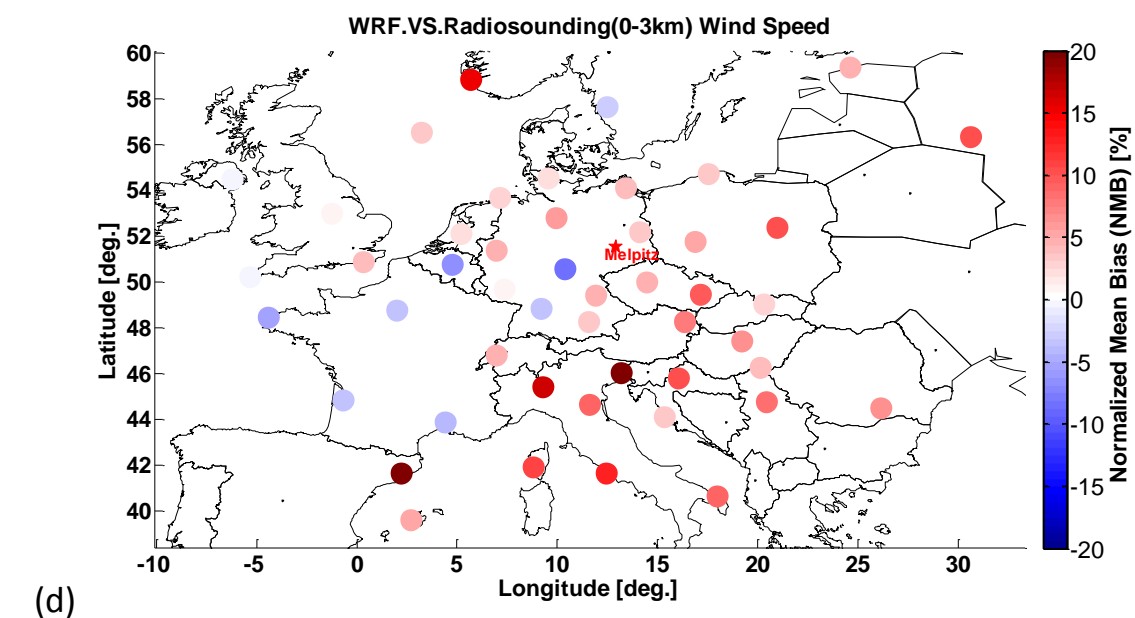

**Figure 3.** Comparison between WRF-Chem model and radio-sounding measurements (0-3 km). Melpitz is marked as red star. (a) R map of potential temperature; (b) R map of wind speed; (c) NMB map of potential temperature; (d) NMB map of wind speed. Note that the panels have the different color-bar scale.

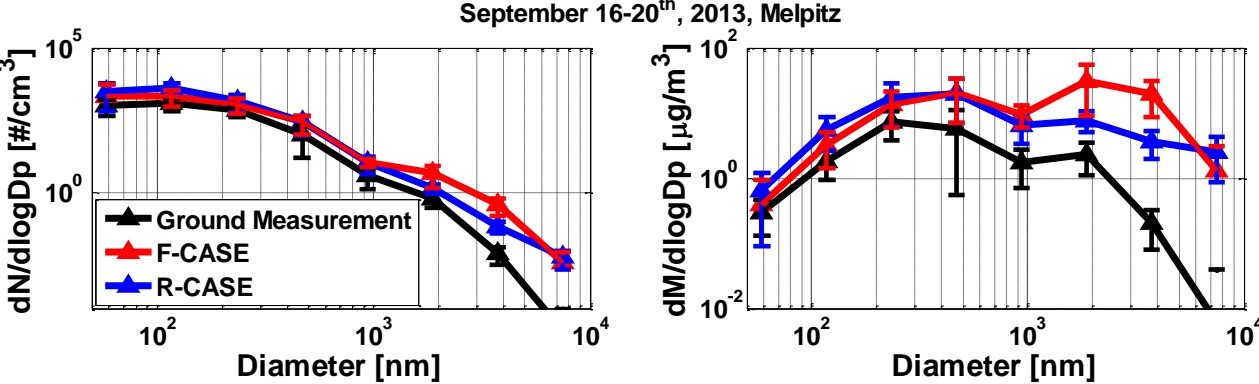

**Figure 4.** Comparison of Particle Number Size Distribution (PNSD, left) and Particle Mass Size Distribution (PMSD, right) between simulation results (F-CASE and R-CASE) and Melpitz measurements. The results are averaged during September 16-20, 2013; the error bars indicate the upper and lower limits.

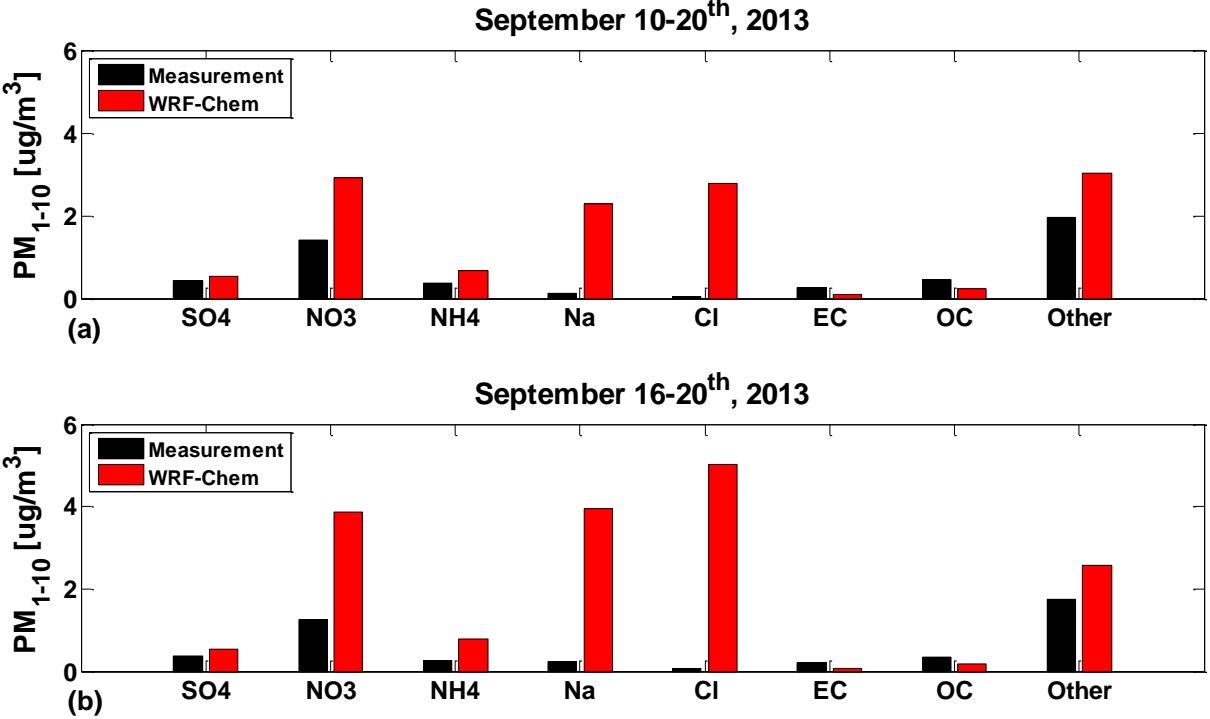

**Figure 5.** Comparison of coarse mode aerosol ($PM_{1-10}$) chemistry compounds between the F-CASE results and Melpitz measurements. (a) averaged during the HOPE-Campaign period of September 10-20, 2013; (b) averaged during the marine air mass period of September 16-20, 2013.

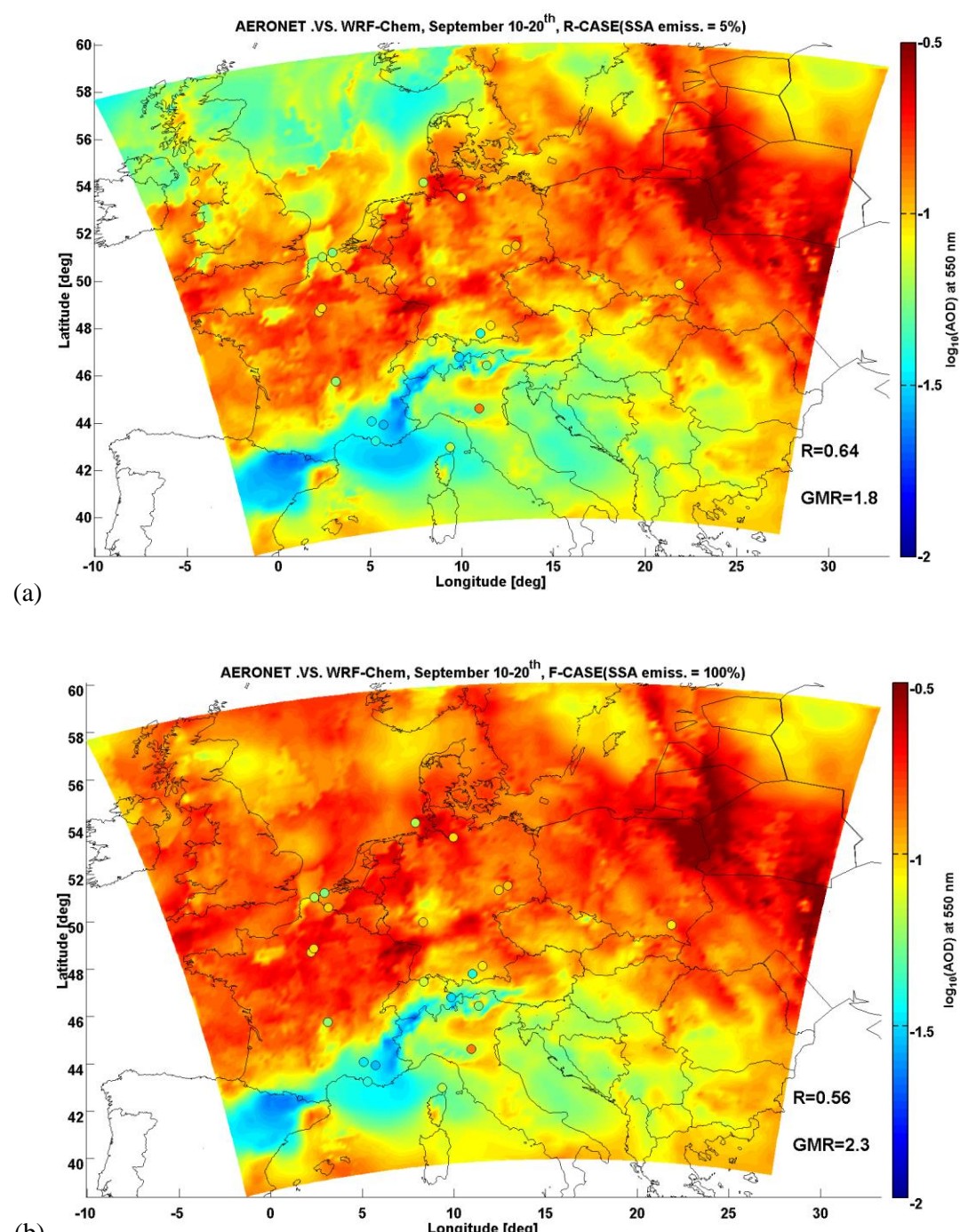

**Figure 6.** Comparisons of AOD at 550 nm wave-length between AERONET measurements and WRF-Chem results, averaged during September 10-20. The correlation coefficient (R) and geometric mean ratio (GMR) are shown in the figure. The plotted WRF-Chem AOD results are divided by 2, in order to see details with AERONET AOD in one color-bar. (a) R-CASE result; (b) F-CASE result.

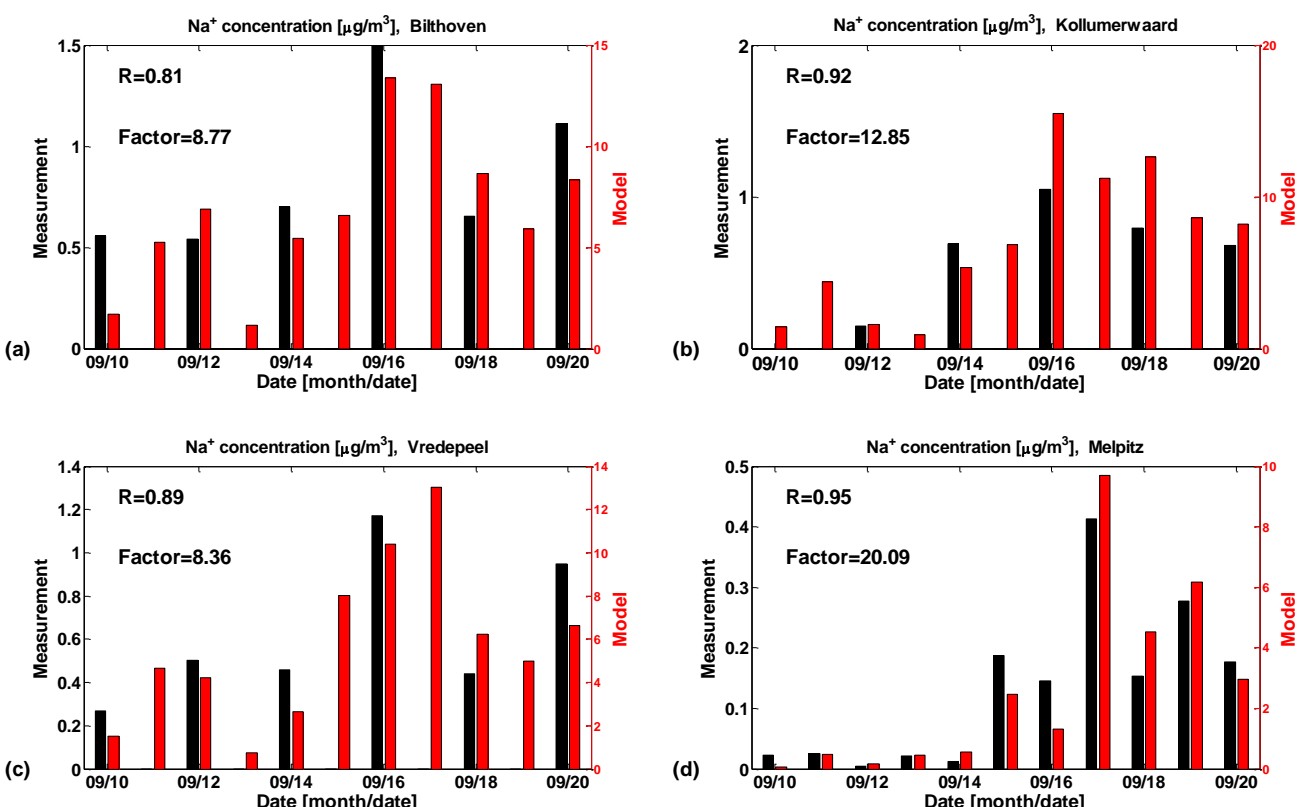

**Figure 7.** Comparison of Na$^+$ mass concentration in PM$_{10}$ between the filter sampler measurements (left y-axis) in 4 EMEP stations and the F-CASE results (right y-axis). (a) Bilthoven; (b) Kollumerwaard; (c) Vredepeel; (d) Melpitz. The locations of the stations are shown in Figure 2.

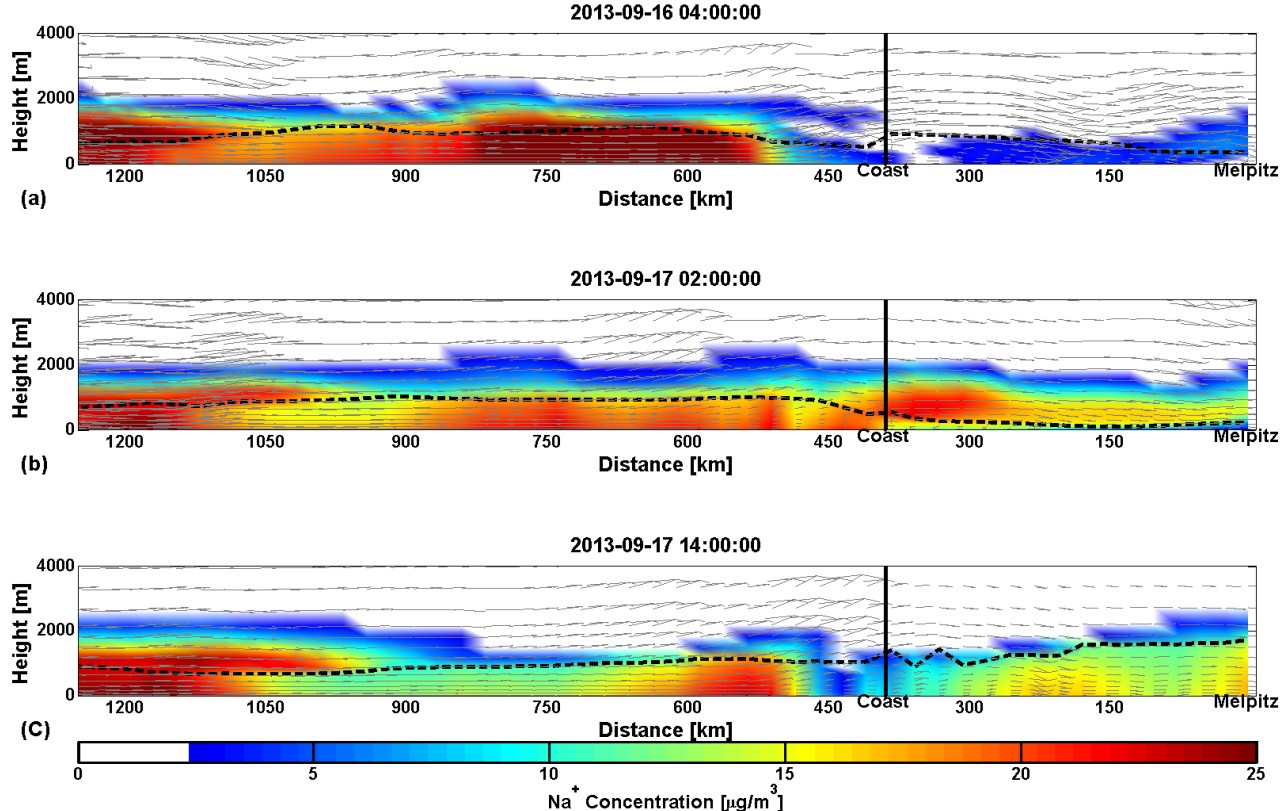

**Figure 8.** WRF-Chem result of the sea salt ($Na^+$) concentration on the vertical cross section, which is shown by the black dash line in Figure 2. The locations of Melpitz and coast (black line) are marked. The grey arrows indicate the wind field, and the black dash line indicates the planetary boundary layer (PBL) height. (a) 2013-09-16, 04:00 LT; (b) 2013-09-17, 02:00 LT; (c) 2013-09-17, 14:00 LT.

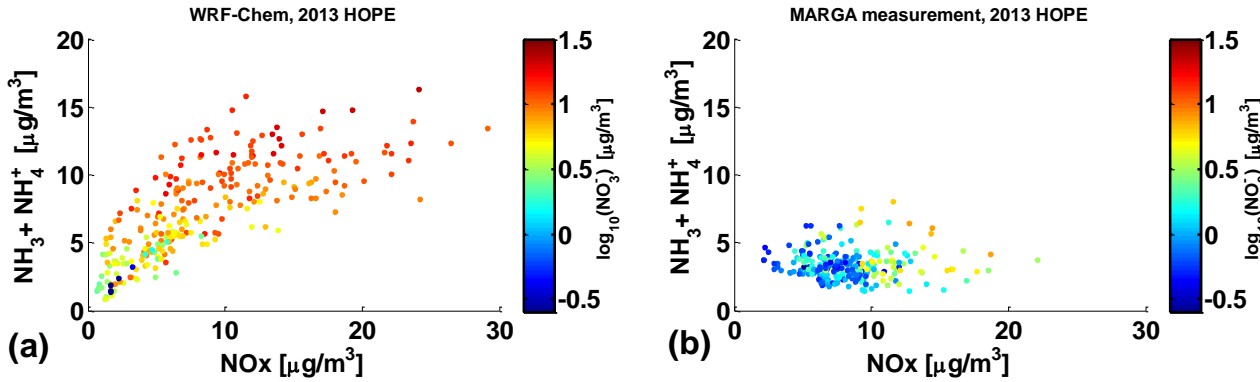

**Figure 9.** Relationship between nitrate, total ammonia and NOx during September 10-20, 2013 at Melpitz. The color indicates the nitrate mass concentration in logarithmic scale. (a) WRF-Chem model results; (b) MARGA measurement results.

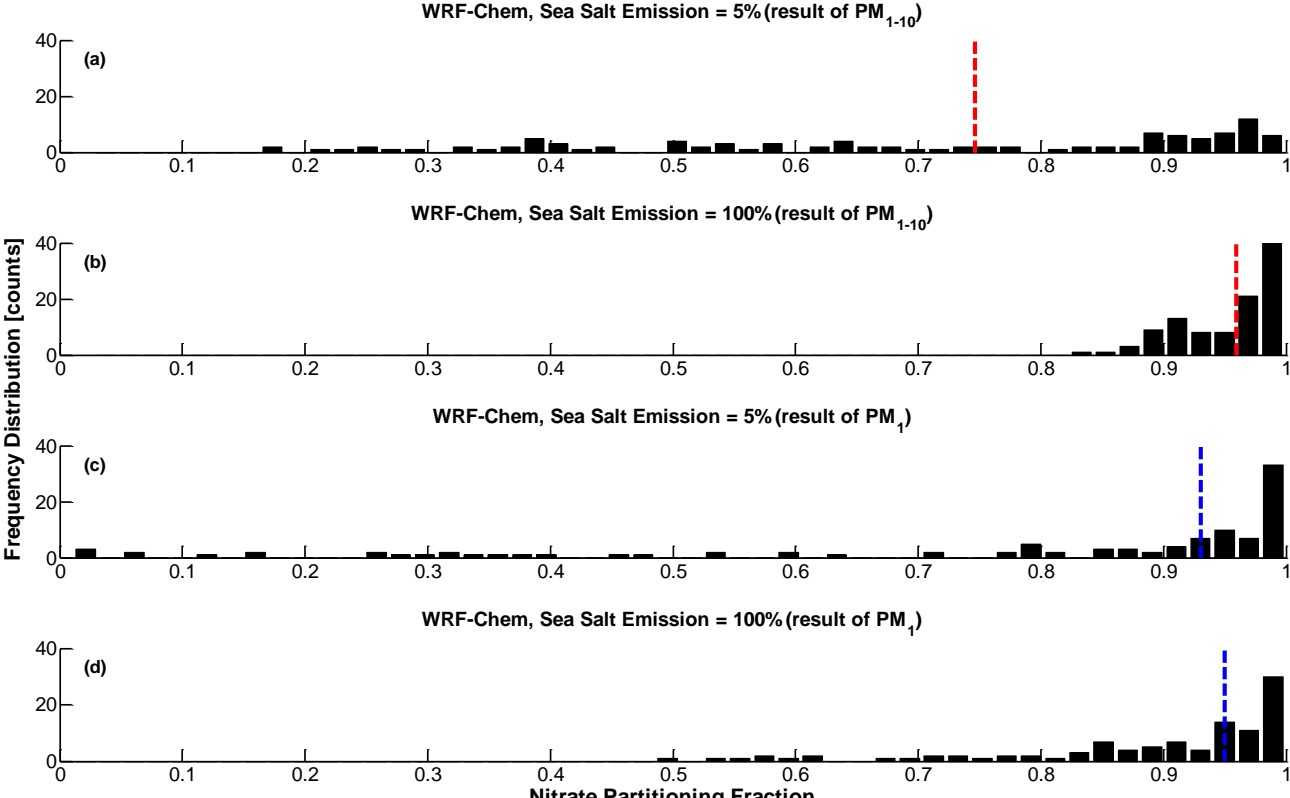

**Figure 10.** WRF-Chem results of the frequency distribution of PF_nitrate at Melpitz. The results were analyzed during the marine period ($[Na^+] > 1.8$ μg/m$^3$ in the F-CASE). The dash lines (coarse mode: red; fine mode: blue) indicate the median value (with 50% probability in both sides). (a) R-CASE result of the size range $PM_{1-10}$; (b) F-CASE result of the size range $PM_{1-10}$; (c) R-CASE result of the size range $PM_1$; (d) F-CASE result of the size range $PM_1$.

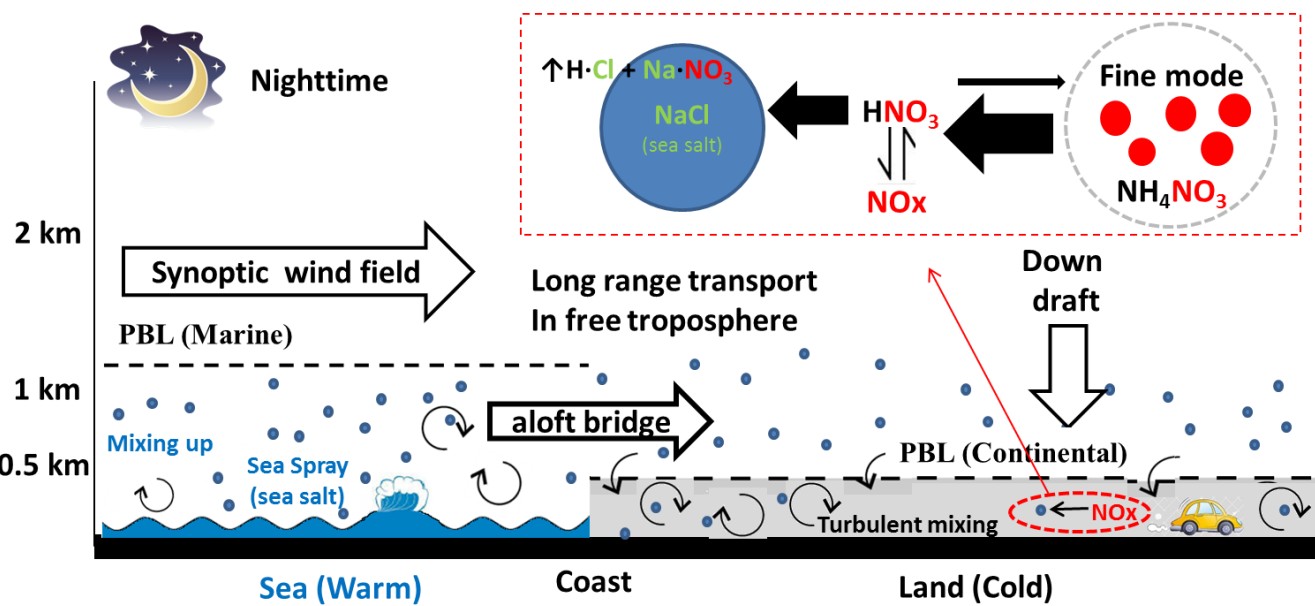

**Figure 11.** Schematic of sea salt transportation and its influence on nitrate particle formation.