# Peer review of "Sea salt emission, transport and influence on size-segregated nitrate simulation: a case study in Northwestern Europe by WRF-Chem"

_Atmospheric Chemistry and Physics, 2016_

## Referee Comment (RC1) · Anonymous Referee #1 · 27 May 2016

The manuscript *Sea salt emission, transportation and influence on nitrate simulation: a case study in Europe* compares modeling and measurement data obtained during the HOPE-Campaign in September 2013. Sea salt sodium concentrations, nitrate concentrations and the particle size distribution are evaluated at the inland station Melpitz. The concentrations of these species are also evaluated at three coastal Dutch EMEP stations. Moreover, the vertical distribution and the medium range transport of sea salt particles is described and discussed in detail, which is one of the highlight topics of this manuscript. A comparison of modeled columnar particle concentrations with measurements – e.g. via AOD data – would be of great value for this manuscript. The authors employed a coupled meteorology chemistry transport model in this study, which is another highlight. Although a comparison with results obtained via an uncoupled model system would be very interesting, it would be too time consuming to perform such

model runs for this study (maybe the authors could keep this in mind for future studies). However, the authors could highlight the advantages of a coupled model setup in the beginning of the manuscript. The impact of sea salt particles on atmospheric nitrate mass concentrations is analyzed in the end of of the manuscript. The presented and discussed results are not new and could be enhanced or removed (see text to questions 13).

The Figures included in the manuscript as well as in the supplement are of good quality and present the results in a clear manner.

The text is easy to understand but has deficits in the scientific language and in the choice of suitable wording in some passages. Moreover, grammar errors or misspellings complicate the understanding of some long nested sentences. Therefore, a revision of the language is recommended.

1. **Does the paper address relevant scientific questions within the scope of ACP?** Yes. The impact of sea salt particles on other atmospheric compounds and the vertical distribution of sea salt particles were evaluated. Both are topics relevant topics within the scope of ACP.

2. **Does the paper present novel concepts, ideas, tools, or data?** Yes. Most previous model studies on the atmospheric transport of sea salt particles and their impact on other atmospheric compounds were performed with uncoupled meteorology and chemistry transport models. In contrast this study is one of the first evaluating with topic by means of a coupled model system (WRF-Chem). Additionally, the authors evaluate the vertical distribution of sea salt particles. However, the evaluation of atmospheric sea salt concentrations against EMEP measurement data is not novel as well as the evaluation of the impact of sea salt on atmospheric nitrate. The discussion of the vertical sea salt concentration profiles would greatly benefit if measurement data on the column sea salt concentrations were additionally considered - e.g. AERONET AOD data.

[Figure]

3. **Are substantial conclusions reached?** Yes, partly. The authors discussed on a quite detailed level why sea salt particles are transported to a measurement station in the hinterland. Further it is found that sea salt concentrations are overestimated by the model which a common result of recent model sea salt studies.

4. **Are the scientific methods and assumptions valid and clearly outlined?** Yes, the methods are clearly outlined. The reasoning for some assumptions in section 3.4 should be revised.

5. **Are the results sufficient to support the interpretations and conclusions?** Yes, the results and their representation in the manuscript are sufficient. The reproducibility would be facilitated if the plotted data were attached as supplement information (as text-CSV, netCDF or another appropriate format).

6. **Is the description of experiments and calculations sufficiently complete and precise to allow their reproduction by fellow scientists (traceability of results)?** Yes.

7. **Do the authors give proper credit to related work and clearly indicate their own new/original contribution?** Yes. In a few situations (see my comments below), additional references were appropriate.

8. **Does the title clearly reflect the contents of the paper?** Yes, in principle it does. The authors might consider to add *Northwestern* in front of *Europe* and to replace *simulation* by *in a coupled meteorology and chemistry transport model* (or *in WRF-Chem*). Since the usage of a coupled model is quite novel with respect to this topics, it is reasonable to add this information.

9. **Does the abstract provide a concise and complete summary?** Yes, it provides a concise and complete summary and is well written.

10. **Is the overall presentation well structured and clear?** The *Data and Methods*, *Results and Discussions*, and *Conclusions* sections are well structured. Some descriptions in the *Results and Discussions* section should be moved to the *Data and Methods* section as indicated by some of the comments below. The authors might consider to restructure the *Introduction* section. I am missing a clear "story line" in the latter section.

11. **Is the language fluent and precise?** The language is fluent but not completely clear throughout the text. In some situations the used expressions are rather colloquial than scientific. In the comments below, some of these expressions are listed. Although it should be noted that the colloquial expressions make the text easier and more fluent to read than with the correct scientific expression (and partly more lengthly formulations). The usage of the definite article "the" and of the indefinite article ("a" in singular; nothing in plural) is mixed in several passages. In the *Introduction* and section 3.4, some sentence structures are not clear – it is unclear weather spelling or grammar mistakes are the reason. A comma (",") has to be placed after "Thus", "Additionally" and similar words, which start sentences. I suggest to revise the manuscript with a focus on these points.

12. **Are mathematical formulae, symbols, abbreviations, and units correctly defined and used?** Yes. The concentrations of substances are written as [X] and X (where "X" is the substance's chemical formula). The writing should be either [X] or X - not mixed.

13. **Should any parts of the paper (text, formulae, figures, tables) be clarified, reduced, combined, or eliminated?** In the current version, section 3.4 does not present new and/or unexpected results. It should be enhanced by new considerations (e.g. the impact of sea salt particles on the vertical distribution of nitrate) or removed.

14. **Are the number and quality of references appropriate?** Yes. Additional references might be reasonable in some passages. These passages are listed in the comments below.

15. **Is the amount and quality of supplementary material appropriate?** Yes, the supplements adds value to the main manuscript. I would suggest to put more than one figure/table on each page in order to saves pages and avoid large white spaces. Page numbers would be favorable. Furthermore, the authors should verify whether reprint of the three plots in Fig. 4 violate Copyright laws. Also see the answer to point 5.

**General and Scientific Comments:**

1. p.1, l.20: "... and has significant impact on the formation on secondary inorganic aerosol particles on global scale.". Reading this sentence might imply that the presence of sea salt particles favors the formation of sec. inorg. aerosols (SIA). However, this is not the case as the authors probably know. Sea salt has an indirect impact on particle formation because compounds such as H2SO4, $HNO_3$, and $HNO_3$, which tend to form new particles, condense onto sea salt particles surfaces instead. Hence, the particles formation is decreased. Please reformulate the sentence.

2. p.2, l.8-9: In which context do sea salt particles **participate** in **heterogeneous reactions**? The salt particles provide surface area for heterogeneous reactions but using the work "participate" might be misleading. Please also clarify the meaning of "leading to the formation of secondary aerosols" (see 1.).

3. p.2, l.10: "... significant influence on **nitrate formation** ...". The deprotonation of an acid ($HNO_3$) should not be denoted as the **formation** of the deprotonated version of this acid ($NO_3^-$). If nitrate was formed from different compounds via heterogeneous reactions at the particle surface, **nitrate formation** was appropriate. However, the latter situation is not the case, here. The usage of "formation" in connection with "nitrate" arises in further text passages, such as p.2, l.17. Please considered replacing "formation".

4. p.2, l.11: "chlorine deficit": Commonly, it is denoted as "chlorine displacement".

5. p.2, l.13-15: Please give the reason for the difference in the cations bound to nitrate between central/western and northern/southern Europe. → extensive animal husbandry in central and western Europe associated with high ammonia emissions.

6. p.2, l.16: "short life-time due to its quick deposition within the coastal region": The other way around: Coarse sea salt particles have a short life time and, therefore, they depose close their source. If the sea salt is emitted close to the coast the it also deposes in coastal regions.

7. p.2, l.24: Please give a reference for the first sentence's statement (and "much more" is colloquial style).

8. p.2, l.25-26: "The parameterization schema . . . ": consider introducing an abbreviation for the schema, such as GO03 as common in the literature.

9. p.2, l.35: Consider starting a new paragraph, here.

10. p.2, l.36-37: "The uncertainty of the SSA emission scheme directly determines the uncertainty of the evaluation of SSE radiative forcing." This is partly correct, because the deposition – particularly variable dry deposition for variable sea salt particle size distributions – plays a relevant role.

11. p.2, l.37-39: Consider switching (and slightly reformulating) the two sentences starting with "Additionally" and "the heterogeneous".

12. p.3, l.3: "nitrate simulation": reformulate; "nitrate prediction"?

13. p.3, l.6-9: Please consider to mention "MOSAIC" and "CBMZ" already in this first paragraph. The detailed explanation further below is fine.

14. p.3, l.6-7: Consider extending "...regional air quality model." to "...regional meteorology and air quality model system.".

15. p.3, l.8-9: "In addition to meteorology, aerosols, trace gases and interactive processes ...": "meteorology" was not mentioned before but everything behind meteorology was indirectly mentioned by writing "air quality model". Please consider reformulating the sentence.

16. p.3, l.10-11: Please clarify in the text that MOSAIC is the employed aerosol module in WRF-Chem and not an individual modeling system.

17. p.3, l.13-16: Please state a first, why the bin is split (PM1 and PM1-10 calculation), and then, how it is done.

18. p.3, l.18-19: "Both particle mass concentrations and particle number concentrations are simulated.". Question (I am not familiar with the sectional particle representation in MOSAIC): Should the particle number and mass concentrations not be related via the size range of the bin? If the number concentration of particles of a pre-defined size (e.g. 625 nm to 1250 nm in size bin 5) is known, then the particle volume concentrations (assuming uniform size distribution in this bin) and mass concentrations can be directly calculated. Why are number and mass individually modeled per bin (which could result indirectly in particle sizes outside of the bin's size range).

19. p.3, l.39 to p.4, l.1: Please consider to describe (a) the outer, the intermediate, and the inner domain (in this order) or (b) the inner, intermediate, and outer domain but not (c) the outer, the inner and then the intermediate domain.

20. p.4, l.2: Consider adding "time" or "period" after "spin-up".

21. p.4, l.3: Please add "NCEP" in front of "sea surface temperature". Were the FNL data used as meteorological boundary conditions for the outer model domain?

22. p.4, l.7: Please update the url to MOZART (http://www.acom.ucar.edu/wrf-chem/mozart.shtml).

23. p.4, l.6: "The initial chemical and boundary conditions ...". Please switch the position of "chemical": "The chemical initial and boundary conditions ...".

24. p.4, l.9: Please introduce and describe the F-CASE and and the R-CASE.

25. p.4, l.10: "SSA results from **dried sea spray** ...": The use of "dried" could be misinterpreted by readers as "dry sea salt".

26. p.4, l.12: Neumann et al. (2016) is no primary reference for this statement.

27. p.4, l.20: A side note to the choice of the adjustable parameter: Gantt et al. (2015) (doi: 10.5194/gmd-8-3733-2015) suggests a value of 8 instead of 30.

28. p.5, l.18: "with a temporal resolution of 2 days." The formulation is misleading, because it might be understood by readers that two-day filter samples (48-hour averages) are collected at the Dutch stations. Actually, one-day filter samples (24-hour average) are collected every second day as the authors know and correctly plotted in Fig. 5. Therefore, please reformulate.

29. p.6, l.1: "be unrealistic sources" → "be unrealistic **high** sources"?

30. p.6, l.22-25: The cited studies do not explicitly focus on Northwestern Europe. Tsyro et al. (2011) (doi: 10.5194/acp-11-10367-2011) presented an extensive model study on sea salt concentrations in Europe spanning several years. Manders et al. (2010) (doi: 10.1016/j.atmosenv.2010.03.028) also compared sodium model and measurement data at several EMEP stations. Both found overestimations. Neumann et al. (2016) found overestimations in winter. The authors might consider the refer to these studies, because they focus on a similar region as this manuscript does.

31. p.7, l.3: "...SSA was emitted near the surface layer ...": Sea salt particles should be emitted into the surface apart from the situation, in which sea salt is emitted from the top of a giant wave higher than the model surface layer. However, in the latter situation we probably need another emission parameterization and should not use 10-m wind data.

32. p.7, l.11-22: The authors could consider to describe some information of this paragraph in the Data and Methods section.

33. p.7, l.13: "...promoting the formation of secondary inorganic aerosol (SIA), ...": Secondary aerosols or secondary particles denotes the **new** formation of particles in the atmosphere. The presence sea salt particles enhances the $HNO_3/NO_3^-$ condensation and, hence, one could reformulate the sentence into "...promoting the formation of secondary inorganic particle mass ...".

34. p.7, l.15: "Part of $HNO_3$ will participate in the partitioning process and form particulate nitrate." Misleading. The whole $HNO_3$ is involved into the partitioning process. One part remains in the atmosphere and the other part condenses. The condensed part becomes nitrate. → Suggestion: "$HNO_3$ undergoes (maybe another word) a partitioning process between gas phase and liquid particle phase via condensation. The condensed $HNO_3$ deprotonates to $NO_3^-$."

35. p.7, l.18: "irreversible reaction", better "irreversible process"

36. p.7, l.35-36: Why are 5% of the original sea salt emissions emitted in the R-CASE and not 20%, 10%, or 1%? Is this value arbitrarily set or is it related to the 20-fold overestimation of sodium PM10 by the model (1/20 = 5%)?

37. p.7, l.39: "simulation" → "prediction"? (see comment 12.)

38. p.7, l.43-45: Consider switching the order of both sentences.

39. p.8, l.1-7: Data and Methods section?

40. p.8, l.8: "probability density function": "frequency distribution" might be more appropriate (also at the subsequent occurrences in the paragraph)

41. p.8, l.9-10: Please clarify "marine period (Na+ $> 1.8\mu g/m^3$ in F-CASE).". It means that only PFnitrate values of model time steps with [Na+] $> 1.8\mu g/m^3$ were considered? How many model time steps were considered? Please be consistent with the notation

of concentrations: Above, [HNO₃] denotes the concentration of HNO₃ but, here, Na+ (and not [Na+]) denotes the concentration of Na+.

42. p.8, l.12-13: "also there was uncertainty of the precursors emissions in the model." There are always uncertainties in the emissions. Therefore, the uncertainty is not a general reason not to compare model and measurement data. Note: It should be "precursor's emissions" or "precursor emissions".

43. p.8, l.15: "were used as reference": A reference for what?

44. p.8, l.15-16: " Considering that most of the SSA was **emitted** as coarse mode particles (about 88% in both file measurement and simulation on September 17, 2013 **at Melpitz**), ... ": The formulation is misleading. There are no emissions measured at Melpitz but concentrations. Since coarse particles have a higher dry deposition velocity than fine particle one can expect that the emissions consisted by more than 88% of coarse particles.

45. p.8, l.17-18: "... also more sensitive to the change of the SSA emission.": Why? The particle surface area is the parameter governing the condensation of nitric acid. Higher mass emissions cannot be directly related to higher particle surface area emissions.

46. p.8, l.20-21: "... median ... was about 0.75 ... broad spreading ... ": There should be something like "... and distribution " between the "0.75" and "broad spreading".

47. p.8, l.20-21: "... increased the coarse mode nitrate partitioning fraction by 0.2.": Unclear whether 0.2 is a difference or a quotient. It is the difference, but the formulation is ambiguous.

48. p.8, l.22-24: The conclusion is justified because the ammonium mass concentrations are quite similar in both cases - otherwise not. Therefore, it might be reasonable to repeat that information in this summarizing sentence. Please consider splitting this sentence into two sentences.

49. p.8, l.25-26: "...indicating that in our case the overestimation of SSA emission is mainly in the coarse mode.": What is the reasoning for this conclusion? No comparison against measurement data were performed.

50. p.8, l.28: "consumption of precursors": Why plural? Are there other precursors than $HNO_3$ for particulate $NO_3^-$?

51. p.8, l.29: "PM10" → "fine PM" or "PM1"?

52. p.8, l.33; p.9, l.23: "overestimated by 0.2": see comment 46.

53. p.8, l.35: "particle number distribution": There were no number concentrations considered in this study.

54. p.8, l.39; p.9, l.20: "simulation": see comment 12.

55. p.9, l.3, 10, 20, 30: "uncertainty": Uncertainty describes *instable* deviations (in some situations values are overestimated and in other situations they are underestimated). Here, the parameterization clearly overestimates the emissions. Therefore, "uncertainty" is not necessarily the correct word.

56. p.9, l.6: "The variations" → "The spatial variations"

57. p.9, l.8: "...the overestimation in ..." → "...the overestimation of emissions in ..."

58. p.9, l.20: "Fig. 9": Please do not include new Figures in the Conclusions. This Figure should be described in an earlier passage of the manuscript or removed. The first choice is favorable because the figure describes the transport of sea salt particles to Melpitz very well and clear.

59. p.9, l.25: "gas-phase precursors", see comment 50.

60. p.9, l.31: "formation of secondary inorganic aerosol", see comment 33.

61. p.9, l.37-38: Last sentence: If the authors want to write about $NO_X$ it should be

done in an extra paragraph of the *Conclusions* section and not in the last sentence. The second last sentence might be a nice last sentence.

**Comments on Figures and Tables:**

1. p.16, Table 2: The authors might consider to split the three columns into five (one column each for "Factor" and "R"). "Factor" should be explained in the caption.

2. p.17, Table 3: see comment 61.

3. p.18, Fig. 1: "shown in Figure 5": it is **Figure 6**; "domain 02" == intermediate domain?

4. p.20 and 21, Fig. 3 and 4: The title above the plots is inconsistently written: "10-20th Sep. 2013, Melpitz" (Fig. 3) and "10-20th September, 2013".

5. p.22, Fig. 5: The fact that two y-axes exist per plot should be noted in the caption. The authors could consider to order the plots by the stations' distance to the coast or geographic location (== Melpitz as plot (d)) because it is more intuitive with respect to Fig. 1.

6. p.23, Fig. 6: The authors might consider inverting the x-axis because it is more intuitive for the reader to have the coast on the left and Melpitz on the right.

7. p.24, Fig. 7: The length of the x-axis could be cropped.

8. p.25, Fig. 8: The caption is complicated formulated. Consider reformulating it. Additionally: "probability" (1st and 4th line), see general comment 40.

9. p.26, Fig. 9: see comment 6 to Fig. 6 and general comment 58.

**Comments on Language and Spelling:**

1. p.1, l.33-36: Please split this sentence into two. schulz 2. p.1, l.41: Please change "Atmospheric aerosol plays" to "Atmospheric aerosols play" to be consistent with the

next sentence ("Further **they** ...").

3. p.1, l.42: Please change "could either be" from conjunctive to indicative ("are either").

4. p.1, l.43: "constitute" → "constituent".

5. p.2, l.3-6: Please reformulate the sentence. There are some typos or the grammar is incorrect.

6. p.2, l.7: no conjunctive, see 3.

7. p.2, l.13: "... sodium nitrate is largely contributed to nitrates ... ": Please use active voice ("... sodium nitrate contributes ... ").

8. p.2, l.16: "quick" → wording

9. p.2, l.16: "... region (Grythe et al., 2014), thus ... " → "... region (Grythe et al., 2014). Thus, ... " (split sentence; comma after "thus").

10. p.2, l.17: "cannot reach the distant inland area." the meaning is clear but colloquial language; Why "**the** distant inland area."?

11. p.2, l.20-21: Please move "later on" to the end of the sentence because it specifies a time.

12. p.2, l.22-23: "... provide an opportunity ... " → colloquial. If the guards in a prison do not look after the prisoners, then they provide an opportunity for a prison break. However, the mechanisms do not provide an opportunity for sea salt particles. Also colloquial: "... make their influence more extensive ... ".

13. p.2, l.26: "... is still highly uncertain ... ": please reformulate

14. p.2, l.34: "... in varying degrees ... ": possibly "... **by** varying degrees... " might be correct; please check

15. p.2, l.39: "... **needs** the **participation** of ... " → colloquial

16. p.2, l.41-42: "... **make** the importance of SSA indirect effect on nitrate formation over a **broader region**.". Please reformulate.

17. p.2, l.43-44: "In this study ... by a case study ...". Please remove the duplication of "study".

18. p.3, l.21-22: Please reformulate the sentence starting with "However,".

19. p.3, l.23: "The formation mechanism of ...". Please change to "The formation of ..." or "A formation mechanism of ...".

20. p.3, l.38-39: "from September 10-20, 2013". Please change to "from September 10 to 20, 2013" or "in the time period September 10-20, 2013" or choose another formulation.

21. p.3, l.40: "... covers **the** whole Europe, part of the North Sea and the North Africa ...". Please remove the two bold "the" and add an "a" in front of "part" and please do the same in the succeeding lines.

22. p.4, l.11: "... emitted from bubble bursting or breaking waves torn by winds at wave crests." → "... emitted **by** bubble bursting or breaking waves **or** torn **of** wave crests by winds."

23. p.4, l.16: "was" → "is"

24. p.4, l.22: "temporal" → "temporally"

25. p.4, l.24: move "code" behind the bracket ("(SNAP) code")

26. p.4, l.26: insert "the" in front of "anthropogenic emission inventory"

27. p.4, l.31: "consists with" → "has"

28. p.5, l.1-2: "the stations all over ... vertical structures." Please reformulate.

29. p.5, l.42: "bin 05-08" → "bin**s** 05-08". Occurs several times.

[Figure]

30. p.6, l.10-11: Please reformulate the sentence.

31. p.6, l.17: "variance/trend": Maybe "temporal pattern"?

32. p.6, l.36: "400 km away from coast" → "400 km distant to the coast"?

33. p.6, l.39-40: "...about 30-40% of SSA mass concentration was actually transported to the inland (Melpitz) comparing to the coast regions." → "about 30-40% of **the initial** SSA mass **at coastal stations** was actually transported to the inland **station of Melpitz**.".

34. p.6, l.41: "will be discussed": Unclear; Where? When?

35. p.6, l.43-44: "...the warmer sea surface resulted in a higher planetary boundary layer (...) than that over the continent." → "...the warmer sea surface resulted in a higher planetary boundary layer (...) **above the sea than** over the continent."

36. p.7, l.4: "...was able to penetrate ..." → "...penetrated ...".

37. p.7, l.11: "from" → "by"

38. p.7, l.12-14: First part of the sentence unclear. Please reformulate.

39. p.7, l.20: "The participation of SSA ..." → "The presence of SSA ..."

40. p.7, l.25: "Either it could result from inaccurate emission of precursors or an improper chemical pathway." → "The overestimation could result either from inaccurate emissions of precursors or from an improper modeled chemical pathway." (suggestion)

41. p.7, l.26-34: Please reformulate the passage. One can interpret what is meant in this passage but the formulation and sentence structure make the understanding difficult.

42. p.7, l.35: "a sensitive study" → "a sensitivity study"

44. p.9, l.11: "continent" → "the continental"

45. p.9, l.12-15: Split into to sentences and replace "participate" by another verb.

46. p.9, l.17-18: "made the SSA overestimated by a factor of 20 at Melpitz": Consider replacing "made" by "led" or "yielded" and reformulate.

47. p.9, l.20-22: Please reformulate.

48. p.9, l.26-27: "..., resulted from coarse mode nitrate formation with participation of SSA, may slow down the formation of fine mode nitrate." → "..., resulting (or: which resulted) from coarse mode nitrate formation, reduced the formation of fine mode nitrate."

49. p.9, l.32-33: "Later on, these changes will alter ..." → "These changes alter ...".

---

## Referee Comment (RC2) · Anonymous Referee #2 · 28 Jun 2016

**General Comments:**

The manuscript "Sea salt emission, transportation and influence on nitrate simulation: a case study in Europe" studies the transport of sea salt aerosol using the WRF-CHEM model and compares the modelling results to measurements obtained during the HOPE-Campaign in September 2013. The meteorology simulations were validated against surface meteorological observations as well as the vertical distribution of meteorological parameters obtained by radiosonde measurements, and both confirmed that the simulation could capture the meteorological condition very well. The aerosol number/mass concentration distribution, however, displayed a large discrepancy in the coarse mode size range, which the author attributes to overestimated SSA emissions in the model emission scheme. The author studies the difference in thermodynamic stratification over land and sea and points out the mechanism for the long-range transport of SSA, which extends the influencing range of SSA further inland to the Melpitz station. The author further studies the effect of overestimated SSA on particulate nitrate simulation results. Here are some general comments:

1. The impact of SSA on nitrate partition seems to be nothing new. The author mentions at the end of the conclusions the potential impact of overestimated SSA and nitrate on radiative forcing and aerosol hygroscopicity, it would be perhaps more interesting to see some discussion on that.

2. The model output frequency is not clarified in section 3. Did you compare hourly model data with observations? In the comparison of simulated & observed meteorological data, the author calculates correlation coefficient. However, many meteorological parameters, such as temperature and wind, have significant diurnal variations, which can be easily captured in the model. If you calculate correlation coefficients between hourly data, the diurnal variations which agree with each other very well might also lead to high correlation coefficients, which does not necessarily mean that you could capture the day-to-day variation well. Why did you not directly compare the absolute values between model & measurements, especially for the wind direction data?

3. Although the manuscript is easy to understand, there are still many grammatical errors and the scientific language is not always precise, please go through the whole text carefully and revise the language to improve the reading experience of your readers.

1. Does the paper address relevant scientific questions within the scope of ACP?

   *Yes.*

2. Does the paper present novel concepts, ideas, tools, or data?

   *Yes.*

3. Are substantial conclusions reached?

   *Yes.*

4. Are the scientific methods and assumptions valid and clearly outlined?

   *Yes. However, there can be improvements in the methods section.*

5. Are the results sufficient to support the interpretations and conclusions?

   *Yes.*

6. Is the description of experiments and calculations sufficiently complete and precise to allow their reproduction by fellow scientists (traceability of results)?

   *Yes.*

7. Do the authors give proper credit to related work and clearly indicate their own new/original contribution?

   *Yes.*

8. Does the title clearly reflect the contents of the paper?

   *Yes.*

9. Does the abstract provide a concise and complete summary?

   *Yes.*

10. Is the overall presentation well structured and clear?

    *Yes.*

11. Is the language fluent and precise?

    *It is overall fluent, however, improvements are needed to make it more precise.*

12. Are mathematical formulae, symbols, abbreviations, and units correctly defined and used?

    *Yes.*

13. Should any parts of the paper (text, formulae, figures, tables) be clarified, reduced, combined, or eliminated?

    *No.*

14. Are the number and quality of references appropriate?

*Yes.*

15. Is the amount and quality of supplementary material appropriate?

    *Yes.*

**Specific Comments:**

**Abstract:**

1. P1L26: "…, the modeled SSA concentrations were overestimated by a factor of 8-20."→ ", the model overestimated SSA concentrations by factors of 8-20.

2. P1L27: "…over North Sea…"→"…over **the** North Sea…", this needs also to be corrected for the later occurrences in the manuscript.

3. P1L32: "broadened" → "extended"

4. P1L35-36: "increased by about 0.2 for the coarse mode nitrate…., but no significant difference in the partitioning fraction for the fine mode nitrate." → "increased by about **20%** for the coarse mode nitrate…, but no significant difference in the partitioning fraction for the fine mode nitrate **was found**."

**Introduction**

1. P1L41: "Atmospheric aerosol plays… Further they have an …"    rephrase these two sentences, if you want to use "they", you should change the first sentence to "Atmospheric aerosols…"

2. P1L43: change to "on **a** global scale"

3. P2L1: "…,  comparable with…"

4. P2L3-5: Rephrase to "Waves breaking in the surf zone, where **there are** more whitecaps and stronger **SSA (?)** emission due to **increased** ocean bottom and higher intensity of wave breaking, may affect **SSA concentrations at areas within 25 km distance from** the coastline and can dominate **the SSA concentration** at the coastal region"

5. P2L9-10: "nitrate formation" is slightly inappropriate, since the $HNO_3$ was already formed in the atmosphere. The SSA only influenced its gas and aerosol phase partitioning. Please consider

rephrasing.

6. P2L13-14: Change to "…, sodium nitrate largely contributes to nitrates in northern and southern Europe"

7. P2L22-23: Change to "…and thereby could expand/extend their influencing range from coastal to regional or even global."

8. P2L24-25: Change to "However, in terms of global mass concentration, …"

9. P2L35: Change to "…for the evaluation of the **its** climate effect"

10. P2L41-42: Change to "Furthermore, the long-range transport mechanisms, as mentioned above, extends the **impact** of SSA indirect effect on nitrate formation **to** a broader region."

11. P2L44: Rephrase as "The model parameterization schemes…"

12. P3L1-3: Please change the tense in these three lines to present tense.

**Section 2**

1. P3L41: Consider adding the domain range of D01 to Figure 1.

2. P4L2: "The spin-up **time** of the model run was 2 days."

3. P4L8: "More details **on simulation**  setups and parameterizations  are given in Table 1."

4. P4L10: Rephrase to "SSA are produced through the evaporation of sea sprays, which were ejected into the atmosphere from the sea surface."

5. P4L12-13: "The parameterization scheme for SSA emission coupled in the WRF-Chem model follows the Gong (2003) scheme."

6. P4L17: "…, which controls the shape of submicron **SSA** size distributions"

7. P4L31: "…and has  the same spatial resolution"

8. P4L42: "Measurements of **the** HOPE-Campaign". The "the" is often missing, please go through the manuscript carefully and make the language more fluent.

9. P5L3: "The Melpitz Obervatory is representative of  the regional background of Central Europe"

10. P5L5,9: There are many abbreviations in the text that appear without explaining what they stand for, e.g. WMO-GAW, ACTRIS, MARGA, etc.

11. P5L11-12: "This instrument provided 1-hour data of secondary inorganic aerosols (…) and

gaseous counterparts (…)." → I would suggest adding the detailed species that were measured into these brackets.

12. P5L12-14: Did you have two high volume samplers respectively for PM10 and PM1? If yes, rephrase to: "The high volume samplers DIGITEL DHA-80 (Walter RiemerMesstechnik, Germany), with **a** sampling flow of about 30 m³h⁻¹, **were used to collect 24-hour PM10 and PM1 filter samples simultaneously** (Spindler et al., 2013).

13. P5L14-16: "Information on the **coarse mode (PM1-10)** aerosol chemical compositions, such as nitrate and sodium etc.,  were obtained from the difference between the results of PM10 and PM1.

14. P5L14-16: "Additionally, **24-hour** filter sampler measurements with PM10 inlets (EMEP, 2014) at 3 coastal EMEP station near the SSA transportation pathway (Bilthoven, Vredepeel, and Kollumerwaad, see Fig. 1)**, which were collected every second day,** were obtained from EBAS (http://ebas.nilu.no/)"

**Section 3**

1. P5L21: "over  Northern Germany"

2. P5L25: "Evidently, strong vertical **motion** occurred in the coastal region, which  lifted SSA upward."

3. P5L28-29: "Simulated **surface** temperature, relative humidity, wind speed and wind direction were in good agreement with **ground** measurements, with  correlation coefficients…"

4. P5L36-37: "Corresponding, R values were 0.99, 0.96, 0.84 and 0.92 for potential temperature, wind speed, wind direction and water vapor mixing ratio, respectively." Are these vertically averaged correlation coefficients between simulated vertical profiles and radiosonde measurements? If so, please rephrase the sentence to make that clear.

5. P6L1: Rephrase as "Therefore, unrealistic sources of coarse particles might be the cause for the overestimation."

6. P6L13-14: "Marine air masses **first** arrived at **the** three coastal stations.

7. P6L17: "As shown in Fig. 5 the **day-to-day variation** of Na+ concentrations can be captured by the model…"

8. P6L26-27: "The uncertainties of this scheme may be **attributed** to **the lack** of parameters, …"

9. P6L32-33: "Generally, SSA is mostly in coarse mode with **a** lifetime shorter than 2 days in the continental boundary layer **and reaching** about 1 week in free troposphere"

10. P6L35: This sentence is hard to understand and needs rephrasing, consider "According to the simulation results, the component of the 10m wind vector that is directed from the coast to Melpitz shows a wind speed in the range of 2-3 m s$^{-1}$"

11. P6L35-36:"It **would**  therefore **take** about 1.5-2 days for SSA to be transported to Melpitz (~400 km away from coast)."

12. P6L36-38:The result (Fig. S5) from **the** Deposition-Lifetime Concept Model (Chen et al., 2016; Croft et al., 2014) **indicates** that on average only  10-35% of **the emitted** SSA could be transported to Melpitz through the surface pathway.

13. P7L6-8: "Therefore, about 70-85% of SSA (Fig. S5) could be carried further towards **the** inland in free troposphere, and arrived **at** the Melpitz region in the early morning of September 17 (Fig. 6b)."

14. P7L11-12: "As discussed above, the **over-production** of SSA from the WRF-Chem SSA emission scheme will **lead** to an 8-20 times overestimation of the primary sea salt mass concentration."

15. P7L15: Rephrase to: "Part of HNO$_3$ will be partitioned into the condensed phase and form particulate nitrate."

16. P7L17-18: " The other one is the irreversibe reaction with SSA (NaCl) and the formation of sodium nitrate with depletion of chloride.

17. P7L21-22: I believe what you want to say is that the condensation process of HNO$_3$ onto particles is facilitated by the participation of SSA, replace "partition" with "condensation": "The participation of SSA might facilitate the **condensation** process of nitrate."

18. P7L25: "This could either result from an inaccurate emission of precursors or from an improper chemical pathway in the model."

19. P7L30-34: Please consider rephrasing this part into: " However, **even under the same mass concentrations of precursors, the simulated nitrate mass concentrations (Fig. 7a) were still much higher than the observed ones (Fig. 7b), which indicates that** in addition to an overestimation caused by overestimated

**NH₃** emission (see also Table 2), improper chemical pathway also contributed to the nitrate overestimation. "

20. P7L35-36: "In order to quantify the influence of NaCl on the nitrate partitioning, a **sensitivity** study was implemented with only 5% of SSA emission (R-CASE)."

21. P7L42: "However, NOx and total ammonia **concentration** results of **the R-CASE did not show significant changes** (Table 2)."

22. P8L10-13: 1. The later sentence is incomplete; 2. The difference in size range is a reasonable reason why the two should not be directly compared. The uncertainties in measurements and in the model emissions always exist, we need to keep those in mind when comparing measurements with model results, but they are not the reason why the two should not be compared. Consider rephrasing this part into: "Since the MARGA measurements were only available for the size range of PM10, PF_nitrate derived from MARGA observations should not be directly compared with the simulated one. Additionally, we need to keep in mind that high uncertainties exist in the HNO₃ measurements due to its sticky property and in the model precursor emissions, which brings further difficulty into the comparison between measurements and simulation."

23. P8L18-20: This sentence needs rephrasing, consider "As shown in Fig. 8a and Fig. 8b, the median value of coarse mode PF_nitrate in the R-CASE was about 0.75, with the distribution broadly spread in the range of ~0.2 to 1, whereas in the F-CASE the median value increased to 0.96, with a much narrower distribution."

24. P8L26-27: "Although the fine mode PF_nitrate **revealed** no significant difference between R-CASE and F-CASE simulations…"

**Conclusions**

1. P8L39-40: "…, the WRF-Chem model was used to simulate **the aerosol** physical and chemical properties during **the** HOPE Campaign…"

2. P9L2-4: The overestimate in coarse mode nitrate is also caused by the overestimate in SSA emissions, which is also summarized later on in the following text. I would suggest not to

mention it here, rephrase as: "The coarse mode particles were**,** however**,** significantly overestimated both in number and mass, due to an overestimate in SSA emissions caused by the current SSA emission scheme.

3.  P9L6: "The **day-to-day** variations of SSA mass concentrations…"

4.  P9L19-20: Change to "The overestimation in SSA emissions not only influences the primary SSA simulation itself, but also leads to significant uncertainties in the particulate nitrate simulation."

5.  P9L25: "However, the increas**ed** consumption of the gas-phase precursor ($HNO_3$), **caused by the** coarse mode nitrate formation with **the** participation of SSA, may **inhibit/repress/reduce (?)** the formation of fine mode nitrate."

6.  P9L35-39: Change to: "Due to the "aloft bridge" transport mechanism, as described in this paper, the influences of SSA are not only confined to the coastal region, but are extended to a broader region reaching as far as 400 km from coast. Meanwhile, the outflow of continental air mass can transport NOx to the ocean region (Fig. S1), where these influences of SSA on nitrate may also be significant."

---

## Referee Comment (RC3) · Anonymous Referee #3 · 28 Jun 2016

The authors apply WRF-Chem to investigate the effect of sea salt on aerosol nitrate concentrations and the transport mechanisms of sea salt aerosol to an inland site. Additionally, the results of the applied WRF-Chem setup are evaluated against observations. Although the impact of sea salt on aerosol nitrate in general is nothing new, the paper includes sufficient novel aspects and interesting details for a publication in ACP. One important finding is the overestimation of sea salt emissions by WRF-Chem's Gong (2003) sea salt emission scheme. Some more in-depth discussion seems desirable here, e.g. how well the wind speed in the source area are represented or how the applied scheme compares against the other sea salt emission schemes which are included in WRF-Chem.

The paper is easily comprehensible. However, it includes numerous language lapses, such as wrong usage of singular and plural, missing articles etc. The co-authors are

requested to support the lead author here. Also, some of the figures could be improved in some aspects.

Detailed comments:

P 5, l 30: How were the correlations calculated, from hourly values or from mean values? How well are spatial patterns represented? Please discuss also absolute error or mean bias.

P 5, l 42: How well match observed and simulated concentrations of the small particles?

P 6, l 1 and2: Please give some more evidence for this statement.

P 6, l 43: How was the PBL height estimated?

P 7, l 9: According to Fig. 6b, the sea salt layer does not yet touch the surface. What is the contribution of turbulent mixing after sunrise?

P 7, l 25 and 26: There could be also some other reasons, wrong turbulent exchange, wrong water uptake (also due to wrong relative humidity), . . .

P 7, L 29: Why can this be expected?

P 7, l 1 – 10: Please change the order of the figure, Figure 9 should be discussed here.

P 8, l 8: Is this really a probability distribution or a frequency distribution?

Figure 1 and Figure 6: Please consider using a different color scheme. In particular, the dark blue color for the low values is quite unfavorable and the blue arrows (and the map in Fig. 1) can hardly be recognized.

Figure 3: Please show also the R-case.

Caption of Fig. 4: Please mention which case is shown.

Caption of Fig. 5: Please mention particle size. Please mention the different scale for

observations and model results.

Figure 6 and (current) Figure 9: These figures should be oriented from West (left)to East (right). No need for the star, as Melpitz is located at the Eastern end of the figures.

Minor issues: P 2, l7: Partitioning is no 'formation'.

P 2, l 13: '. . . sodium nitrate is largely contributed to nitrates': please reword.

P 2, l 21, 22: 'opportunity' and 'make their influence more extensive': please reword

P 2, l 33: Southern ???

P 2, l 43: influence on what?

P 3, l 27: Please mention first that a resistance approach is applied.

P 6, l 10: Please reword: an event cannot be emitted.

P 7, l 33: A word seems to be missing here.

P 9, l 38: the last sentence is incomprehensible.

---

## Author Comment (AC1) · 18 Aug 2016

**Response to comments of referee #1**

**General Comments:**

The manuscript Sea salt emission, transportation and influence on nitrate simulation: a case study in Europe compares modeling and measurement data obtained during the HOPE-Campaign in September 2013. Sea salt sodium concentrations, nitrate concentrations and the particle size distribution are evaluated at the inland station Melpitz. The concentrations of these species are also evaluated at three coastal Dutch EMEP stations. Moreover, the vertical distribution and the medium range transport of sea salt particles is described and discussed in detail, which is one of the highlight topics of this manuscript. A comparison of modeled columnar particle concentrations with measurements – e.g. via AOD data – would be of great value for this manuscript. The authors employed a coupled meteorology chemistry transport model in this study, which is another highlight. Although a comparison with results obtained via an uncoupled model system would be very interesting, it would be too time consuming to perform such model runs for this study (maybe the authors could keep this in mind for future studies). However, the authors could highlight the advantages of a coupled model setup in the beginning of the manuscript. The impact of sea salt particles on atmospheric nitrate mass concentrations is analyzed in the end of the manuscript. The presented and discussed results are not new and could be enhanced or removed (see text to questions 13).

The Figures included in the manuscript as well as in the supplement are of good quality and present the results in a clear manner. The text is easy to understand but has deficits in the scientific language and in the choice of suitable wording in some passages. Moreover, grammar errors or misspellings complicate the understanding of some long nested sentences. Therefore, a revision of the language is recommended.

**Response:**

*Many thanks to the reviewer for the comments and suggestions. We have improved the manuscript accordingly. The language in the manuscript has also been edited throughout.*

*Following the referee's comments, we have added the comparison between modelled AOD and AERONET measured AOD into the revised manuscript. Please find more details in the point-by-point response below. And thanks for the suggestion for the future studies; it would be an interesting topic to compare the results with an off-line model in the next step study.*

*The order of Figures was changed in the revised version manuscript. However, in this response we keep the order consistent (unless specified) with the original version manuscript for easily understood. The changes of the Figures order are shown in Table R1.*

*Table R1. The changing of Figures order in the revised manuscript*

| Original version | Revised version |
|---|---|
| Manuscript | |
| -- | Figure 1 (newly added) |
| Figure 1 | Figure 2 |
| Figure 2 | Figure 3 |
| Figure 3 | Figure 4 |
| Figure 4 | Figure 5 |
| -- | Figure 6 (newly added) |
| Figure 5 | Figure 7 |
| Figure 6 | Figure 8 |
| Figure 7 | Figure 9 |
| Figure 8 | Figure 10 |
| Figure 9 | Figure 11 |
| Supplement | |
| Figure S1 | Figure S1 |
| Figure S2 | Replaced by revised version Figure 1 |
| Figure S3 | Figure S2 |
| Figure S4 | Figure S3 |
| -- | Figure S4 (newly added) |
| Figure S5 | Figure S5 |

1. **Does the paper address relevant scientific questions within the scope of ACP?** Yes. The impact of sea salt particles on other atmospheric compounds and the vertical distribution of sea salt particles were evaluated. Both are topics relevant topics within the scope of ACP.

**Response:**

*Thanks for the comments.*

2. **Does the paper present novel concepts, ideas, tools, or data?** Yes. Most previous model studies on the atmospheric transport of sea salt particles and their impact on other atmospheric compounds were performed with uncoupled meteorology and chemistry transport models. In contrast this study is one of the first evaluating with topic by means of a coupled model system (WRF-Chem). Additionally, the authors evaluate the vertical distribution of sea salt particles. However, the evaluation of atmospheric sea salt concentrations against EMEP measurement data is not novel as well as the evaluation of the impact of sea salt on atmospheric nitrate. The discussion of the vertical sea salt concentration profiles would greatly benefit if measurement data on the column sea salt concentrations were additionally considered - e.g. AERONET AOD data.

**Response:**

*Thanks for the comments.*

*We agree that the previous researches also evaluated the atmospheric sea salt concentrations against EMEP measurement data. For example, as published in the latest research (Neumann et al., 2016a): "The comparisons with observational data (EMEP) show that sea salt concentrations are commonly overestimated at coastal stations and partly underestimated farther inland". However, it is worth to emphasis that the purpose of our study is to introduce a specific long-range transport mechanism of sea salt, resulting from the "aloft bridge" (see Fig. 9). As shown in the manuscript, the sea salt is overall overestimated at both coastal and farther inland stations (e.g., Melpitz). This transport mechanism is shown in a clearer way with the help of the evaluation of sea salt concentration against EMEP measurement data. Furthermore, built on the overestimated the sea salt emission, this transport mechanism further extends the influence of sea salt to a much larger inland region instead of being confined to the coastal region, which in addition leads to overestimation of nitrate concentration in the coarse mode aerosol as well.*

*The influence of sea salt on nitrate has been discussed in previous studies (Neumann et al., 2016a; Liu et al., 2015; Im, 2013; Athanasopoulou et al., 2008), but mainly focus on the bulk nitrate mass concentrations and did not shown the influence on the nitrate within different size mode (fine mode and coarse mode). In this study, we quantified the sea salt influence for both fine mode and coarse mode nitrate particle formation respectively. By looking into size-segregated details, we found that sea salt facilitates the coarse mode nitrate particle (NaNO₃) formation (as found in most previous studies), but it may inhibit the fine mode nitrate particle (NH₄NO₃) formation. This effect can change the particle mass size distribution (PMSD) of nitrate, moves nitrate from fine mode nitrate particles to coarse mode nitrate particles (see Fig. 9), which is crucial for aerosol deposition, hygroscopicity, and optical properties etc. Please also see General Comments Point-13 for more details.*

[Figure]

*Figure 9. Schematic of sea salt transportation and its influence on nitrate particle formation.*

*Following referee's suggestion, we have added the comparison with AERONET AOD data in the end of section 3.2. And the description of AERONET AOD data has also been added in section 2.3 accordingly. As shown below:*

*"The AERONET (AErosol RObotic NETwork, http://aeronet.gsfc.nasa.gov) dataset over Europe was utilized to validate the aerosol optical depth (AOD) simulation. The AERONET AOD was derived from Sun photometer measurements of the direct (collimated) solar radiation. The level 2.0 AOD data, with pre and post field calibrated, automatically cloud*

*cleared and manually inspected, were used in this study. The AOD at 500 nm wave-length and the Angstrom index are directly available in AERONET dataset, and AOD at 550 nm wave-length was derived. More detailed information is given in http://aeronet.gsfc.nasa.gov."*

*"The column accumulated aerosol property AOD was evaluated for R-CASE (Fig. 6a, newly added in the revised manuscript) and F-CASE (Fig. 6b) respectively. Since AERONET AOD data only can be measured during daytime under clear-sky condition, the corresponding simulation results were analyzed.*

*As shown in Fig. 6a, except the Modena station in Italy, in the R-CASE the spatial distribution of AOD over Europe can be captured by the model in general with correlation coefficient (R) value 0.64: the highest AOD value (about 0.15-0.3) over inland region, relatively high value (about 0.07-0.15) over coastal region of North Sea and Baltic Sea, relatively low value (about 0.03-0.1) over the Southern Europe, and extremely low value (about 0.01-0.03) over Alpine Mountain region. And the R-CASE result showed a moderate AOD range (about 0.05-0.12) over the North Sea, which was comparable with the Southern European region (e.g.: Italy and Greece). However, the R-CASE result overestimated the AOD in general with a geometric mean ratio (GMR) value 1.8, which could be resulting from the overestimation of nitrate particles. The nitrate particle mass concentrations in $PM_{10}$ were overestimated by a factor of ~5 in the R-CASE at Melpitz (see Table 2). Although some shortcomings can be identified, the overall performance of AOD simulation is satisfactory and in line with previous studies (e.g. Banzhaf et al., 2013;Li et al., 2013).*

*The spatial distribution of AOD was less matched between modelled result and AERONET AOD measurements in the F-CASE (Fig. 6b). The R value reduced to 0.56, with much higher overestimation of AOD and GMR increased to 2.3. The modelled AOD over the North Sea is significantly increased to an unreasonable value, which was comparable with the central Europe. This is because GO03 overestimated SSA emission over the North Sea. The detailed evaluation of SSA mass concentration is given in the next section 3.3."*

[Figure]

[Figure]

*Figure 6 (newly added in the revised manuscript). Comparisons of AOD at 550 nm wavelength between AERONET measurements and WRF-Chem results, averaged during September 10-20. The correlation coefficient (R) and geometric mean ratio (GMR) are shown in the figure. The plotted WRF-Chem AOD results are divided by 2, in order to show more details of AERONET AOD data within one color-bar. (a) R-CASE result; (b) F-CASE result.*

3. **Are substantial conclusions reached?** Yes, partly. The authors discussed on a quite detailed level why sea salt particles are transported to a measurement station in the hinterland. Further it is found that sea salt concentrations are overestimated by the model which a common result of recent model sea salt studies.

**Response:**

*Thanks for the comments.*

4. **Are the scientific methods and assumptions valid and clearly outlined?** Yes, the methods are clearly outlined. The reasoning for some assumptions in section 3.4 should be revised.

**Response:**

*Thanks for the comments. The section 3.4 has been revised accordingly. Please find the details about revision in the following General Comments Point-13.*

5. **Are the results sufficient to support the interpretations and conclusions?** Yes, the results and their representation in the manuscript are sufficient. The reproducibility would be facilitated if the plotted data were attached as supplement information (as text-CSV, netCDF or another appropriate format).

**Response:**

*Thanks for the comments. All the plotted data will be attached as supplement information.*

6. **Is the description of experiments and calculations sufficiently complete and precise to allow their reproduction by fellow scientists (traceability of results)?** Yes.

**Response:**

*Thanks for the comments.*

7. **Do the authors give proper credit to related work and clearly indicate their own new/original contribution?** Yes. In a few situations (see my comments below), additional references were appropriate.

**Response:**

*Thanks for the comments.*

8. **Does the title clearly reflect the contents of the paper?** Yes, in principle it does. The authors might consider to add Northwestern in front of Europe and to replace simulation by in a coupled meteorology and chemistry transport model (or in WRF-Chem). Since the usage of a coupled model is quite novel with respect to this topics, it is reasonable to add this information.

**Response:**

*Thanks for the suggestion. The title "Sea salt emission, transportation and influence on nitrate simulation: a case study in Europe" has been **revised as**:*

*"Sea salt emission, transport and influence on size-segregated nitrate simulation: a case study in Northwestern Europe by WRF-Chem"*

9. **Does the abstract provide a concise and complete summary?** Yes, it provides a concise and complete summary and is well written.

**Response:**

*Thanks for the comments.*

10. **Is the overall presentation well structured and clear?** The Data and Methods, Results and Discussions, and Conclusions sections are well structured. Some descriptions in the Results and Discussions section should be moved to the Data and Methods section as indicated by some of the comments below. The authors might consider to restructure the Introduction section. I am missing a clear "story line" in the latter section.

**Response:**

*Thanks for the comments. The descriptions of the F-CASE and the R-CASE have been moved to section 2 "Data & Method". The information about deliquescent aerosol particles, ionization equilibrium and the Kelvin effect has been moved to section 2 "Data & Method" (see Scientific Point-32). The definition of nitrate partitioning fraction has been moved to section 2 "Data & Method" (see Scientific Point-39). And the Introduction section has been revised accordingly. Please see the revised manuscript for more details.*

11. **Is the language fluent and precise?** The language is fluent but not completely clear throughout the text. In some situations the used expressions are rather colloquial than scientific. In the comments below, some of these expressions are listed. Although it should be noted that the colloquial expressions make the text easier and more fluent to read than with the correct scientific expression (and partly more lengthly formulations). The usage of the definite article "the" and of the indefinite article ("a" in singular; nothing in plural) is mixed in several passages. In the Introduction and section 3.4, some sentence structures are not clear – it is unclear weather spelling or grammar mistakes are the reason. A comma (",") has to be placed after "Thus", "Additionally" and similar words, which start sentences. I suggest to revise the manuscript with a focus on these points.

**Response:**

*Thank you very much for the comments. The language in the manuscript has been edited throughout accordingly.*

12. **Are mathematical formulae, symbols, abbreviations, and units correctly defined**

**and used?** Yes. The concentrations of substances are written as $[X]$ and $X$ (where "$X$" is the substance's chemical formula). The writing should be either $[X]$ or $X$ - not mixed.

**Response:**

*Thank you very much for the comments. The notations of concentrations in the manuscript have been revised throughout accordingly. Please see details in Scientific Comments Point-41.*

13. **Should any parts of the paper (text, formulae, figures, tables) be clarified, reduced, combined, or eliminated?** In the current version, section 3.4 does not present new and/or

unexpected results. It should be enhanced by new considerations (e.g. the impact of sea salt particles on the vertical distribution of nitrate) or removed.

**Response:**

*Thank you very much for the comments. As pointed out by reviewer in Scientific Comments Point-2, the sea salt can facilitate the nitrate particle formation by providing surface area. We agree with reviewer that impact of sea salt on bulk nitrate particle formation is not novel. But, in this study: (1) we not only evaluated the sea salt influence on bulk nitrate particle mass concentration, but also for the fine mode and coarse mode nitrate particles respectively; (2) we not only took the contribution of surface area by SSA into consideration, but also the heterologous reaction on the sea salt (NaCl) surface with the production of thermodynamic stable NaNO$_3$ (NaCl + HNO$_3$ → NaNO$_3$ + HCl↑, also see the Scientific Comments Point-2). The fully coupled size-resolved sectional aerosol module (MOSAIC) gives us a chance to look into details of the size resolved sea salt influence on nitrate. As shown in section 3.4, the sea salt indeed facilitates the nitrate particle formation in coarse mode; however, it on the other hand inhibits the nitrate particle formation in fine mode indirectly. This effect can change the particle mass size distribution (PMSD) of nitrate, which is crucial for aerosol deposition, hygroscopicity, and optical properties etc. In section 3.4, we explained and quantified this effect in detail. We thus would like to keep this section. However, as pointed out by reviewer, section 3.4 did not clearly highlight out this scientific point. Therefore, in order to emphasize this scientific point, the title, section 3.4, section of Introduction and Figure 9 (please find the revised Fig. 9 in General Comments Point-2) have been revised. The detailed revisions are shown as following.*

*The title has been revised as suggested in the General Comment Point-8:*

[revised manuscript text omitted]

14. **Are the number and quality of references appropriate?** Yes. Additional references might be reasonable in some passages. These passages are listed in the comments below.

**Response:**

*Thanks for the comments and complement of previous researches.*

15. **Is the amount and quality of supplementary material appropriate?** Yes, the supplements adds value to the main manuscript. I would suggest to put more than one figure/table on each page in order to saves pages and avoid large white spaces. Page numbers would be favorable. Furthermore, the authors should verify whether reprint of the three plots in Fig. 4 violate Copyright laws. Also see the answer to point 5.

**Response:**

*Thanks for the comments and very careful consideration about the Copyright. The reprint permission of the three plots in Fig. S4 has been confirmed. The supplementary has been revised as suggested, and the plotted data will be attached as supplement information.*

**Scientific Comments:**

1. p.1, l.20: "... and has significant impact on the formation on secondary inorganic aerosol particles on global scale.". Reading this sentence might imply that the presence of sea salt particles favors the formation of sec. inorg. aerosols (SIA). However, this is not the case as the authors probably know. Sea salt has an indirect impact on particle formation because compounds such as $H_2SO_4$, $HNO_3$, and $HNO_3$, which tend to form new particles, condense onto sea salt particles surfaces instead. Hence, the particles formation is decreased. Please reformulate the sentence.

**Response:**

*Thanks for the comments. The corresponding sentence has been revised as:*

*"... and has significant impact on the formation of secondary inorganic particle mass on a global scale."*

2. p.2, l.8-9: In which context do sea salt particles **participate** in **heterogeneous reactions**? The salt particles provide surface area for heterogeneous reactions but using the work

"participate" might be misleading. Please also clarify the meaning of "leading to the formation of secondary aerosols" (see 1.).

**Response:**

*Thanks for the comments. Yes, sea salt particles could provide surface area for heterogeneous reactions, which would facilitate the nitrate/sulfate formation. In additional, NaCl is the main compound of sea salt. As known by reviewer and pointed out in the next Scientific Comments Point-4: NaCl would not only provide the surface area, but also participate the heterogeneous reaction with chlorine displacement (NaCl + HNO₃ → NaNO₃ + HCl↑). This produces thermodynamic stable sodium nitrate (NaNO₃), which is a key compound in this study, since it will not go back to the gas phase precursors as the semi-volatile ammonium nitrate does (NH₄NO₃ → NH₃ + HNO₃). Due to this reason, the fine mode ammonium nitrate particles keep moving to the coarse mode sodium nitrate particles when sea salt concentration is high (also see the above General Comments Point-13). Therefore, we prefer to use the word "participate".*

*The sentence "leading to the formation of secondary aerosols" has been revised as "leading to the formation of secondary aerosols on SSA surfaces".*

3. p.2, l.10: "... significant influence on **nitrate formation** ...". The deprotonation of an acid (HNO₃) should not be denoted as the **formation** of the deprotonated version of this acid (NO⁻₃). If nitrate was formed from different compounds via heterogeneous reactions at the particle surface, **nitrate formation** was appropriate. However, the latter situation is not the case, here. The usage of "formation" in connection with "nitrate" arises in further text passages, such as p.2, l.17. Please considered replacing "formation".

**Response:**

*Thanks for the comments and very clear explanation. The word "nitrate formation" has been replaced as "nitrate particle formation" in the corresponding context.*

4. p.2, l.11: "chlorine deficit": Commonly, it is denoted as "chlorine displacement".

**Response:**

*Thanks for the correction. The statement has been revised as suggested.*

5. p.2, l.13-15: Please give the reason for the difference in the cations bound to nitrate between central/western and northern/southern Europe. → extensive animal husbandry in central and western Europe associated with high ammonia emissions.

**Response:**

*Thanks for the suggestions. The sentence "The reason is the enhanced ammonia emission from husbandry and agricultural sources in central and western Europe (Backes et al., 2016b;Backes et al., 2016a)" has been added accordingly.*

6. p.2, l.16: "short life-time due to its quick deposition within the coastal region": The other way around: Coarse sea salt particles have a short life time and, therefore, they depose close their source. If the sea salt is emitted close to the coast then it also deposes in coastal regions.

**Response:**

*Thanks for the suggestion. The corresponding sentence has been revised as suggested.*

7. p.2, l.24: Please give a reference for the first sentence's statement (and "much more" is colloquial style).

**Response:**

*Thanks for the comment. The sentence has been revised and the citation has been added. As shown below:*

*"SSA contributes to the global aerosol burden multiple times more than the anthropogenic aerosol (Grythe et al., 2014)."*

8. p.2, l.25-26: "The parameterization schema . . . ": consider introducing an abbreviation for the schema, such as GO03 as common in the literature.

**Response:**

*Thanks for the suggestion. The abbreviation GO03 has been used throughout the manuscript to represent the sea salt emission schema (Gong, 2003).*

9. p.2, l.35: Consider starting a new paragraph, here.

**Response:**

*Thanks for the suggestion. The corresponding sentence has been revised as suggested.*

10. p.2, l.36-37: "The uncertainty of the SSA emission scheme directly determines the uncertainty of the evaluation of SSE radiative forcing." This is partly correct, because the deposition – particularly variable dry deposition for variable sea salt particle size distributions – plays a relevant role.

**Response:**

*Thanks for the comments. We agree with reviewer, the sentence has been removed.*

11. p.2, l.37-39: Consider switching (and slightly reformulating) the two sentences starting with "Additionally" and "the heterogeneous".

**Response:**

*Thanks for the suggestion. The corresponding sentence has been revised as suggested. The sentence has been revised as:*

*"The heterogeneous reaction could amplify the uncertainty of total aerosol burden due to the influence of SSA on secondary aerosol formation (e.g., nitrate, Seinfeld, 2006), therefore, SSA has an indirect effect on the total aerosol burden."*

12. p.3, l.3: "nitrate simulation": reformulate; "nitrate prediction"?

**Response:**

*Thanks for the comment. The sentence "In section 3.4, the influence of SSA on the nitrate simulation was quantitatively analyzed." has been revised as:*

*"In section 3.4, the influence of SSA on the size-segregated nitrate particle prediction was quantitatively analyzed."*

13. p.3, l.6-9: Please consider to mention "MOSAIC" and "CBMZ" already in this first paragraph. The detailed explanation further below is fine.

**Response:**

*Thanks for the comment. The sentence "The gas-phase atmospheric chemistry was presented by the Carbon-Bond Mechanism version Z (CBMZ), which is coupled with the MOdel for Simulating Aerosol Interactions and Chemistry (MOSAIC, Zaveri et al., 2008)." has been moved from p.3, l.32-33 to the end of this first paragraph.*

14. p.3, l.6-7: Consider extending ". . . regional air quality model." to ". . . regional meteorology and air quality model system.".

**Response:**

*Thanks for the comment. The sentence has been revised as suggested.*

15. p.3, l.8-9: "In addition to meteorology, aerosols, trace gases and interactive processes . . . ": "meteorology" was not mentioned before but everything behind meteorology was indirectly mentioned by writing "air quality model". Please consider reformulating the sentence.

**Response:**

*Thanks for the comment. The sentence has been revised as suggested by Scientific Comments Point-14.*

16. p.3, l.10-11: Please clarify in the text that MOSAIC is the employed aerosol module in WRF-Chem and not an individual modeling system.

**Response:**

*Thanks for the comment. The sentence "In order to represent the properties of size-resolved aerosol particles, MOSAIC is utilized in this study." has been revised as:*

*"In order to represent the properties of size-resolved aerosol particles, the fully coupled sectional aerosol module MOSAIC was chosen in this study."*

17. p.3, l.13-16: Please state a first, why the bin is split (PM1 and PM1-10 calculation), and then, how it is done.

**Response:**

*Thanks for the comment. The sentence "The fine mode (PM$_1$) and coarse mode (PM$_{1-10}$) particle mass concentration can be derived from this eight size bins." has been added before the sentence "The size range of the fifth bin is from 625 nm to 1250 nm."*

18. p.3, l.18-19: "Both particle mass concentrations and particle number concentrations are simulated.". Question (I am not familiar with the sectional particle representation in MOSAIC): Should the particle number and mass concentrations not be related via the size range of the bin? If the number concentration of particles of a pre-defined size (e.g. 625 nm to 1250 nm in size bin 5) is known, then the particle volume concentrations (assuming uniform size distribution in this bin) and mass concentrations can be directly calculated. Why are number and mass individually modeled per bin (which could result indirectly in particle sizes outside of the bin's size range).

**Response:**

*It is a very good question. MOSAIC is a sectional aerosol module with 8 size bins in WRF-Chem, as shown in Fig. S2. And as described by the MOSAIC developer (Zaveri et al., 2008):*

*"MOSAIC is implemented in the sectional framework where the aerosol size distribution is divided into discrete size bins. The size bins are defined by their lower and upper dry particle diameters, so water uptake or loss does not transfer particles between bins. Furthermore, each bin is assumed to be internally mixed so that all particles within a bin have the same chemical composition, while particles in different bins are externally mixed.*

*Both mass and number are simulated for each bin. Particle growth or shrinkage resulting from the dynamic gas-particle partitioning of trace gases (H$_2$SO$_4$, CH$_3$SO$_3$H, HNO$_3$, HCl, NH$_3$, and eventually secondary organic species) is first calculated in a Lagrangian manner. Transfer of particles between bins is then calculated using the (default) two-moment approach of Simmel and Wurzler [2006] or the moving section approach of Jacobson [1997a]."*

*Therefore, the particle number and mass concentrations are not exactly one-by-one related.*

19. p.3, l.39 to p.4, l.1: Please consider to describe (a) the outer, the intermediate, and the inner domain (in this order) or (b) the inner, intermediate, and outer domain but not (c) the outer, the inner and then the intermediate domain.

**Response:**

*Thanks for the comment. The sentences have been revised as suggested. As shown below:*

*"The outer domain covers the whole Europe, part of the North Sea and the North Africa with a spatial resolution of 54 km, providing the boundary conditions for the inner domains. The intermediate domain (D02, Fig. 1) was centered at Melpitz, and covers part of the North Sea, the central and southern Europe with a spatial resolution of 18 km. The innermost domain was also centered at Melpitz, and had a spatial resolution of 6 km."*

20. p.4, l.2: Consider adding "time" or "period" after "spin-up".

**Response:**

*Thanks for the comment. The sentence has been revised as suggested. As shown below:*

*"The spin-up time of the model run was 2 days."*

21. p.4, l.3: Please add "NCEP" in front of "sea surface temperature". Were the FNL data used as meteorological boundary conditions for the outer model domain?

**Response:**

*Thanks for the comment. The "NCEP" has been added as suggested. Yes, as reviewed understood, FNL data was used as the meteorological boundary conditions and initial conditions for the outer model domain.*

22. p.4, l.7: Please update the url to MOZART (http://www.acom.ucar.edu/wrf-chem/ mozart.shtml).

**Response:**

*Thanks for the comment. The URL has been updated as suggested.*

23. p.4, l.6: "The initial chemical and boundary conditions . . . ". Please switch the position of chemical": "The chemical initial and boundary conditions . . . ".

**Response:**

*Thanks for the comment. The sentence has been revised as suggested.*

24. p.4, l.9: Please introduce and describe the F-CASE and the R-CASE.

**Response:**

*Thanks for the comment. The introduction and description of the F-CASE and the R-CASE have been added at the end of the first paragraph of section "2.2 Emission". As shown below:*

*"In order to quantify the influence of SSA on the nitrate particles formation in this study, a sensitivity study was implemented with only 5% of SSA emission (R-CASE) and compared with the full (100%) SSA emission case (F-CASE)."*

25. p.4, l.10: "SSA results from dried sea spray . . . ": The use of "dried" could be misinterpreted by readers as "dry sea salt".

**Response:**

*Thanks for the comment. The sentence has been revised as suggested by the reviewer 2#:*

*"SSA is produced through the evaporation of sea sprays, which were ejected into the atmosphere from the sea surface."*

26. p.4, l.12: Neumann et al. (2016) is no primary reference for this statement.

**Response:**

*Thanks for the comment. The citation (Neumann et al., 2016a) have been removed here, and just keep the primary reference (Monahan et al., 1986) for this statement.*

27. p.4, l.20: A side note to the choice of the adjustable parameter: Gantt et al. (2015) (doi: 10.5194/gmd-8-3733-2015) suggests a value of 8 instead of 30.

**Response:**

*Thanks for the comment. The corresponding paragraph has been revised and one sentence has been added to note the choice of the adjustable parameter value from (Gantt et al., 2015). As shown below:*

*"Gong (2003) introduced an adjustable parameter to improve the results. He found the value 30 produced the best results. Therefore, this value was also used in WRF-Chem simulation; although Gantt et al. (2015) suggested that value 8 maybe better for the adjustable parameter in some conditions."*

28. p.5, l.18: "with a temporal resolution of 2 days." The formulation is misleading, because it might be understood by readers that two-day filter samples (48-hour averages) are collected at the Dutch stations. Actually, one-day filter samples (24-hour average) are collected every second day as the authors know and correctly plotted in Fig. 5. Therefore, please reformulate.

**Response:**

*Thanks for the comment. The corresponding sentence has been revised as suggested by the reviewer 2#. As shown below:*

*"Additionally, 24-hour filter sampler measurements with $PM_{10}$ inlets (EMEP, 2014) at 3 coastal EMEP station near the SSA transportation pathway (Bilthoven, Vredepeel, and Kollumerwaad, see Fig. 1), which were collected every second day, were obtained from EBAS (http://ebas.nilu.no/)."*

29. p.6, l.1: "be unrealistic sources" → "be unrealistic high sources"?

**Response:**

*Thanks for the comment. The corresponding sentence has been revised as suggested.*

30. p.6, l.22-25: The cited studies do not explicitly focus on Northwestern Europe. Tsyro et al. (2011) (doi: 10.5194/acp-11-10367-2011) presented an extensive model study on sea salt

concentrations in Europe spanning several years. Manders et al. (2010) (doi: 10.1016/j.atmosenv.2010.03.028) also compared sodium model and measurement data at several EMEP stations. Both found overestimations. Neumann et al. (2016) found overestimations in winter. The authors might consider the refer to these studies, because they focus on a similar region as this manuscript does.

**Response:**

*Thanks for the comment. The corresponding sentences have been revised as suggested, as described in following.*

*"The overestimation was consistent with previous modeling studies using WRF-Chem: in the Southeast Pacific ocean (Saide et al., 2012), in the coast region of California USA (Saide et al., 2013), over Europe (Nordmann et al., 2014; Zhang et al., 2013) and over the cold waters of the Southern, North Pacific and North Atlantic Oceans (Jaeglé et al., 2011)."* **changed to:**

*"The overestimation is consistent with previous modeling studies using WRF-Chem: in the Southeast Pacific ocean (Saide et al., 2012), in the coast region of California USA (Saide et al., 2013), over Europe (Nordmann et al., 2014;Zhang et al., 2013;Tsyro et al., 2011;Manders et al., 2010) and over the cold waters of the Southern, North Pacific and North Atlantic Oceans (Jaeglé et al., 2011). Similarly, (Neumann et al., 2016b) found overestimations over Europe during winter and attributed the reason to the missing of SST influence in GO03."*

31. p.7, l.3: ". . .SSA was emitted near the surface layer . . . ": Sea salt particles should be emitted into the surface apart from the situation, in which sea salt is emitted from the top of a giant wave higher than the model surface layer. However, in the latter situation we probably need another emission parameterization and should not use 10-m wind data.

**Response:**

*Thanks for the comment. The sentences "SSA was emitted near the surface layer of North Sea and lifted upward by convective mixing and turbulence." has been **revised as**:*

*"SSA was emitted into the surface layer of the North Sea and lifted upward by convective mixing and turbulence."*

32. p.7, l.11-22: The authors could consider to describe some information of this paragraph in the Data and Methods section.

**Response:**

*Thanks for the comment. The information (as shown below) about deliquescent aerosol particles, ionization equilibrium and the Kelvin effect has been moved to the Data and Method section.*

*"The heterogeneous reaction of nitric acid on SSA surface with the production of sodium nitrate is considered in MOSAIC. For the deliquescent aerosol particles at high RH, the ionization equilibrium and the Kelvin Effect are also taken into consideration in MOSAIC. More detailed descriptions are given in Zaveri et al. (2008)."*

33. p.7, l.13: ". . . promoting the formation of secondary inorganic aerosol (SIA), . . . ": Secondary aerosols or secondary particles denotes the new formation of particles in the atmosphere. The presence sea salt particles enhances the $HNO_3/NO_3^-$ condensation and, hence, one could reformulate the sentence into ". . . promoting the formation of secondary inorganic particle mass . . . ".

**Response:**

*Thanks for the comment and clear explanation. The corresponding sentence has been revised as suggested.*

34. p.7, l.15: "Part of $HNO_3$ will participate in the partitioning process and form particulate nitrate." Misleading. The whole $HNO_3$ is involved into the partitioning process. One part remains in the atmosphere and the other part condenses. The condensed part becomes nitrate. ! Suggestion: "$HNO_3$ undergoes (maybe another word) a partitioning process between gas phase and liquid particle phase via condensation. The condensed $HNO_3$ deprotonates to $NO_3^-$"

**Response:**

*Thanks for the comment and clear explanation. The corresponding sentence has been revised as suggested. As shown below:*

*"HNO₃ undergoes a partitioning process between gas phase and liquid particle phase via condensation. The condensed HNO₃ deprotonates to NO₃⁻"*

35. p.7, l.18: "irreversible reaction", better "irreversible process"

**Response:**

*Thanks for the comment. The terminology has been revised as suggested.*

36. p.7, l.35-36: Why are 5% of the original sea salt emissions emitted in the R-CASE and not 20%, 10%, or 1%? Is this value arbitrarily set or is it related to the 20-fold overestimation of sodium PM10 by the model (1/20 = 5%)?

**Response:**

*Thanks for the comment. 5% of original sea salt emission was chosen for the sensitivity study, because of the 20-fold overestimation of sodium $PM_{10}$ by the model. And as explained in p.8 l.13-15 (original version of manuscript):*

*"In this study, the R-CASE had a much more reasonable SSA prediction (within a factor of ~1 at Melpitz) than the F-CASE; therefore, the simulated values of PF_nitrate from the R-CASE were used as the reference."*

*So, based on the comparison between the F-CASE and the R-CASE, we can quantify the influence on nitrate particle formation in a more precise way. And this influence is mainly due to the overestimation of SSA.*

37. p.7, l.39: "simulation" → "prediction"? (see comment 12.)

**Response:**

*Thanks for the comment. The terminology has been revised as suggested.*

38. p.7, l.43-45: Consider switching the order of both sentences.

**Response:**

*Thanks for the comment. The sentence has also been revised as suggested. As shown below:*

*"Therefore, the difference of nitrate between the F-CASE and the R-CASE should mainly arise from the influence of the SSA concentrations. The overestimation factor of nitrate at Melpitz dramatically decreased from 2.1 to 0.73 in coarse mode (PM$_{1-10}$, see Table 2)."*
***changed to:***

*"The factor for nitrate between model and measurement at Melpitz dramatically decreased from 2.1 to 0.73 in coarse mode (PM$_{1-10}$, see Table 2), thus changing from overestimation to underestimation. Therefore, the difference of nitrate between the F-CASE and the R-CASE should mainly arise from the influence of the SSA concentrations."*

39. p.8, l.1-7: Data and Methods section?

**Response:**

*Thanks for the comment. The corresponding part has been moved to Data and Method section as suggested. As shown below:*

*"2.4 Nitrate partitioning fraction*

*The participation of SSA changes the partitioning processes of nitrate. In order to quantify this effect, the nitrate partitioning fraction (PF_nitrate) was analyzed for coarse mode (PM$_{1-10}$) and fine mode (PM$_1$) particles. The definition of PF_nitrate for coarse/fine mode is shown in Eq.1.*

$$PF\_nitrate_{coarse/fine} = \frac{[NO_3^-]_{coarse/fine}}{[NO_3^-]_{coarse/fine} + [HNO_3]} \tag{1}$$

*where [NO$_3^-$]coarse/fine is the coarse/fine mode particulate nitrate mass concentration, [HNO$_3$] is the nitric acid mass concentration. "*

40. p.8, l.8: "probability density function": "frequency distribution" might be more appropriate (also at the subsequent occurrences in the paragraph)

**Response:**

*Thanks for the comment. We have modified it as suggested. The "probability density function" has been replaced by "frequency distribution", the patterns are the same between them two. As shown below:*

[Figure]

***Figure 8 (revised).*** *WRF-Chem results of the frequency distribution of PF_nitrate at Melpitz. The result was analyzed during the marine period ($[Na^+] > 1.8$ μg/m$^3$ in the F-CASE). The dash lines (coarse mode: red; fine mode: blue) indicate the median value (with 50% probability in both sides). (a) PM$_{1-10}$ result of 5% SSA emission (R-CASE); (b) PM$_{1-10}$ result of the F-CASE; (c) PM$_1$ result of the R-CASE; (d) PM$_1$ result of the F-CASE.*

[Figure]

*Figure 8 (original). WRF-Chem results of the probability density function of nitrate partitioning fraction (PF_nitrate) at Melpitz in the marine period during September 10-20, 2013. The marine period is defined as the $Na^+$ mass concentration higher than 1.8 μg/m³ in F-CASE. The blue dash lines indicate the median value (with 50% probability in both sides). (a) $PM_{1-10}$ result of 5% SSA emission (R-CASE); (b) $PM_{1-10}$ result of F-CASE; (c) $PM_1$ result of R-CASE; (d) $PM_1$ result of F-CASE.*

41. p.8, l.9-10: Please clarify "marine period ($Na^+ > 1.8$ ug/m³ in F-CASE).". It means that only PF_nitrate values of model time steps with $[Na^+] > 1.8$ ug/m³ were considered? How many model time steps were considered? Please be consistent with the notation of concentrations: Above, $[HNO_3]$ denotes the concentration of $HNO_3$ but, here, $Na^+$ (and not $[Na^+]$) denotes the concentration of $Na^+$.

**Response:**

*Thanks for the comment. There are 105 time steps with $[Na^+] > 1.8$ ug/m³ in total. This information has been added in the manuscript. And the notations of concentrations have been revised throughout the manuscript. As shown below:*

*"during the marine period ($[Na^+]>1.8$ μg/m³ in the F-CASE, 105 time steps in total)."*

42. p.8, l.12-13: "also there was uncertainty of the precursors emissions in the model." There are always uncertainties in the emissions. Therefore, the uncertainty is not a general reason not to compare model and measurement data. Note: It should be "precursor's emissions" or "precursor emissions".

**Response:**

*Thanks for the comments. Reviewer is correct, the uncertainties in measurements and in the model emissions always exist, we need to keep those in mind when comparing measurements with model results, but they are not the reason why the two should not be compared. The sentences have been revised as suggested by the reviewer 2#. And some references have been added in this sentence, as shown below.*

*"Since MARGA measurements were only available for the size range of $PM_{10}$, and high uncertainty of the $HNO_3$ measurement due to its sticky property; also there was uncertainty of the precursors emissions in the model."* **changed to**

*"Since the MARGA measurements were only available for the size range of $PM_{10}$, PF_nitrate derived from MARGA observations should not be directly compared with the simulated one. Additionally, we need to keep in mind that high uncertainties exist in the $HNO_3$ measurements due to its sticky property (Rumsey et al., 2014;Neuman et al., 1999), which brings further difficulty into the comparison between measurements and simulation."*

*And it should be "precursor emissions" as used in Fry et al., (2012).*

43. p.8, l.15: "were used as reference": A reference for what?

**Response:**

*Thanks for the comment. The word "reference" has been replaced by "basic simulation". So, the sentence has been revised as:*

*"In this study, the R-CASE had a much more reasonable SSA prediction (within a factor of ~1 at Melpitz) than the F-CASE; therefore, the simulated values of PF_nitrate from R-CASE were used as the reference."* **changed to:**

*"In this study, the R-CASE had a much more reasonable SSA prediction (within a factor of ~1 at Melpitz) than the F-CASE; therefore, the simulated values of PF_nitrate from the R-CASE were used as the basic simulation."*

44. p.8, l.15-16: " Considering that most of the SSA was emitted as coarse mode particles (about 88% in both file measurement and simulation on September 17, 2013 at Melpitz), . . . ": The formulation is misleading. There are no emissions measured at Melpitz but concentrations. Since coarse particles have a higher dry deposition velocity than fine particle one can expect that the emissions consisted by more than 88% of coarse particles.

**Response:**

*Thanks for the comment. The reviewer is correct. The sentence has been reformulated as"*

*"Considering that most of the SSA was emitted as coarse mode particles (about 88% in both filter measurement and simulation on September 17, 2013 at Melpitz)"* **changed to:**

*"In September 17 at Melpitz, about 88% SSA mass was concentrated in the coarse mode particles in both simulation and filter measurement results. Since coarse particles have a higher dry deposition velocity than fine particle one, we can expect that the SSA emissions consisted by more than 88% of coarse particles."*

45. p.8, l.17-18: ". . . also more sensitive to the change of the SSA emission.": Why? The particle surface area is the parameter governing the condensation of nitric acid. Higher mass emissions cannot be directly related to higher particle surface area emissions.

**Response:**

*Thanks for the comment. As known by reviewer and pointed out in the next Scientific Comments Point-4: SSA (NaCl) would not only provide the surface area, but also participate the heterogeneous reaction with chlorine displacement (NaCl + HNO$_3$ → NaNO$_3$ + HCl↑). This produced thermodynamic stable sodium nitrate (NaNO$_3$) is a key compound in this study, since it will not go back to the gas phase precursors as the semi-volatile ammonium nitrate does (NH$_4$NO$_3$ → NH$_3$ + HNO$_3$). Therefore, SSA would not only provide a physical pathway (surface area) but also a chemical pathway for nitrate particle formation. In the chemical pathway, the SSA mass concentration counts. In this study, we focused more on this heterogeneous reaction. Please also see details in the above General Comments Point-13.*

46. p.8, l.20-21: ". . . median . . .was about 0.75 . . . broad spreading . . . ": There should be something like ". . . and distribution " between the "0.75" and "broad spreading".

**Response:**

*Thanks for the comment. The sentence has been revised as suggested by the reviewer 2#:*

*"As shown in Fig. 8a and Fig. 8b, the median value of coarse mode PF_nitrate was about 0.75 in R-CASE broadly spreading from ~0.2 to 1, and it increased to 0.96 in F-CASE with a much narrowed distribution." changed to:*

*"As shown in Fig. 8a and Fig. 8b, the median value of coarse mode PF_nitrate in the R-CASE was about 0.75, with the distribution broadly spread in the range of ~0.2 to 1; whereas in the F-CASE the median value increased to 0.96with a much narrowed distribution."*

47. p.8, l.20-21: ". . . increased the coarse mode nitrate partitioning fraction by 0.2.": Unclear whether 0.2 is a difference or a quotient. It is the difference, but the formulation is ambiguous.

**Response:**

*Thanks for the comment. The sentence ". . . increased the coarse mode nitrate partitioning fraction by 0.2" has been revised as:*

*". . . increased the coarse mode nitrate partitioning fraction from 0.75 to 0.96"*

48. p.8, l.22-24: The conclusion is justified because the ammonium mass concentrations are quite similar in both cases - otherwise not. Therefore, it might be reasonable to repeat that information in this summarizing sentence. Please consider splitting this sentence into two sentences.

**Response:**

*Thanks for the comment. The sentence has been revised as suggested, as shown below:*

*"This indicated that the participation of SSA increased the coarse mode nitrate partitioning fraction by ~0.2, which contributed to about 140% overestimation of the coarse mode nitrate (Table 2). In this case study, SSA was highly overestimated by the model in F-CASE and the overestimated amount was transported to the surface layer at Melpitz, which means more SSA*

*participated in the nitrate partitioning process and formed stable sodium nitrate and accumulated in the coarse mode."* **changed to:**

*"In this study, the ammonium mass concentration was quite similar in both the R-CASE and the F-CASE; SSA was highly overestimated by the model in the F-CASE and the overestimated amount was transported to the surface layer at Melpitz; and the coarse mode nitrate partitioning fraction increased from 0.75 (R-CASE) to 0.96 (F-CASE). These indicated that the participation of SSA in the nitrate partitioning process facilitated the coarse mode nitrate particle formation, which accumulated as the thermodynamic stable sodium nitrate. About 140% overestimation of the coarse mode nitrate was resulted from this reason (Table 2)."*

49. p.8, l.25-26: ". . . indicating that in our case the overestimation of SSA emission is mainly in the coarse mode.": What is the reasoning for this conclusion? No comparison against measurement data was performed.

**Response:**

*Thanks for the comment. The reviewer is correct. It could just indicate that most of the modelled SSA mass concentrated in the coarse mode particles. And this information had been stated in the previous context. Therefore, this sentence had been removed here.*

50. p.8, l.28: "consumption of precursors": Why plural? Are there other precursors than HNO3 for particulate $NO_3^-$?

**Response:**

*Thanks for the comment. The word "precursors" has been revised to "precursor" here.*

51. p.8, l.29: "$PM_{10}$" → "fine PM" or "$PM_1$"?

**Response:**

*Thanks for the comment. We think "$PM_{10}$" is correct here. Here, we want to describe that overestimation of SSA facilitated the coarse mode ($PM_{1-10}$) nitrate particle formation; however it inhibited the fine mode ($PM_1$) nitrate formation. One facilitation and the other one*

*inhibition, they cancelled out each other. Therefore, the total nitrate ($PM_{10}$) was not changed so much. The sentence has been revised in order to describe this more clearly. As shown below:*

*"Therefore, the total nitrate mass concentrations in size range of $PM_{10}$ were similar between R-CASE and F-CASE (Table 2)." changed to*

*"Therefore nitrate particle mass moved from fine mode to coarse mode, the total nitrate mass concentrations in size range of $PM_{10}$ were similar between the R-CASE and the F-CASE (Table 2)."*

52. p.8, l.33; p.9, l.23: "overestimated by 0.2": see comment 46.

**Response:**

*Thanks for the comment. The "overestimated by ~0.2" has been revised to "increased from 0.75 to 0.96".*

53. p.8, l.35: "particle number distribution": There were no number concentrations considered in this study.

**Response:**

*Thanks for the comment. The reviewer is correct. "particle number size distribution" has been removed.*

54. p.8, l.39; p.9, l.20: "simulation": see comment 12.

**Response:**

*Thanks for the comment. The "simulation" has been revised to "prediction".*

55. p.9, l.3, 10, 20, 30: "uncertainty": Uncertainty describes instable deviations (in some situations values are overestimated and in other situations they are underestimated). Here, the parameterization clearly overestimates the emissions. Therefore, "uncertainty" is not necessarily the correct word.

**Response:**

*Thanks for the comment. The "uncertainty" has been revised to "overestimation".*

56. p.9, l.6: "The variations" → "The spatial variations"

**Response:**

*Thanks for the comment. The statement has been revised as suggested by Comments on Language and Spelling Point-31.*

57. p.9, l.8: ". . . the overestimation in . . . " → ". . . the overestimation of emissions in . . . "

**Response:**

*Thanks for the comment. The statement has been revised as suggested.*

58. p.9, l.20: "Fig. 9": Please do not include new Figures in the Conclusions. This Figure should be described in an earlier passage of the manuscript or removed. The first choice is favorable because the figure describes the transport of sea salt particles to Melpitz very well and clear.

**Response:**

*Thanks for the comment. The description of "Figure 9)" has been moved to the last paragraph of section 3.4. In order to emphasize the SSA's influence on nitrate particle formation in chemical way, Figure 9 has also been revised. Please see details in General Comments Point – 2 & 13.*

59. p.9, l.25: "gas-phase precursors", see comment 50.

**Response:**

*Thanks for the comment. The "precursors" has been revised to "precursor".*

60. p.9, l.31: "formation of secondary inorganic aerosol", see comment 33.

**Response:**

*Thanks for the comment. The "formation of secondary inorganic aerosol" has been revised to "formation of secondary inorganic particle mass"*

61. p.9, l.37-38: Last sentence: If the authors want to write about NOX it should be done in an extra paragraph of the Conclusions section and not in the last sentence. The second last sentence might be a nice last sentence.

**Response:**

*Thanks for the comment. The last sentence has been removed as suggested.*

**Comments on Figures and Tables:**

1. p.16, Table 2: The authors might consider to split the three columns into five (one column each for "Factor" and "R"). "Factor" should be explained in the caption.

**Response:**

*Thanks for the comment. The Table 2 has been revised as suggested.*

2. p.17, Table 3: see comment 61.

**Response:**

*Sorry, do you mean comment 1? Thanks for the comment and the Table 3 has also been revised as reviewer suggested.*

3. p.18, Fig. 1: "shown in Figure 5": it is Figure 6; "domain 02" == intermediate domain?

**Response:**

*Thanks for the comment and sorry for the typo. The figure number has been corrected. Yes, domain 02 is "intermediate domain", and this information has been added accordingly.*

4. p.20 and 21, Fig. 3 and 4: The title above the plots is inconsistently written: "10-20$^{th}$ Sep. 2013, Melpitz" (Fig. 3) and "10-20th September, 2013".

**Response:**

*Thanks for the comment. All the plots titles have been written as "September 10-20$^{th}$, 2013".*

5. p.22, Fig. 5: The fact that two y-axes exist per plot should be noted in the caption. The authors could consider to order the plots by the stations' distance to the coast or geographic location (== Melpitz as plot (d)) because it is more intuitive with respect to Fig. 1.

**Response:**

*Thanks for the comment. The Fig. 5 has been revised as suggested.*

6. p.23, Fig. 6: The authors might consider inverting the x-axis because it is more intuitive for the reader to have the coast on the left and Melpitz on the right.

**Response:**

*Thanks for the comment. The Fig. 6 has been revised as suggested.*

7. p.24, Fig. 7: The length of the x-axis could be cropped.

**Response:**

*Thanks for the comment. The Fig. 7 has been revised as suggested.*

8. p.25, Fig. 8: The caption is complicated formulated. Consider reformulating it. Additionally: "probability" (1$^{st}$ and 4$^{th}$ line), see general comment 40.

**Response:**

*Thanks for the comment. Figure 8 and the caption have been revised. Please find the revised Figure 8 in the Scientific Comment Point-40.*

9. p.26, Fig. 9: see comment 6 to Fig. 6 and general comment 58.

**Response:**

*Thanks for the comment. The Figure 9 has been revised as suggested. Please find it in General Comments Point-2.*

**Comments on Language and Spelling:**

1. p.1, l.33-36: Please split this sentence into two.

**Response:**

*Thanks for the comment. The sentence has been revised as shown below:*

*"As a result, nitrate partitioning fraction (ratio between particulate nitrate and the summation of particulate nitrate and gas-phase nitric acid) increased by about 0.2 for the coarse mode nitrate due to the overestimation of SSA at Melpitz, but no significant difference in the partitioning fraction for the fine mode nitrate." **changed to***

*"As a result, nitrate partitioning fraction (ratio between particulate nitrate and the summation of particulate nitrate and gas-phase nitric acid) increased by about 0.2 for the coarse mode nitrate due to the overestimation of SSA at Melpitz. However, no significant difference in the partitioning fraction for the fine mode nitrate was found."*

2. p.1, l.41: Please change "Atmospheric aerosol plays" to "Atmospheric aerosols play" to be consistent with the next sentence ("Further **they** ...").

**Response:**

*Thanks for the comment. The sentence has been revised as suggested.*

3. p.1, l.42: Please change "could either be" from conjunctive to indicative ("are either").

**Response:**

*Thanks for the comment. The sentence has been revised as suggested.*

4. p.1, l.43: "constitute" → "constituent".

**Response:**

*Thanks for the comment. The word has been revised as suggested.*

5. p.2, l.3-6: Please reformulate the sentence. There are some typos or the grammar is incorrect.

**Response:**

*Thanks for the comment. The sentence has been revised as suggested by reviewer 2#:*

*"Waves breaking in the surf zone, where has more whitecaps and stronger emission due to increasing ocean bottom and higher intensity of wave breaking, may affect areas at a distance of 25 km from the coastline and can dominate the coastal region"* **changed to**

*"Waves breaking in the surf zone, where there are more whitecaps and stronger SSA emission due to increased ocean bottom and higher intensity of wave breaking, may affect SSA concentrations at areas within 25 km distance from the coastline and can dominate the SSA concentration at the coastal region".*

6. p.2, l.7: no conjunctive, see 3.

**Response:**

*Thanks for the comment. One sentence has been added at the very beginning of this paragraph, in order to make a conjunctive with previous context. As shown below:*

*"SSA exerts an influence on the mass concentration of other aerosols, which makes the intensity of SSA emission even more important."*

7. p.2, l.13: "... sodium nitrate is largely contributed to nitrates ... ": Please use active voice ("... sodium nitrate contributes ... ").

**Response:**

*Thanks for the comment. The sentence has been revised as suggested by reviewer 2#:*

*"sodium nitrate largely contributes to nitrates in northern and southern Europe"*

8. p.2, l.16: "quick" → wording

**Response:**

*Thanks for the comment. The sentence has been reformulated as shown below:*

*"Usually, SSA has a short life-time due to its quick deposition within the coastal region (Grythe et al., 2014), thus its influence on nitrate formation cannot reach the distant inland area."* **changed to**

*"Coarse sea salt particles have a short life-time (Grythe et al., 2014), usually depositing close to their source. SSA emitted near the shore will therefore deposit mainly in coastal regions. Its influence on nitrate particle formation is thus expected to be of less importance over Central Europe, where nitrate concentrations are high due to land-based sources (Xu et al., 2012)."*

9. p.2, l.16: ". . . region (Grythe et al., 2014), thus ... " → ". . . region (Grythe et al., 2014). Thus, . . . " (split sentence; comma after "thus").

**Response:**

*Thanks for the comment. The sentence has been revised as suggested. As shown in the Comments on Language and Spelling Point-8.*

10. p.2, l.17: "cannot reach the distant inland area." the meaning is clear but colloquial language; Why "the distant inland area."?

**Response:**

*Thanks for the comment. The sentence has been revised. As shown in the Comments on Language and Spelling Point-8.*

11. p.2, l.20-21: Please move "later on" to the end of the sentence because it specifies a time.

**Response:**

*Thanks for the comment. The word has been revised as suggested.*

12. p.2, l.22-23: ". . . provide an opportunity . . ." → colloquial. If the guards in a prison do not look after the prisoners, then they provide an opportunity for a prison break. However, the mechanisms do not provide an opportunity for sea salt particles. Also colloquial: ". . . make their influence more extensive . . .".

**Response:**

*Thanks for the comment. The sentence has been revised as suggested by the reviewer 2#. As shown below:*

*"These mechanisms provide an opportunity for SSA to be transported inland, and thereby make their influence more extensive, from coastal to regional or even global." **changed to***

*"These mechanisms facilitate the long-range transportation of SSA, and thereby expand their influences from coast to a broader region."*

13. p.2, l.26: ". . . is still highly uncertain . . .": please reformulate

**Response:**

*Thanks for the comment. The sentence has been revised. As shown below:*

*"The parameterization scheme (Gong, 2003; Monahan et al., 1986, i.e.: GO03) of SSA emissions in WRF-Chem is still highly uncertain (Grythe et al., 2014; Neumann et al., 2016)." **changed to***

*"There is still high uncertainty (Grythe et al., 2014; Neumann et al., 2016a; Neumann et al., 2016b) in the parameterization scheme (Gong, 2003; Monahan et al., 1986, i.e.: GO03) of SSA emissions in WRF-Chem."*

14. p.2, l.34: ". . . in varying degrees . . .": possibly ". . . by varying degrees. . . " might be correct; please check

**Response:**

*Thanks for the comment. We have checked that "in varying degrees" or "to varying degrees" both are correct, but no "by varying degrees".*

15. p.2, l.39: ". . . needs the participation of . . ." → colloquial

**Response:**

*Thanks for the comment. The sentence has been revised. As shown below:*

*"Such indirect effect needs the participation of gaseous pollutants such as nitrogen oxides (NOx), which are not only abundant along the coast area (Fig. S1)." **changed to***

*"The participation of gaseous pollutants (e.g.: NOx) is needed for this indirect effect. However, NOx is not only abundant along the coast area, but also some inland regions (Fig. S1)."*

16. p.2, l.41-42: ". . . make the importance of SSA indirect effect on nitrate formation over a broader region.". Please reformulate.

**Response:**

*Thanks for the comment. The sentence has been revised as suggested by the reviewer 2#.*

*". . . make the importance of SSA indirect effect on nitrate formation over a broader region." **changed to***

*"...extends the impact of SSA indirect effect on nitrate particle formation to a broader region."*

17. p.2, l.43-44: "In this study . . . by a case study . . . ". Please remove the duplication of "study".

**Response:**

*Thanks for the comment. The words "by a case study" have been removed.*

18. p.3, l.21-22: Please reformulate the sentence starting with "However,".

**Response:**

*Thanks for the comment. The sentence has been reformulated as shown below.*

*"However, the transfer of particles between the bins could results from the growth or shrink of particles due to chemical processes (e.g., uptake or release of trace gases, etc.) and physical processes (e.g., coagulation, etc., Chapman et al., 2009)."* **changed to**

*"However, particle growth or shrink due to chemical processes (e.g., chemical reaction, uptake/release of trace gases, etc.) and/or physical processes (e.g., coagulation, etc.) will result in the transfer of particles between the bins (Chapman et al., 2009)."*

19. p.3, l.23: "The formation mechanism of . . . ". Please change to "The formation of . . . " or "A formation mechanism of . . . ".

**Response:**

*Thanks for the comment. "The formation mechanism of . . ." has been revised to "The formation of ...."*

20. p.3, l.38-39: "from September 10-20, 2013". Please change to "from September 10 to 20, 2013" or "in the time period September 10-20, 2013" or choose another formulation.

**Response:**

*Thanks for the comment. "from September 10-20, 2013" has been changed to "from September 10 to 20, 2013".*

21. p.3, l.40: ". . . covers **the** whole Europe, part of the North Sea and the North Africa . . . ". Please remove the two bold "the" and add an "a" in front of "part" and please do the same in the succeeding lines.

**Response:**

*Thanks for the comment. The sentence has been revised as suggested.*

22. p.4, l.11: ". . . emitted from bubble bursting or breaking waves torn by winds at wave crests." → ". . . emitted by bubble bursting or breaking waves or torn of wave crests by winds."

**Response:**

*Thanks for the comment. The sentence has been revised as suggested.*

23. p.4, l.16: "was" →"is"

**Response:**

*Thanks for the comment. The verb has been revised as suggested.*

24. p.4, l.22: "temporal" →"temporally"

**Response:**

*Thanks for the comment. The word has been revised as suggested.*

25. p.4, l.24: move "code" behind the bracket ("(SNAP) code")

**Response:**

*Thanks for the comment. The word has been revised as suggested.*

26. p.4, l.26: insert "the" in front of "anthropogenic emission inventory"

**Response:**

*Thanks for the comment. The sentence has been revised as suggested.*

27. p.4, l.31: "consists with" → "has"

**Response:**

*Thanks for the comment. The verb has been revised as suggested.*

28. p.5, l.1-2: "the stations all over . . . vertical structures." Please reformulate.

**Response:**

*Thanks for the comment. The sentence has been revised as shown below.*

*"In addition, radiosonde datasets (http://www.weather.uwyo.edu/upperair/sounding.html) of Melpitz and the stations all over Europe were utilized to evaluate the modelled atmospheric vertical structures." changed to*

*"In addition, the modelled atmospheric vertical thermodynamic structures were validated by the radiosonde measurements all over Europe (http://www.weather.uwyo.edu/upperair/sounding.html)."*

29. p.5, l.42: "bin 05-08" → "bins 05-08". Occurs several times.

**Response:**

*Thanks for the comment. The words have been corrected as suggested.*

30. p.6, l.10-11: Please reformulate the sentence.

**Response:**

*Thanks for the comment. The sentence has been revised. As shown below:*

*"For the sea salt event studied here, the abrupt SSA event was found to be emitted over the North Sea and overall overestimated in the coastal and continental regions during September 16-20, as shown in Fig. 5." changed to*

*"For the sea salt event studied here, the abrupt SSA emission event was found to happen over the North Sea. And SSA mass concentration was overall overestimated in the coastal and continental regions during September 16-20, as shown in Fig. 5."*

31. p.6, l.17: "variance/trend": Maybe "temporal pattern"?

**Response:**

*Thanks for the comment. The statement has been revised as suggested.*

32. p.6, l.36: "400 km away from coast" → "400 km distant to the coast"?

**Response:**

*Thanks for the comment. The statement has been revised as suggested.*

33. p.6, l.39-40: ". . . about 30-40% of SSA mass concentration was actually transported to the inland (Melpitz) comparing to the coast regions." → "about 30-40% of the initial SSA mass at coastal stations was actually transported to the inland station of Melpitz.".

**Response:**

*Thanks for the comment. The statement has been revised as suggested.*

34. p.6, l.41: "will be discussed": Unclear; Where? When?

**Response:**

*Thanks for the comment. The "will be discussed" has been revised to "will be discussed in following".*

35. p.6, l.43-44: ". . . the warmer sea surface resulted in a higher planetary boundary layer (. . . ) than that over the continent." → ". . . the warmer sea surface resulted in a higher planetary boundary layer (. . . ) above the sea than over the continent."

**Response:**

*Thanks for the comment. The statement has been revised as suggested.*

36. p.7, l.4: ". . .was able to penetrate . . . " → ". . . penetrated . . . ".

**Response:**

*Thanks for the comment. The statement has been revised as suggested.*

37. p.7, l.11: "from" → "by"

**Response:**

*Thanks for the comment. The preposition has been revised as suggested.*

38. p.7, l.12-14: First part of the sentence unclear. Please reformulate.

**Response:**

*Thanks for the comment. The sentence has been revised as shown below.*

*"However, its influence on the simulation is not only on the primary SSA, but also on promoting the formation of secondary inorganic aerosol (SIA), such as nitrate (Neumann et al., 2016; Seinfeld, 2006)." **changed to***

*"However, its influence is not only on the aerosol burden of SSA itself, but also on promoting the formation of secondary inorganic particle mass, such as nitrate (Neumann et al., 2016a; Seinfeld, 2006)."*

39. p.7, l.20: "The participation of SSA . . . " → "The presence of SSA . . . "

**Response:**

*Thanks for the comment. The statement has been revised as suggested.*

40. p.7, l.25: "Either it could result from inaccurate emission of precursors or an improper chemical pathway." → "The overestimation could result either from inaccurate emissions of precursors or from an improper modeled chemical pathway." (suggestion)

**Response:**

*Thanks for the comment. The statement has been revised as suggested.*

41. p.7, l.26-34: Please reformulate the passage. One can interpret what is meant in this passage but the formulation and sentence structure make the understanding difficult.

**Response:**

*Thanks for the comment. The passage has been revised as shown below.*

*"The difference between Fig. 7a and Fig. 7b indicates that in addition to an overestimation caused by overestimated $NH_3$ emission (see also Table 2), improper chemical pathway also contributed to the nitrate overestimation. Since the simulated nitrate mass concentrations*

*(Fig. 7a) were still much higher than the observed one (Fig. 7b), even though where had the same mass concentrations of precursors."* **changed to**

*"However, even with the same mass concentrations of precursors, the simulated nitrate mass concentrations (Fig. 7a) were still significantly higher than the observed ones (Fig. 7b). This indicated that in addition to an overestimation caused by overestimated $NH_3$ (see also Table 2), improper chemical pathway in the model also contributed to the nitrate overestimation."*

42. p.7, l.35: "a sensitive study" → "a sensitivity study"

**Response:**

*Thanks for the correction. The word has been revised as suggested.*

44. p.9, l.11: "continent" → "the continental"

**Response:**

*Thanks for the comment. The statement has been revised as suggested.*

45. p.9, l.12-15: Split into to sentences and replace "participate" by another verb.

**Response:**

*Thanks for the comment. The statement has been revised as suggested.*

46. p.9, l.17-18: "made the SSA overestimated by a factor of 20 at Melpitz": Consider replacing "made" by "led" or "yielded" and reformulate.

**Response:**

*Thanks for the comment. The word "made" has been replaced by "led".*

47. p.9, l.20-22: Please reformulate.

**Response:**

*Thanks for the comment. The sentence has been revised as shown below.*

*"As described in Fig. 9, more nitrate and precursors can be locked in the sodium nitrate which is thermodynamically stable, due to the participation of more SSA in the nitrate partitioning process."* **changed to**

*"As described in Fig. 9, nitrate and precursors can be locked in the particulate phase by the thermodynamically stable sodium nitrate, which is produced from the heterogeneous reaction on SSA surface."*

48. p.9, l.26-27: ". . . , resulted from coarse mode nitrate formation with participation of SSA, may slow down the formation of fine mode nitrate." → ". . . , resulting (or: which resulted) from coarse mode nitrate formation, reduced the formation of fine mode nitrate."

**Response:**

*Thanks for the comment. The sentence has been revised as suggested.*

49. p.9, l.32-33: "Later on, these changes will alter . . . " → "These changes alter . . . ".

**Response:**

*Thanks for the comment. The sentence has been revised as suggested.*

**References:**

[revised manuscript text omitted]

---

## Author Comment (AC2) · 18 Aug 2016

**Response to comments of referee #2**

**General Comments:**

The manuscript "Sea salt emission, transportation and influence on nitrate simulation: a case study in Europe" studies the transport of sea salt aerosol using the WRF-CHEM model and compares the modelling results to measurements obtained during the HOPE-Campaign in September 2013. The meteorology simulations were validated against surface meteorological observations as well as the vertical distribution of meteorological parameters obtained by radiosonde measurements, and both confirmed that the simulation could capture the meteorological condition very well. The aerosol number/mass concentration distribution, however, displayed a large discrepancy in the coarse mode size range, which the author attributes to overestimated SSA emissions in the model emission scheme. The author studies the difference in thermodynamic stratification over land and sea and points out the mechanism for the long-range transport of SSA, which extends the influencing range of SSA further inland to the Melpitz station. The author further studies the effect of overestimated SSA on particulate nitrate simulation results. Here are some general comments:

**Response:**

*Many thanks to the reviewer for the comments and suggestions. We have improved the manuscript accordingly. The language in the manuscript has also been edited throughout.*

*The order of Figures was changed in the revised version manuscript. However, in this response we keep the order consistent (unless specified) with the original version manuscript for easily understood. The changes of the Figures order are shown in Table R1.*

*Table R1. The changing of Figures order in the revised manuscript*

| *Original version* | *Revised version* |
|---|---|
| *Manuscript* | |
| *--* | *Figure 1 (newly added)* |
| *Figure 1* | *Figure 2* |
| *Figure 2* | *Figure 3* |

| | |
|---|---|
| *Figure 3* | *Figure 4* |
| *Figure 4* | *Figure 5* |
| *--* | *Figure 6 (newly added)* |
| *Figure 5* | *Figure 7* |
| *Figure 6* | *Figure 8* |
| *Figure 7* | *Figure 9* |
| *Figure 8* | *Figure 10* |
| *Figure 9* | *Figure 11* |
| ***Supplement*** | |
| *Figure S1* | *Figure S1* |
| *Figure S2* | *Replaced by revised version Figure 1* |
| *Figure S3* | *Figure S2* |
| *Figure S4* | *Figure S3* |
| *--* | *Figure S4 (newly added)* |
| *Figure S5* | *Figure S5* |

(1) The impact of SSA on nitrate partition seems to be nothing new. The author mentions at the end of the conclusions the potential impact of overestimated SSA and nitrate on radiative forcing and aerosol hygroscopicity, it would be perhaps more interesting to see some discussion on that.

**Response:**

*Thank you very much for the comments.*

*We agree that the influence of sea salt on nitrate has been studied in lots of previous studies (Neumann et al., 2016a; Liu et al., 2015; Im, 2013; Athanasopoulou et al., 2008), but mainly focus on the bulk nitrate mass concentrations and did not shown the influence on the nitrate within different size mode (fine mode and coarse mode). In this study, we quantified the sea salt influence on the both fine mode and coarse mode nitrate particles formation respectively.*

*By looking into size-segregated details, we found that sea salt facilitates the coarse mode nitrate particle ($NaNO_3$) formation (as found in most previous studies), but it may inhibit the fine mode nitrate particle ($NH_4NO_3$) formation. This effect can change the particle mass size distribution (PMSD) of nitrate, moves nitrate from fine mode nitrate particles to coarse mode nitrate particles (see Fig. 9), which is crucial for aerosol deposition, hygroscopicity, and optical properties etc. This research could serve as a cornerstone for future detailed research about the impact of sea salt on these properties of nitrate.*

*However, as pointed out by the reviewers, the re-distribution effect of nitrate PMSD due to the participation of SSA was not clearly highlight out in the manuscript. Therefore, in order to emphasize this scientific point, the title, section 3.4, section of Introduction and Figure 9 have been revised. The detailed revisions are shown as following.*

*The title has been revised as suggested by reviewer 1#:*

[revised manuscript text omitted]

*The Figure 9 has been also revised, in order to include this scientific point, as shown below:*

[Figure]

**Figure 9**. *Schematic of sea salt transportation and its influence on nitrate particle formation.*

*A short discussion of the influence of nitrate PMSD re-distribution on the aerosol particle hygroscopicity, deposition and optical properties has been added in the conclusion, as shown below. The detailed evaluation and studies about these further influences will be presented in the further research paper.*

*"these changes will alter the physical and chemical aerosol properties, e.g. particle number/mass size distribution and hygroscopicity, which are crucial for climate change evaluation. Furthermore, the direct and indirect radiative forcing evaluation will also be influenced."* **changed to:**

*"Such changes will also alter the physical and chemical aerosol properties, e.g. particle mass size distribution and hygroscopicity. A nitrate coating on a SSA surface may reduce the hygroscopicity of coarse mode particles, and the re-distribution of nitrate from fine mode to coarse mode may increase its deposition rate. Furthermore, the direct and indirect radiative forcing evaluation will also be influenced, since the optical properties (e.g.: single scattering albedo) are strongly related to the size of particles. All these influences are crucial for climate change evaluation."*

(2) The model output frequency is not clarified in section 3. Did you compare hourly model data with observations? In the comparison of simulated & observed meteorological data, the author calculates correlation coefficient. However, many meteorological parameters, such as temperature and wind, have significant diurnal variations, which can be easily captured in the model. If you calculate correlation coefficients between hourly data, the diurnal variations which agree with each other very well might also lead to high correlation coefficients, which does not necessarily mean that you could capture the day-to-day variation well. Why did you not directly compare the absolute values between model & measurements, especially for the wind direction data?

**Response:**

*Thank you very much for the comments. Yes, as reviewer understood, the output frequency is one hour, and the hourly model data was compared with the observations. This has been clarified in the revised manuscript, as shown below.*

*"Meteorology simulated by WRF frequency was evaluated with the near-ground measurements at Melpitz and radio-sounding measurements all over Europe." **changed to***

*"Meteorology simulated by WRF with hourly output frequency was evaluated with the near-ground measurements at Melpitz and radio-sounding measurements all over Europe."*

*We agree with the reviewer that the agreement of diurnal variations may lead to the high correlation coefficients. And in this study the day-to-day variation was also well captured by the model, as shown in Figure S4. The corresponding sentence has also been revised, as shown below.*

*"Simulated temperature, relative humidity, wind speed and wind direction were in good agreement with measurements, with a correlation coefficients (R) of 0.94, 0.85, 0.86, and 0.86 respectively." **changed to:***

*"Simulated temperature, relative humidity, wind speed and wind direction were in good agreement with Melpitz near-ground hourly measurements (Fig. S4, newly added in the revised version), with a correlation coefficients (R) of 0.94, 0.85, 0.86 and 0.86 respectively, and with mean bias (MB) 0.38 $^{o}C$, 9.1%, -0.18 $m\ s^{-1}$ and 10.62$^{o}$ respectively."*

[Figure]

***Figure S4** (newly added in the revised version). The comparisons between the simulation results and measurements at Melpitz near-ground layer. The correlation coefficient (R) and mean bias (MB) are marked on the top of each panel. (a) Temperature; (b) relative humidity*

*(RH); (c) wind speed; (d) wind direction.*

(3) Although the manuscript is easy to understand, there are still many grammatical errors and the scientific language is not always precise, please go through the whole text carefully and revise the language to improve the reading experience of your readers.

**Response:**

*Thank you very much for the comments. The language has been edited throughout.*

(4) 1. Does the paper address relevant scientific questions within the scope of ACP?

Yes.

2. Does the paper present novel concepts, ideas, tools, or data?

Yes.

3. Are substantial conclusions reached?

Yes.

4. Are the scientific methods and assumptions valid and clearly outlined?

Yes. However, there can be improvements in the methods section.

5. Are the results sufficient to support the interpretations and conclusions?

Yes.

6. Is the description of experiments and calculations sufficiently complete and precise to allow their reproduction by fellow scientists (traceability of results)?

Yes.

7. Do the authors give proper credit to related work and clearly indicate their own new/original contribution?

Yes.

8. Does the title clearly reflect the contents of the paper?

Yes.

9. Does the abstract provide a concise and complete summary?

Yes.

10. Is the overall presentation well structured and clear?

Yes.

11. Is the language fluent and precise?

It is overall fluent, however, improvements are needed to make it more precise.

12. Are mathematical formulae, symbols, abbreviations, and units correctly defined and used?

Yes.

13. Should any parts of the paper (text, formulae, figures, tables) be clarified, reduced, combined, or eliminated?

No.

14. Are the number and quality of references appropriate?

Yes.

15. Is the amount and quality of supplementary material appropriate?

Yes.

**Response:**

*Thank you very much for the comments. The method section has been improved. The language has also been edited throughout.*

**Specific Comments: Abstract**

(1) P1L26: "…, the modeled SSA concentrations were overestimated by a factor of 8-20." → ", the model overestimated SSA concentrations by factors of 8-20."

**Response:**

*Thank you very much for the comments. The sentence has been revised as suggested.*

(2) P1L27: "…over North Sea" → "…over **the** North Sea", this needs also to be corrected for the later occurrences in the manuscript.

**Response:**

*Thank you very much for the comments. The terminology has been corrected as suggested.*

(3) P1L32: "broadened" → "extended"

**Response:**

*Thank you very much for the comments. The word has been corrected as suggested.*

(4) P1L35-36: "increased by about 0.2 for the coarse mode nitrate…., but no significant difference in the partitioning fraction for the fine mode nitrate." → "increased by about **20%** for the coarse mode nitrate…, but no significant difference in the partitioning fraction for the fine mode nitrate **was found**."

**Response:**

*Thank you very much for the comments. The sentence has been revised as suggested.*

**Specific Comments: Introduction**

(1) P1L41: "Atmospheric aerosol plays… Further they have an …" rephrase these two sentences, if you want to use "they", you should change the first sentence to "Atmospheric aerosols…"

**Response:**

*Thank you very much for the comments. The "Atmospheric aerosol…" has been revised to "Atmospheric aerosols…".*

(2) P1L43: change to "on a global scale"

**Response:**

*Thank you very much for the comments. The sentence has been revised as suggested.*

(3) P2L1: "…,  comparable with…"

**Response:**

*Thank you very much for the comments. The sentence has been revised as suggested.*

(4) P2L3-5: Rephrase to "Waves breaking in the surf zone, where **there are** more whitecaps and stronger **SSA (?)** emission due to **increased** ocean bottom and higher intensity of wave breaking, may **affect SSA concentrations at areas within 25 km distance from** the coastline and can dominate **the SSA concentration** at the coastal region"

**Response:**

*Thank you very much for the comments. The sentence has been revised as suggested.*

(5) P2L9-10: "nitrate formation" is slightly inappropriate, since the HNO3 was already formed in the atmosphere. The SSA only influenced its gas and aerosol phase partitioning. Please consider

**Response:**

*Thank you very much for the comments. The terminology has been revised to "nitrate particle formation".*

(6) P2L13-14: Change to "…, sodium nitrate largely contributes to nitrates in northern and southern Europe"

**Response:**

*Thank you very much for the comments. The sentence has been revised as suggested.*

(7) P2L22-23: Change to "…and thereby could expand/extend their influencing range from coastal to regional or even global."

**Response:**

*Thank you very much for the comments. The sentence has been revised as suggested.*

(8) P2L24-25: Change to "However, in terms of global mass concentration, …"

**Response:**

*Thank you very much for the comments. The sentence has been revised as suggested.*

(9) P2L35: Change to "…for the evaluation of the its climate effect"

**Response:**

*Thank you very much for the comments. The sentence has been revised as suggested.*

(10) P2L41-42: Change to "Furthermore, the long-range transport mechanisms, as mentioned above, extends the impact of SSA indirect effect on nitrate formation to a broader region."

**Response:**

*Thank you very much for the comments. The sentence has been revised as suggested.*

(11) P2L44: Rephrase as "The model parameterization schemes…"

**Response:**

*Thank you very much for the comments. The sentence has been revised as suggested.*

(12) P3L1-3: Please change the tense in these three lines to present tense.

**Response:**

*Thank you very much for the comments. The sentences have been revised as suggested.*

**Specific Comments: Section 2**

(1) P3L41: Consider adding the domain range of D01 to Figure 1.

**Response:**

*Thank you very much for the comments. In order to see more detail of our interesting region, we prefer to just show D02 in Figure 1, instead of imbedding D02 inside D01 in Figure 1. However, the range of D01 and its relative location with D02 are given in Chen et al. (2016). We have added this information in the manuscript:*

*"More details on simulation about setups and parameterizations are given in Table 1 and Chen et al. (2016)."*

(2) P4L2: "The spin-up **time** of the model run was 2 days."

**Response:**

*Thank you very much for the comments. The sentence has been revised as suggested.*

(3) P4L8: "More details **on simulation** about setups and parameterizations  are given in Table 1."

**Response:**

*Thank you very much for the comments. The sentence has been revised as suggested.*

(4) P4L10: Rephrase to "SSA are produced through the evaporation of sea sprays, which were ejected into the atmosphere from the sea surface."

**Response:**

*Thank you very much for the comments. The sentence has been revised as suggested.*

(5) P4L12-13: "The parameterization scheme for SSA emission coupled in the WRF-Chem model follows the Gong (2003) scheme."

**Response:**

*Thank you very much for the comments. The sentence has been revised as suggested.*

(6) P4L17: "…, which controls the shape of submicron **SSA** size distributions"

**Response:**

*Thank you very much for the comments. The sentence has been revised as suggested.*

(7) P4L31: "…and has  the same spatial resolution"

**Response:**

*Thank you very much for the comments. The sentence has been revised as suggested.*

(8) P4L42: "Measurements of **the** HOPE-Campaign". The "the" is often missing, please go through the manuscript carefully and make the language more fluent.

**Response:**

*Thank you very much for the comments. The sentence has been revised as suggested.*

(9) P5L3: "The Melpitz Obervatory is representative of  the regional background of Central Europe"

**Response:**

*Thank you very much for the comments. The sentence has been revised as suggested.*

(10) P5L5,9: There are many abbreviations in the text that appear without explaining what they stand for, e.g. WMO-GAW, ACTRIS, MARGA, etc.

**Response:**

*Thank you very much for the comments. The full names of the abbreviations have been added. As shown below:*

*WMO-GAW (World Meteorological Organization – Global Atmospheric Watch);*

*ACTRIS (Aerosols, Clouds, and Trace gases Research InfrastraStructure Network);*

*MARGA (Monitor for AeRosols and GAses in ambient air).*

(11) P5L11-12: "This instrument provided 1-hour data of secondary inorganic aerosols (…) and

gaseous counterparts (…)."    I would suggest adding the detailed species that were measured into these brackets.

**Response:**

*Thank you very much for the comments. The detailed species have been added. As shown below:*

*"secondary inorganic aerosols ($NH_4^+$, $NO_3^-$, $SO_4^{2-}$, $Cl^-$, $Ca^{2+}$, $Mg^{2+}$ and $K^+$) and gaseous counterparts ($NH_3$, $HNO_3$, $HNO_2$, $SO_2$, $HCl$)."*

(12) P5L12-14: Did you have two high volume samplers respectively for PM10 and PM1? If yes, rephrase to: "The high volume samplers DIGITEL DHA-80 (Walter RiemerMesstechnik, Germany), with **a** sampling flow of about 30 m3h−1, **were used to collect 24-hour PM10 and PM1 filter samples simultaneously** (Spindler et al., 2013).

**Response:**

*Thank you very much for the comments. Yes, we have two high volume samplers. Therefore we use the proposed text.*

(13) P5L14-16: "Information on the **coarse mode (PM1-10)** aerosol chemical compositions, such as nitrate and sodium etc.,  were obtained from the difference between the results of PM10 and PM1".

**Response:**

*Thank you very much for the comments. The sentence has been revised as suggested.*

(14) P5L14-16: "Additionally, **24-hour** filter sampler measurements with PM10 inlets (EMEP, 2014) at 3 coastal EMEP station near the SSA transportation pathway (Bilthoven, Vredepeel, and Kollumerwaad, see Fig. 1)**, which were collected every second day,** were obtained from EBAS (http://ebas.nilu.no/)"

**Response:**

*Thank you very much for the comments. We wrote the sentence as proposed.*

**Specific Comments: Section 3**

(1) P5L21: "over  Northern Germany"

**Response:**

*Thank you very much for the comments. The sentence has been revised as suggested.*

(2) P5L25: "Evidently, strong vertical **motion** occurred in the coastal region, which  lifted SSA upward."

**Response:**

*Thank you very much for the comments. The sentence has been revised as suggested.*

(3) P5L28-29: "Simulated **surface** temperature, relative humidity, wind speed and wind direction were in good agreement with **ground** measurements, with a—correlation coefficients…"

**Response:**

*Thank you very much for the comments. The sentence has been revised as suggested.*

(4) P5L36-37: "Corresponding, R values were 0.99, 0.96, 0.84 and 0.92 for potential temperature, wind speed, wind direction and water vapor mixing ratio, respectively." Are these vertically averaged correlation coefficients between simulated vertical profiles and radiosonde measurements? If so, please rephrase the sentence to make that clear.

**Response:**

*Thank you very much for the comments. The sentence has been revised to make it clearer. As shown below:*

*"Corresponding, **the averaged R values of vertical profiles** were 0.99, 0.96, 0.84 and 0.92 for potential temperature, wind speed, wind direction and water vapor mixing ratio, respectively."*

(5) P6L1: Rephrase as "Therefore, unrealistic sources of coarse particles might be the cause for the overestimation."

**Response:**

*Thank you very much for the comments. The sentence has been revised as suggested.*

(6) P6L13-14: "Marine air masses **first** arrived at **the** three coastal stations.

**Response:**

*Thank you very much for the comments. The sentence has been revised as suggested.*

(7) P6L17: "As shown in Fig. 5 the **day-to-day variation** of Na+ concentrations can be captured by the model…"

**Response:**

*Thank you very much for the comments. The sentence has been revised as suggested.*

(8) P6L26-27: "The uncertainties of this scheme may be **attributed** to **the lack** of parameters, …"

**Response:**

*Thank you very much for the comments. The sentence has been revised as suggested.*

(9) P6L32-33: "Generally, SSA is mostly in coarse mode with **a** lifetime shorter than 2 days in the continental boundary layer,  **and reaching** about 1 week in free troposphere"

**Response:**

*Thank you very much for the comments. The sentence has been revised as suggested.*

(10) P6L35: This sentence is hard to understand and needs rephrasing, consider "According to the simulation results, the component of the 10m wind vector that is directed from the coast to Melpitz shows a wind speed in the range of 2-3 m s$^{-1}$"

**Response:**

*Thank you very much for the comments. The sentence has been revised as suggested.*

(11) P6L35-36:"It **would**  therefore **take** about 1.5-2 days for SSA to be transported to Melpitz (~400 km away from coast)."

**Response:**

*Thank you very much for the comments. The sentence has been revised as suggested.*

(12) P6L36-38:The result (Fig. S5) from **the** Deposition-Lifetime Concept Model (Chen et al., 2016; Croft et al., 2014) **indicates** that on average only  10-35% of **the emitted** SSA could be transported to Melpitz through the surface pathway.

**Response:**

*Thank you very much for the comments. The sentence has been revised as suggested.*

(13) P7L6-8: "Therefore, about 70-85% of SSA (Fig. S5) could be carried further towards **the** inland in free troposphere, and arrived **at** the Melpitz region in the early morning of September 17 (Fig. 6b)."

**Response:**

*Thank you very much for the comments. The sentence has been revised as suggested.*

(14) P7L11-12: "As discussed above, the **over-production** of SSA from the WRF-Chem SSA emission scheme will **lead** to an 8-20 times overestimation of the primary sea salt mass concentration."

**Response:**

*Thank you very much for the comments. The sentence has been revised as suggested.*

(15) P7L15: Rephrase to: "Part of HNO3 will be partitioned into the condensed phase and form particulate nitrate."

**Response:**

*Thank you very much for the comments. The sentence has been revised as suggested by the reviewer 1#. As shown below:*

*"The condensed $HNO_3$ deprotonates to $NO_3^-$ ".*

(16) P7L17-18: " The other one is the irreversibely reaction with SSA (NaCl) and the formation of sodium nitrate with depletion of chloride.

**Response:**

*Thank you very much for the comments. The sentence has been revised as suggested.*

(17) P7L21-22: I believe what you want to say is that the condensation process of HNO3 onto particles is facilitated by the participation of SSA, replace "partition" with "condensation": "The participation of SSA might facilitate the **condensation** process of nitrate."

**Response:**

*Thank you very much for the comments. The sentence has been revised as suggested.*

(18) P7L25: "This could either result from an inaccurate emission of precursors or from an improper chemical pathway in the model."

**Response:**

*Thank you very much for the comments. This sentence has been removed as suggested by the reviewer 3#.*

(19) P7L30-34: Please consider rephrasing this part into: " However, **even under the same mass concentrations of precursors, the simulated nitrate mass concentrations (Fig. 7a) were still much higher than the observed ones (Fig. 7b), which indicates that** in addition to an overestimation caused by overestimated **NH3** emission (see also Table 2), improper chemical pathway also contributed to the nitrate overestimation. "

**Response:**

*Thank you very much for the comments. The sentences have been revised as suggested.*

(20) P7L35-36: "In order to quantify the influence of NaCl on the nitrate partitioning, a **sensitivity** study was implemented with only 5% of SSA emission (R-CASE)."

**Response:**

*Thank you very much for the comments. The sentence has been revised as suggested. And according to the suggestion of the reviewer 1#, this sentence has been moved to the section 2 in order to introduce the R-CASE in the method section.*

(21) P7L42: "However, NOx and total ammonia **concentration** results of **the R-CASE did not show significant changes** (Table 2)."

**Response:**

*Thank you very much for the comments. The sentence has been revised as suggested.*

(22) P8L10-13: 1. The later sentence is incomplete; 2. The difference in size range is a reasonable reason why the two should not be directly compared. The uncertainties in measurements and in the model emissions always exist, we need to keep those in mind when comparing measurements with model results, but they are not the reason why the two should not be compared. Consider rephrasing this part into: "Since the MARGA measurements were only available for the size range of PM10, PF_nitrate derived from MARGA observations should not be directly compared with the simulated one. Additionally, we need to keep in mind that high uncertainties exist in the HNO3 measurements due to its sticky property and in the model precursor emissions, which brings further difficulty into the comparison between measurements and simulation."

**Response:**

*Thank you very much for the comments. The sentences have been revised as suggested.*

(23) P8L18-20: This sentence needs rephrasing, consider "As shown in Fig. 8a and Fig. 8b, the median value of coarse mode PF_nitrate in the R-CASE was about 0.75, with the distribution broadly spread in the range of ~0.2 to 1, whereas in the F-CASE the median value increased to 0.96, with a much narrower distribution."

**Response:**

*Thank you very much for the comments. The sentence has been revised as suggested.*

(24) P8L26-27: "Although the fine mode PF_nitrate **revealed** no significant difference between R-CASE and F-CASE simulations…"

**Response:**

*Thank you very much for the comments. The sentence has been revised as suggested.*

**Specific Comments: Conclusion**

(1) P8L39-40: "…, the WRF-Chem model was used to simulate **the aerosol** physical and chemical properties during **the** HOPE Campaign…"

**Response:**

*Thank you very much for the comments. The sentence has been revised as suggested.*

(2) P9L2-4: The overestimate in coarse mode nitrate is also caused by the overestimate in SSA emissions, which is also summarized later on in the following text. I would suggest not to mention it here, rephrase as: "The coarse mode particles were**,** however**,** significantly overestimated both in number and mass, due to an overestimate in SSA emissions caused by the current SSA emission scheme.

**Response:**

*Thank you very much for the comments. The sentence has been revised as suggested.*

(3) P9L6: "The **day-to-day** variations of SSA mass concentrations…"

**Response:**

*Thank you very much for the comments. The sentence has been revised as suggested.*

(4) P9L19-20: Change to "The overestimation in SSA emissions not only influences the primary SSA simulation itself, but also leads to significant uncertainties in the particulate nitrate simulation."

**Response:**

*Thank you very much for the comments. The sentence has been revised as suggested.*

(5) P9L25: "However, the increas**ed** consumption of the gas-phase precursor (HNO3), **caused by the** coarse mode nitrate formation with **the** participation of SSA, may **inhibit/repress/reduce (?)** the formation of fine mode nitrate."

**Response:**

*Thank you very much for the comments. The sentence has been revised as suggested. As shown below:*

*"The nitrate partitioning fraction of fine mode was insensitive to the SSA emission. However, the increased consumption of the gas-phase precursor, caused by the coarse mode nitrate formation with the participation of SSA, may reduce the formation of fine mode nitrate."*

(6) P9L35-39: Change to: "Due to the "aloft bridge" transport mechanism, as described in this paper, the influences of SSA are not only confined to the coastal region, but are extended to a broader region reaching as far as 400 km from coast. Meanwhile, the outflow of continental air mass can transport NOx to the ocean region (Fig. S1), where these influences of SSA on nitrate may also be significant."

**Response:**

*Thank you very much for the comments. The sentences have been revised as suggested.*

---

## Author Comment (AC3) · 18 Aug 2016

**Response to comments of referee #3**

**General Comments:**

The authors apply WRF-Chem to investigate the effect of sea salt on aerosol nitrate concentrations and the transport mechanisms of sea salt aerosol to an inland site. Additionally, the results of the applied WRF-Chem setup are evaluated against observations. Although the impact of sea salt on aerosol nitrate in general is nothing new, the paper includes sufficient novel aspects and interesting details for a publication in ACP. One important finding is the overestimation of sea salt emissions by WRF-Chem's Gong (2003) sea salt emission scheme. Some more in-depth discussion seems desirable here, e.g. how well the wind speed in the source area are represented or how the applied scheme compares against the other sea salt emission schemes which are included in WRF-Chem. The paper is easily comprehensible. However, it includes numerous language lapses, such as wrong usage of singular and plural, missing articles etc. The co-authors are requested to support the lead author here. Also, some of the figures could be improved in some aspects.

**Response:**

*Many thanks to the reviewer for the comments and suggestions. We have improved the manuscript accordingly. The language in the manuscript has also been edited throughout.*

*The order of Figures was changed in the revised version manuscript. However, in this response we keep the order consistent (unless specified) with the original version manuscript for easily understood. The changes of Figures order are shown in Table R1.*

*Table R1. The changing of Figures order in the revised manuscript*

| *Original version* | *Revised version* |
|---|---|
| *Manuscript* | |
| *--* | *Figure 1 (newly added)* |
| *Figure 1* | *Figure 2* |
| *Figure 2* | *Figure 3* |
| *Figure 3* | *Figure 4* |

| | |
|---|---|
| *Figure 4* | *Figure 5* |
| *--* | *Figure 6 (newly added)* |
| *Figure 5* | *Figure 7* |
| *Figure 6* | *Figure 8* |
| *Figure 7* | *Figure 9* |
| *Figure 8* | *Figure 10* |
| *Figure 9* | *Figure 11* |
| ***Supplement*** | |
| *Figure S1* | *Figure S1* |
| *Figure S2* | *Replaced by revised version Figure 1* |
| *Figure S3* | *Figure S2* |
| *Figure S4* | *Figure S3* |
| *--* | *Figure S4 (newly added)* |
| *Figure S5* | *Figure S5* |

(1) Although the impact of sea salt on aerosol nitrate in general is nothing new

**Response:**

*Thank you very much for the comments.*

*We agree that the influence of sea salt on nitrate has been studied in previous studies (Neumann et al., 2016a; Liu et al., 2015; Im, 2013; Athanasopoulou et al., 2008), but mainly focus on the bulk nitrate mass concentrations and did not shown the influence on the nitrate within different size mode (fine mode and coarse mode). In this study, we quantified the sea salt influence on the both fine mode and coarse mode nitrate particles formation respectively. By looking into size-segregated details, we found that sea salt facilitates the coarse mode nitrate particle ($NaNO_3$) formation (as found in most previous studies), but it may inhibit the fine mode nitrate particle ($NH_4NO_3$) formation. This effect can change the particle mass size distribution (PMSD) of nitrate, moves nitrate from fine mode nitrate particles to coarse mode*

*nitrate particles (see Fig. 9), which is crucial for aerosol deposition, hygroscopicity, and optical properties etc. This research could serve as a cornerstone for future detailed research about the impact of sea salt on these properties of nitrate.*

*However, as pointed out by the reviewers, the re-distribution effect of nitrate PMSD due to the participation of SSA was not clearly highlight out in the manuscript. Therefore, in order to emphasize this scientific point, the title, section 3.4, section of Introduction and Figure 9 have been revised. The detailed revisions are shown as following.*

*The title has been revised as suggested by reviewer 1#:*

*"Sea salt emission, transportation and influence on nitrate simulation: a case study in Europe"* **changed to**

*"Sea salt emission, transport and influence on **size-segregated** nitrate simulation: a case study **in Northwestern Europe by WRF-Chem**"*

*One paragraph has been added in Section 3.4 in order to clearly show this effect: the influence of SSA on nitrate PMSD, moving nitrate particle from fine mode to coarse mode. As shown below:*

*"In order to see the influence of SSA on nitrate PMSD in a clearer way, the simulated PMSD during marine period at Melpitz was shown in Fig. 1 (newly added in the revised manuscript). It was clearly shown that the nitrate PMSD decreased in the smaller size bins (bins 01-04) but increased in the larger size bins (bins 05-08). In the F-CASE (Fig. 1b) when the overestimated SSA participated in nitrate particle formation, nitrate particle moved from fine mode to coarse mode compared with the R-CASE (see also Fig. 3)."*

*A paragraph in the Introduction section has been revised, in order to emphasize this scientific point, as shown below:*

*"SSA could participate in heterogeneous reactions by interacting with trace gases, leading to the formation of secondary aerosols (Seinfeld, 2006), such as nitrate, which is one of the most important secondary inorganic aerosol and is the dominant aerosol component in western and central Europe (Schaap et al., 2011). SSA has a significant influence on nitrate formation as shown in previous studies (Neumann et al., 2016a; Liu et al., 2015; Im, 2013; Athanasopoulou et al., 2008). Sodium nitrate is produced with a chloride deficit in the SSA (Schaap et al., 2011; Seinfeld, 2006), and the timescale of the corresponding reaction is about several hours (Meng and Seinfeld, 1996). As reported in previous studies, sodium nitrate is*

*largely contributed to nitrates in northern and southern Europe (Pakkanen et al., 1999), whereas in western and central Europe ammonium nitrate dominates (Schaap et al., 2002; ten Brink et al., 1997)."* **changed to:**

*"SSA can participate in heterogeneous reactions by interacting with trace gases, leading to the formation of secondary aerosols (Seinfeld, 2006), such as nitrate, which is one of the most important secondary inorganic aerosol and is the dominant aerosol component in western and central Europe (Schaap et al., 2011). SSA can also facilitate the formation of nitrate aerosol (Neumann et al., 2016a; Liu et al., 2015; Im, 2013; Athanasopoulou et al., 2008). However, these previous studies mainly focused on the influence of SSA on bulk nitrate mass concentration, and did not address its influence on size-segregated nitrate particles. In this study, we quantified the SSA influence on both fine mode and coarse mode nitrate particles formation respectively. and the effect could be different for the different size mode, resulting from the heterogeneous reaction on SSA surface with the formation of sodium nitrate. The timescale of this reaction is considered to be several hours (Meng and Seinfeld, 1996). Sodium nitrate is produced with a chloride displacement in the SSA (Schaap et al., 2011; Seinfeld, 2006). Importantly, thermodynamically stable sodium nitrate will not return to the gas phase as the semi-volatile ammonium nitrate does (Schaap et al., 2011). According to previous studies, sodium nitrate largely contributes to nitrates in northern and southern Europe (Pakkanen et al., 1999), whereas in western and central Europe ammonium nitrate dominates (Schaap et al., 2002; ten Brink et al., 1997). The reason is enhanced ammonia emission from husbandry and agricultural sources in central and western Europe (Backes et al., 2016b;Backes et al., 2016a)."*

[Figure]

*Figure 1 (newly added in the revised manuscript). WRF-Chem simulation results of particle mass size distribution (PMSD) for each chemical compounds, during marine period at Melpitz. (a) result of the R-CASE; (b) result of the F-CASE. The difference of nitrate PMSD between the R-CASE and the F-CASE for each bin is marked.*

*The Figure 9 has been also revised, in order to include this scientific point, as shown below:*

[Figure]

*Figure 9. Schematic of sea salt transportation and its influence on nitrate particle formation.*

(2) how well the wind speed in the source area are represented

**Response:**

*The ground wind measurements in the source area (over the North Sea) are not available. However, we compared the radio-sounding measurements of wind speed over the Europe, including one station over the North Sea and some coastal stations, which are close to the source area. As shown in Fig. 2b (newly added in the revised version). And the following paragraph has been added in the section 3.1 to discuss about the wind speed simulation.*

*"The vertical pattern of wind speed was also captured by the model, especially well captured over the North Sea and coastal regions (see Fig. 2b). Generally, the correlation coefficient (R) values were higher than 0.6, with the value higher than 0.9 over the SSA emission source area (the North Sea) and coastal regions."*

(a)

[Figure]

[Figure]

(b)

***Figure 2.*** *Correlation coefficient (R) map between WRF-Chem model and radio-sounding measurements (0-3 km). Melpitz is marked as red star. (a) potential temperature; (b) wind speed. Note that the panels (a) and (b) have the different color-bar scale in order to show more details.*

(3) how the applied scheme compares against the other sea salt emission schemes which are included in WRF-Chem.

**Response:**

*Thank you very much for the comments. Unfortunately, there is only Gong (2003) sea salt emission scheme (GO03) is included in WRF-Chem currently. And this scheme is not only coupled with the regional model WRF-Chem, but also with some global models such as GEOS-Chem. Therefore, we think it is important to point out the uncertainties of this GO03 scheme.*

*Jaeglé et al. (2011) also reported that GO03 overestimated coarse mode sea salt by a factor of 2–3 at high wind speeds over the cold waters of the South Pacific, North Pacific and North Atlantic Oceans, by GEOS-Chem model. The comparisons of GO03 with other sea salt emission functions have also been discussed in Jaeglé et al. (2011). The other emission functions include: (1) adjustment with quadratic wind speed dependence; (2) adjustment with*

*sea surface temperature dependence; (3) Monahan et al. (1986) scheme; and (4) Clarke et al. (2006) scheme; (5) Mårtensson et al. (2003) scheme. However we should keep in mind that, with the coarse spatial resolution in global model simulation, the detailed PBL structure cannot be properly captured. And this "aloft bridge" transport mechanism, which reported in our research, may be neglected by the global model. Also, the chemical and physical properties of size-resolved aerosol particles cannot be represented in detail by the global model simulation. So, the impact of sea salt on nitrate PMSD was not investigated in the global model studies. In this paper, we would like to investigate the impact of sea salt on nitrate PMSD. And we also would like to introduce a long-range transport mechanism which could expand this impact to the further inland regions, instead of be confined to the coastal regions.*

**Detailed Comments:**

(1) P 5, l 30: How were the correlations calculated, from hourly values or from mean values? How well are spatial patterns represented? Please discuss also absolute error or mean bias.

**Response:**

*Thanks for the comments. The corresponding discussion has been revised, and as suggested by reviewer 2# a new figure (Figure S4, newly added in the revised version) has been added. And the discussion as shown below:*

*"Simulated temperature, relative humidity, wind speed and wind direction were in good agreement with measurements, with a correlation coefficients (R) of 0.94, 0.85, 0.86, and 0.86 respectively."* **changed to:**

*"Simulated temperature, relative humidity, wind speed and wind direction were in good agreement with Melpitz near-ground hourly measurements (Fig. S4, newly added in the revised version), with a correlation coefficients (R) of 0.94, 0.85, 0.86 and 0.86 respectively, and with mean bias (MB) 0.38 $^oC$, 9.1%, -0.18 m s$^{-1}$ and 10.62$^o$ respectively."*

[Figure]

*Figure S4 (newly added in the revised version). The comparisons between the simulation results and measurements at Melpitz near-ground layer. The correlation coefficient (R) and mean bias (MB) are marked on the top of each panel. (a) Temperature; (b) relative humidity (RH); (c) wind speed; (d) wind direction.*

*The spatial patterns are also well represented by the model, as discussed in General Comments Point-2.*

(2) P 5, l 42: How well match observed and simulated concentrations of the small particles?

**Response:**

*Thanks for the comments. The sentence has been revised to discuss the results of the small particles. As shown following:*

*"The model significantly overestimated the concentration for the size bins 05-08 (625-10,000 nm)." changed to:*

*"Although the simulation of PNSD/PMSD for size bins 01-04 not exactly matched with the measurements, the agreement is in the reasonable range with a factor of ~2 (Fig. 3). But the model significantly overestimated the concentration for the size bins 05-08 (625-10,000 nm) in the F-CASE."*

[Figure]

**Figure 3.** *Comparison of Particle Number Size Distribution (PNSD, left) and Particle Mass Size Distribution (PMSD, right) between the simulations and Melpitz measurements. The results are averaged during September 16-20, 2013; the error bars indicate the upper and lower limits.*

(3) P 6, l 1 and2: Please give some more evidence for this statement.

**Response:**

*Thanks for the comments. The statement has been revised as suggested by the reviewer 2#. As shown below:*

*"Since the meteorology was well reproduced by the model, it can be assumed that the air movement was also reasonably simulated. As a consequence, there might be unrealistic high sources of coarse particles leading to the overestimation."* **changed to**

*"Since the meteorology was well reproduced by the model, it can be assumed that the air movement was also reasonably simulated. Therefore, unrealistic high sources of coarse particles might be the cause for the overestimation, which would be discussed in following."*

(4) P 6, l 43: How was the PBL height estimated?

**Response:**

*Thanks for the comments. The YSU (Hong, 2006)PBL scheme is used in this WRF-Chem study. The PBL height in YSU boundary layer scheme is related to the turbulence diffusion, which keeps the basic concept of HP96 (Hong and L., 1996) but additionally includes an asymptotic*

*entrainment flux term at the inversion layer. And the definition of PBL height in YSU scheme is shown in following, as described by the developer:*

*"The PBL height is defined as the level in which minimum flux exists at the inversion level, whereas in HP96 it is defined as the level that boundary layer turbulent mixing diminishes."*

*More detailed information about this YSU boundary layer scheme is given in (Hong, 2006)*

(5) P 7, l 9: According to Fig. 6b, the sea salt layer does not yet touch the surface. What is the contribution of turbulent mixing after sunrise?

**Response:**

*It is a very good question. One figure (Fig. 6c) and the corresponding discussion have been added, in order to answer this question. As shown following:*

[Figure]

**Figure 6.** *WRF-Chem result of the sea salt ($Na^+$) concentration on the vertical cross section, which is shown by the black dash line in Figure 1. The locations of Melpitz and coast (black line) are marked. The grey arrows indicate the wind field, and the black dash line indicates*

*the planetary boundary layer (PBL) height. (a) 2013-09-16, 04:00 LT; (b) 2013-09-17, 02:00 LT; (c) 2013-09-17, 14:00 LT.*

*"Then the downward draft, resulted from high-pressure ridge, brought the lofted SSA back into the surface layer (Fig. 6b)."* **changed to**

*"Then the downward draft resulted from high-pressure ridge and the turbulent mixing after sunrise (Fig. 6b and 6c), brought the lofted SSA back into the surface layer. The $Na^+$ mass concentration at Melpitz surface increased from ~7 μg/m$^3$ (Fig. 6b) to ~15 μg/m$^3$ (Fig. 6c). About 35% of the lofted SSA contributed to the increase of the $Na^+$ surface concentration. This result is agreement with the previous study (Chen et al., 2009), which reported ~30% of elevated pollutants contributed to the increase of surface pollutants concentration in Beijing, due to the turbulent mixing after sunrise."*

(6) P 7, l 25 and 26: There could be also some other reasons, wrong turbulent exchange, wrong water uptake (also due to wrong relative humidity), . . .

**Response:**

*Thanks for the comments. Yes, the reviewer is correct. The corresponding sentence has been removed.*

(7) P 7, L 29: Why can this be expected?

**Response:**

*Thanks for the comments. It is because: assuming that the chemical mechanism is correctly described in the model, the same concentration of gaseous precursors (NOx and ammonia) should produce the same concentration of nitrate in the model and observation. In order to describe this point more clearly, the paragraph has been revised as suggested by reviewer 2#. As shown below:*

*"The location of the data dots (Fig. 7a) may be shifted due to the uncertainty of precursors emissions, but the nitrate mass concentration is always expected to be consistent with the observed concentration in Fig. 7b. The difference between Fig. 7a and Fig. 7b indicates that*

*in addition to an overestimation caused by overestimated NH₃ emission (see also Table 2), improper chemical pathway also contributed to the nitrate overestimation. Since the simulated nitrate mass concentrations (Fig. 7a) were still much higher than the observed one (Fig. 7b), even though where had the same mass concentrations of precursors." **changed to***

*"The location of the data dots (Fig. 7a) may be shifted due to the uncertainty of precursors emissions, but the nitrate mass concentration is always expected to be consistent with the observed concentration in Fig. 7b when they have the same mass concentration of precursors. However, even under the same mass concentrations of precursors, the simulated nitrate mass concentrations (Fig. 7a) were still significantly higher than the observed ones (Fig. 7b). This indicated that in addition to an overestimation caused by overestimated NH3 (see also Table 2), improper chemical pathway in the model also contributed to the nitrate overestimation."*

[Figure]

***Figure 7.*** *Relationship between nitrate, total ammonia and NOx during September 10-20, 2013 at Melpitz. The color indicates the nitrate mass concentration in logarithmic scale. (a) WRF-Chem model results; (b) MARGA measurement results.*

(8) P 7, l 1 – 10: Please change the order of the figure, Figure 9 should be discussed here.

**Response:**

*Thanks for the comments. The Figure 9 has been revised to include the impact of sea salt on nitrate particle mass size distribution (PMSD), resulting from the heterogeneous reaction on sea salt surface. The revised Figure 9 (please find it in General Comments Point-1) would help this manuscript make a more clear conclusion and connect the scientific points together. Therefore, we move the revised Figure 9 to the end of section 3.4. As shown below:*

*"In general and as illustrated in Fig. 9, the overestimation of SSA emission scheme has a significant influence on the particulate nitrate simulation in both the coarse mode (directly) and the fine mode (indirectly)."*

(9) P 8, l 8: Is this really a probability distribution or a frequency distribution?

**Response:**

*Thanks for the comment. It was really a probability density function. But now, we have modified it as suggested by the reviewer 1#. The "probability density function" has been replaced by "frequency distribution", the patterns are the same between them two. As shown below:*

[Figure]

***Figure 8 (revised).*** *WRF-Chem results of the frequency distribution of PF_nitrate at Melpitz. The result was analyzed during the marine period ($[Na^+] > 1.8 \ \mu g/m^3$ in the F-CASE). The dash lines (coarse mode: red; fine mode: blue) indicate the median value (with 50% probability in both sides). (a) $PM_{1-10}$ result of 5% SSA emission (R-CASE); (b) $PM_{1-10}$ result of the F-CASE; (c) $PM_1$ result of the R-CASE; (d) $PM_1$ result of the F-CASE.*

[Figure]

*Figure 8 (original). WRF-Chem results of the probability density function of nitrate partitioning fraction (PF_nitrate) at Melpitz in the marine period during September 10-20, 2013. The marine period is defined as the $Na^+$ mass concentration higher than 1.8 μg/m³ in the F-CASE. The blue dash lines indicate the median value (with 50% probability in both sides). (a) $PM_{1-10}$ result of 5% SSA emission (R-CASE); (b) $PM_{1-10}$ result of the F-CASE; (c) $PM_1$ result of the R-CASE; (d) $PM_1$ result of the F-CASE.*

(10) Figure 1 and Figure 6: Please consider using a different color scheme. In particular, the dark blue color for the low values is quite unfavorable and the blue arrows (and the map in Fig. 1) can hardly be recognized.

**Response:**

*Thanks for the comments. The color schemes in Figure 1 and Figure 6 have been changed. Please find the revised Figure 6 in the Detailed Comments Point-5. And the revised Figure 1 is shown below:*

[Figure]

**Figure 1.** *The horizontal distribution of surface Na$^+$ mass concentration in domain 02 (intermediate domain) at 2013-09-16, 09:00 LT. The grey arrows indicate the wind field. The locations of 4 EMEP stations (Melpitz, Bilthoven, Kollumerwaard and Vredepeel) are marked. The vertical cross section of dash black line is shown in Figure 6.*

(11) Figure 3: Please show also the R-case.

**Response:**

*Thanks for the comment. The R-CASE result has been added as suggested. Please find the revised Figure 3 in the Detailed Comments Point-2. Comparison between the F-CASE and the R-CASE in Figure 3 also partly supported the scientific point, that the overestimation of SSA inhibited the fine mode nitrate particle formation. This information has been added in section 3.4, as shown below:*

*"In order to see the influence of SSA on nitrate PMSD in a more clear way, the simulated PMSD during marine period at Melpitz was shown in Fig. 1 (newly added in the revised manuscript). It was clearly shown that the nitrate PMSD decreased in the smaller size bins*

*(bins 01-04) but increased in the larger size bins (bins 05-08). In the F-CASE (Fig. 1b) when the overestimated SSA participated in nitrate particle formation, nitrate particle moved from fine mode to coarse mode compared with the R-CASE (see also Fig. 3)."*

(12) Caption of Fig. 4: Please mention which case is shown.

**Response:**

*Thanks for the comment. The caption of Figure 4 has been revised as shown below:*

[Figure]

*Figure 4. Comparison of coarse mode aerosol ($PM_{1-10}$) chemistry compounds between the F-CASE results and Melpitz measurements. (a) averaged during the HOPE-Campaign period of September 10-20, 2013; (b) averaged during the marine air mass period of September 16-20, 2013.*

(13) Caption of Fig. 5: Please mention particle size. Please mention the different scale for observations and model results.

**Response:**

*Thanks for the comment. The caption of Figure 5 has been revised as shown below:*

[Figure]

*Figure 5. Comparison of Na⁺ mass concentration **in PM₁₀** between the filter sampler measurements **(left y-axis)** in 4 EMEP stations and **the F-CASE** results **(right y-axis).** (a) Bilthoven; (b) Kollumerwaard; (c) Vredepeel; (d) Melpitz. The locations of stations are shown in Figure 1.*

(14) Figure 6 and (current) Figure 9: These figures should be oriented from West (left) to East (right). No need for the star, as Melpitz is located at the Eastern end of the figures.

**Response:**

*Thanks for the comment. The Figure 6 and Figure 9 have been revised as suggested by reviewer and the reviewer 1#. Please find the revised Figure 6 in the Detailed Comments Point-5, and the revised Figure 9 in the General Comments Point-1.*

**Minor issues:**

(1) P 2, l7: Partitioning is no 'formation'.

**Response:**

*Thanks for the comment. The sentence has been revised as shown below:*

*"SSA could participate in heterogeneous reactions by interacting with trace gases, leading to the formation of secondary aerosols (Seinfeld, 2006)," **changed to***

*"SSA could participate in heterogeneous reactions by interacting with trace gases, leading to the formation of secondary aerosol particles on SSA surface (Seinfeld, 2006),"*

(2) P 2, l 13: '. . . sodium nitrate is largely contributed to nitrates': please reword.

**Response:**

*Thanks for the comment. The sentence has been revised as suggested by the reviewers 1&2#. As shown below:*

*"sodium nitrate is largely contributed to nitrates in northern and southern Europe" **changed to:***

*"sodium nitrate largely contributes to nitrates in northern and southern Europe"*

(3) P 2, l 21, 22: 'opportunity' and 'make their influence more extensive': please reword

**Response:**

*Thanks for the comment. The sentence has been revised as suggested by the reviewer 2#, as shown below:*

*"These mechanisms provide an opportunity for SSA to be transported inland, and thereby make their influence more extensive, from coastal to regional or even global." **changed to***

*"These mechanisms provide make the long-range transportation of SSA easier, and thereby could expand their influence range from coast to a broader region."*

P 2, l 33: Southern ???

**Response:**

*Thanks for the comment. The "Southern" has been revised to "South Pacific"*

P 2, l 43: influence on what?

**Response:**

*Thanks for the comment. The "influence of SSA" has been revised to "influence of SSA on the size resolved nitrate particle formation".*

P 3, l 27: Please mention first that a resistance approach is applied.

**Response:**

*Thanks for the comment. The sentence has been revised as shown below:*

*"The dry deposition of particles is calculated on the basis of the sublayer resistance, aerodynamic resistance and surface resistance (Grell et al., 2005)." **changed to***

*"The dry deposition of particles is calculated by **a resistance approach**, including sublayer resistance, aerodynamic resistance and surface resistance (Grell et al., 2005)."*

P 6, l 10: Please reword: an event cannot be emitted.

**Response:**

*Thanks for the comment. The "abrupt SSA event was found to be emitted over the North Sea" has been revised to "abrupt SSA emission event happened over the North Sea".*

P 7, l 33: A word seems to be missing here.

**Response:**

*Thanks for the comment. The sentence has been revised as suggested by the reviewer 2#. As shown below:*

[revised manuscript text omitted]

---

## Author Response (AR2)

**Response to Editor:**

Dear ACP Co-Editor,

We would like to thank the two referees for their helpful comments, which have been fully taken into account upon manuscript revision. A point-by-point response to all the comments and a marked-up manuscript version are shown below.

Best Regards,

Ying Chen

**Response to comments of referee #1**

**Minor Comments:**

1. How was the AOD calculated for the model data? A reference is fine. The formula does not need to be included.

**Response:**

*Thanks for the comment. The description of AOD calculation in WRF-Chem has been added in the section 2.1, as shown following.*

*"The aerosol optical depth (AOD) is online calculated in WRF-Chem model by integrating extinction coefficients over all vertical layers, and details are given in Barnard et al. (2010). In general, an internal mixture of all chemical constituents is assumed. The bulk refractive indices for each particle size bin are obtained by a mixing rule based on volume-weighted averaging. The aerosol particle optical properties, such as particle extinction and scattering cross-sections and asymmetry factor, are calculated online by a Mie code described in Ghan et al. (2001)."*

2. p. l19. subscript missing: "PM10"

**Response:**

*Thanks for the comment. The subscript of "PM10" has been corrected and checked throughout the manuscript.*

3. Please include a brief conclusion from the AOD comparison in the conclusion (maybe I also overlooked it).

**Response:**

*Thanks for the comment. The brief conclusion from the AOD comparison has been added in the conclusion section, as shown following.*

*"With reduction of SSA emission to 10%, the simulated AOD results showed a better agreement with the AERONET AOD measurement over Europe. The correlation coefficient (R) between AOD simulation and measurement increased from 0.56 to 0.64, and the geometric mean ratio (GMR) decreased from 2.3 to 1.8."*

**Reference:**

Barnard, J. C., Fast, J. D., Paredes-Miranda, G., Arnott, W. P., and Laskin, A.: Technical Note: Evaluation of the WRF-Chem "Aerosol Chemical to Aerosol Optical Properties" Module using data from the MILAGRO campaign, Atmos. Chem. Phys., 10, 7325–7340, doi:10.5194/acp-10-7325-2010, 2010.

Ghan, S., Laulainen, N., Easter, R., Wagener, R., Nemesure, S., Chapman, E., Zhang, Y., and Leung, R.: Evaluation of aerosol direct radiative forcing in MIRAGE, J. Geophys. Res., 106, 5295–5316, 2001.

**Response to comments of referee #2**

**Minor Comments:**

1. Response to <(2) how well the wind speed in the source area are represented>:

Correlation alone is not a useful measure in this context. Therefore, please add some more information on the wind speed bias (in particular for the coastal region).

**Response:**

*Thanks for the comment. The normalized mean bias (NMB, Balzarini et al., 2015) between radio-sounding and simulated results of wind speed and potential temperature has been added in Fig. 3. And the corresponding text has been revised accordingly, as shown following.*

*"Fig. 3 shows the maps of R values for potential temperature and wind speed between the simulated and measured vertical profile in planetary boundary layer (PBL, 0-3 km). High values of R for potential temperature vertical profile were found at all stations, especially near Melpitz, Germany (R > 0.85) and the coast of North Sea (R > 0.95). The vertical pattern of wind speed was also captured by the model, especially well captured over the North Sea and coastal regions (see Fig. 3b). Generally, the R values were higher than 0.6, with the value higher than 0.9 over the SSA emission source area (the North Sea) and coastal regions."* **changed to**

*"Fig. 3 shows the maps of R values **and normalized mean bias** (**NMB, Balzarini et al. 2015**) for potential temperature and wind speed between the simulated and measured vertical profile in planetary boundary layer (PBL, 0-3 km). High values of R and **low absolute values of NMB** for potential temperature vertical profile were found at all stations, especially near Melpitz, Germany (R > 0.85 **and the absolute NMB < 2%** ), and **the coastal region** of North Sea (R > 0.95 **and the absolute NMB < 2%**). The vertical pattern of wind speed was also captured by the model, especially over the North Sea and coastal regions (see **Fig. 3b and Fig. 3d**). Generally, the R values were higher than 0.6. **Especially, the R values were** higher than 0.9 over the SSA emission source area (the North Sea) and coastal regions**; and the absolute NMB values were lower than 5% over Melpitz region and the coastal region of North Sea.**"*

[Figure]

(a)

[Figure]

(b)

[Figure]

**Figure 3.** Comparison between WRF-Chem model and radio-sounding measurements (0-3 km). Melpitz is marked as red star. (a) R map of potential temperature; (b) R map of wind speed; (c) NMB map of potential temperature; (d) NMB map of wind speed. Note that the panels have the different color-bar scale.

2. Response to <(1) P 2, l7: Partitioning is no 'formation'.>:

Is the number of particles really increased by this process (as the wording in the paper does imply). Or is just the aerosol mass increased due to the newly attached secondary material without increase of the aerosol number? Please clarify.

**Response:**
*Thanks for the comment. Here, the reference (Seinfeld, 2006) means that just the aerosol mass increased due to the newly attached secondary material without increase of the aerosol number. The corresponding sentence has been revised, as shown following.*

*"SSA can participate in heterogeneous reactions by interacting with trace gases, leading to the formation of secondary aerosol particles on SSA surface (Seinfeld, 2006), such as nitrate, which is one of the most important secondary inorganic aerosol and is the dominant aerosol component in western and central Europe (Schaap et al., 2011)." **changed to***

*"SSA can participate in heterogeneous reactions by interacting with trace gases, leading to the formation of **particulate nitrate on SSA surface and increase nitrate particle mass concentration** (Seinfeld, 2006). **Nitrate is one of the** most important secondary inorganic aerosol and is the dominant aerosol component in western and central Europe (Schaap et al., 2011)."*

3. There are still a few language lapses, which will hopefully be removed during copy-editing.

**Response:**
*Thanks for the comment. The language in the manuscript has been edited throughout, and the lapses have been corrected.*

**Reference:**

[revised manuscript text omitted]